# Coupling photocatalytic CO$_2$ reduction and CH$_3$OH oxidation for selective dimethoxymethane production

Yixuan Wang [1,2,3,11], Yang Liu[1,2,11], Lingling Wang[1,2], Silambarasan Perumal [1,2,3], Hongdan Wang[1,2], Hyun Ko[4], Chung-Li Dong [5], Panpan Zhang [6], Shuaijun Wang[7], Ta Thi Thuy Nga [5], Young Dok Kim [1], Yujing Ji [1], Shufang Zhao[1], Ji-Hee Kim[8], Dong-Yub Yee[8], Yosep Hwang [1,2], Jinqiang Zhang [9], Min Gyu Kim [10] & Hyoyoung Lee [1,2,3,4] ✉

Currently, conventional dimethoxymethane synthesis methods are environmentally unfriendly. Here, we report a photo-redox catalysis system to generate dimethoxymethane using a silver and tungsten co-modified blue titanium dioxide catalyst (Ag.W-BTO) by coupling CO$_2$ reduction and CH$_3$OH oxidation under mild conditions. The Ag.W-BTO structure and its electron and hole transfer are comprehensively investigated by combining advanced characterizations and theoretical studies. Strikingly, Ag.W-BTO achieve a record photocatalytic activity of 5702.49 μmol g$^{-1}$ with 92.08% dimethoxymethane selectivity in 9 h of ultraviolet-visible irradiation without sacrificial agents. Systematic isotope labeling experiments, in-situ diffuse reflectance infrared Fourier-transform analysis, and theoretical calculations reveal that the Ag and W species respectively catalyze CO$_2$ conversion to *CH$_2$O and CH$_3$OH oxidation to *CH$_3$O. Subsequently, an asymmetric carbon-oxygen coupling process between these two crucial intermediates produces dimethoxymethane. This work presents a CO$_2$ photocatalytic reduction system for multi-carbon production to meet the objectives of sustainable economic development and carbon neutrality.

Dimethoxymethane (DMM, CH$_3$O·CH$_2$-OCH$_3$) is an attractive compound for numerous applications, including its use as a fuel additive that can enhance diesel fuel yields and as a precursor of oxymethylene dimethyl ether[1–3]. Meeting the growing demand for DMM relies in large part on an indirect synthesis route (Fig. 1a)[4,5]. Specifically, methanol (CH$_3$OH) is oxidized by oxygen (O$_2$) to create formaldehyde (O = CH$_2$), which is then combined with other two CH$_3$OH molecules to produce DMM. Despite the economic efficiency of the process, it necessitates complex operating conditions and inevitably leads to equipment corrosion due to the requirement for strongly acidic catalysts. In

[1]Department of Chemistry, Sungkyunkwan University, 2066 Seobu-Ro, Suwon 16419, Republic of Korea. [2]Creative Research Institute, Sungkyunkwan University, 2066 Seobu-Ro, Suwon 16419, Republic of Korea. [3]CO2 to Multicarbon Production Center, Sungkyunkwan University, 2066 Seobu-Ro, Suwon 16419, Republic of Korea. [4]Institute of Quantum Biophysics, Sungkyunkwan University, 2066 Seobu-Ro, Suwon 16419, Republic of Korea. [5]Department of Physics, Tamkang University, New Taipei City 25137, Taiwan. [6]School of Material Science and Engineering, Jiangsu University, Zhenjiang 212013, People's Republic of China. [7]School of Energy and Power Engineering, Jiangsu University, Zhenjiang 212013, People's Republic of China. [8]Department of Energy Science, Sungkyunkwan University, 2066 Seobu-Ro, Suwon 16419, Republic of Korea. [9]School of Chemical Engineering, The University of Adelaide, Adelaide, SA 5005, Australia. [10]Beamline Research Division, Pohang Accelerator Laboratory, Pohang University of Science and Technology, Pohang 37673, Republic of Korea. [11]These authors contributed equally: Yixuan Wang, Yang Liu. ✉e-mail: hyoyoung@skku.edu

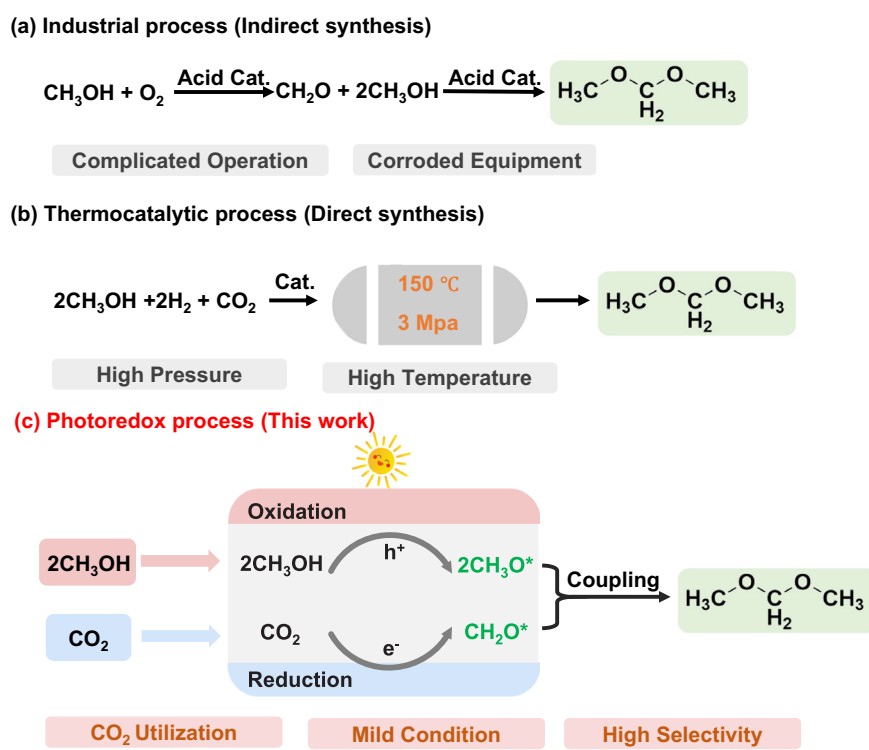

**Fig. 1 | Comparison of the different DMM synthesis route. a** Industrial (indirect) route. **b** Current thermocatalytic (direct) process. **c** Photoredox route.

addition, undesirable peroxide products and carbon oxides are easily produced due to the strong oxidation ability of $O_2$[6,7]. To avoid by-product generation associated with the $CH_3OH$ oxidation route, the $CH_3OH$ dehydrogenation to DMM was explored. The synthesis of DMM via dehydrogenation entails the non-oxidative conversion of methanol into the formaldehyde intermediate, followed by subsequent acetalization reaction of formaldehyde with methanol to yield DMM. For example, Palkovits et al. reached over 80% selectivity of DMM over the Cu/zeolite catalyst under a gas-phase reactor[8]. Most recently, To et al. achieved 40% of the DMM equilibrium-limited yield under mild conditions (200 °C, 1.7 atm) based on Cu-zirconia-alumina (Cu/ZrAlO) catalyst[9]. Nonetheless, the primary challenge is the requirement for harsh conditions, such as high reaction temperatures, to overcome the thermodynamic constraints of gas-phase $CH_3OH$ dehydrogenation.

Since carbon dioxide ($CO_2$) emissions contribute to global warming[10,11], the conversion of $CO_2$ to a high-value chemical with a lower global warming potential could make a substantial contribution to the global effort to mitigate climate change[12]. Recently, $CO_2$ has been considered a feedstock for DMM production through a more environmentally friendly process, one of the most popular approaches is direct synthesis by reacting $CO_2$ and $CH_3OH$. For example, $CO_2$, a weak oxidizer, can selectively oxidize $CH_3OH$ to produce DMM at 150 °C and 3 Mpa (Fig. 1b)[13]. Although such strategies can alleviate environmental concerns and minimize equipment corrosion, synthetic routes require high temperatures and pressures. Alternatively, synthesizing value-added chemicals from a $CO_2$ reduction reaction ($CO_2$RR) using solar energy offers multiple advantages, including minimal pollution, low costs, and ease of operation[14]. In a conventional photocatalytic $CO_2$RR process, $CO_2$ is regularly coupled with water ($H_2O$) to obtain valuable chemicals[15,16]. However, due to the high overpotential of $H_2O$ oxidation to $O_2$ (1.23 V vs. a reversible hydrogen electrode (RHE)), fewer electrons are used, resulting in slower reaction kinetics[17]. $H_2O$ can acquire electrons to produce superoxide ($O_2^{·-}$) or hydroxyl radicals (·OH), which can compete with targeted $CO_2$RRs. Electron donors such as triethanolamine and isopropyl alcohol are

commonly used as sacrificial agents to capture holes and enhance $CO_2$ photoreduction efficiency[18]. Drawbacks of this approach include wasted hole energy, increased system costs, and the transport of useless oxidation products. To tackle these problems, a dual-function, photoredox system offers an appealing option for $CO_2$RR[19]. Organic substrates can potentially substitute for $H_2O$ and hole scavengers in the generating value-added chemicals. Simultaneously, they can contribute by providing the necessary reducing equivalents in the form of protons and electrons to enhance the activation and reduction of $CO_2$, thereby bolstering the stability and overall catalytic efficiency of interconnected reaction systems[17,20]. In this case, coupling photoredox in $CO_2$RR with an organic oxidation reaction ($CH_3OH$ oxidation reaction, MOR) to generate DMM under mild-range conditions appears to be a promising strategy. This is mainly attributed to the lower MOR potential (0.58 V vs. RHE), which results in high reaction efficiency and minimizes environmental impact[21]. Also, the catalytic efficiency of photocatalysis is not as high as that of thermal catalysis. There is still a need to explore photocatalysts with high selectivity and catalytic activity. To our knowledge, few reports of the synthesis of DMM with high selectivity using a photo-redox scheme involving $CO_2$ and $CH_3OH$ have been published. The mechanism and key intermediates responsible for photo-redoxing this carbon-oxygen (C − O) coupling reaction process are also unknown.

Here, we propose that DMM can be produced by coupling $CO_2$RR with MOR in a photo-redox system under mild-range conditions (Fig. 1c). Due to its strong ability to absorb visible light, blue titanium dioxide (BTO) derived from lithium-EDA-treated titanium dioxide ($TiO_2$) was selected as a substrate[22,23]. Deposition of silver (Ag) and doping with a tungsten (W) species were introduced to the BTO (Ag.W-BTO) to create reduction and oxidation sites, respectively, for the dual-functional catalyst. As a result, the DMM yield reached 5702.49 μmol $g^{-1}$ after 9 h ultraviolet (UV)-visible irradiation, and the DMM selectivity approached 92.08% without using sacrificial agents. The photo-induced charge transfer and potential mechanism were systematically explored by femtosecond transient absorption spectroscopy (fs-TA), in-situ diffuse reflectance infrared Fourier-transform spectroscopy

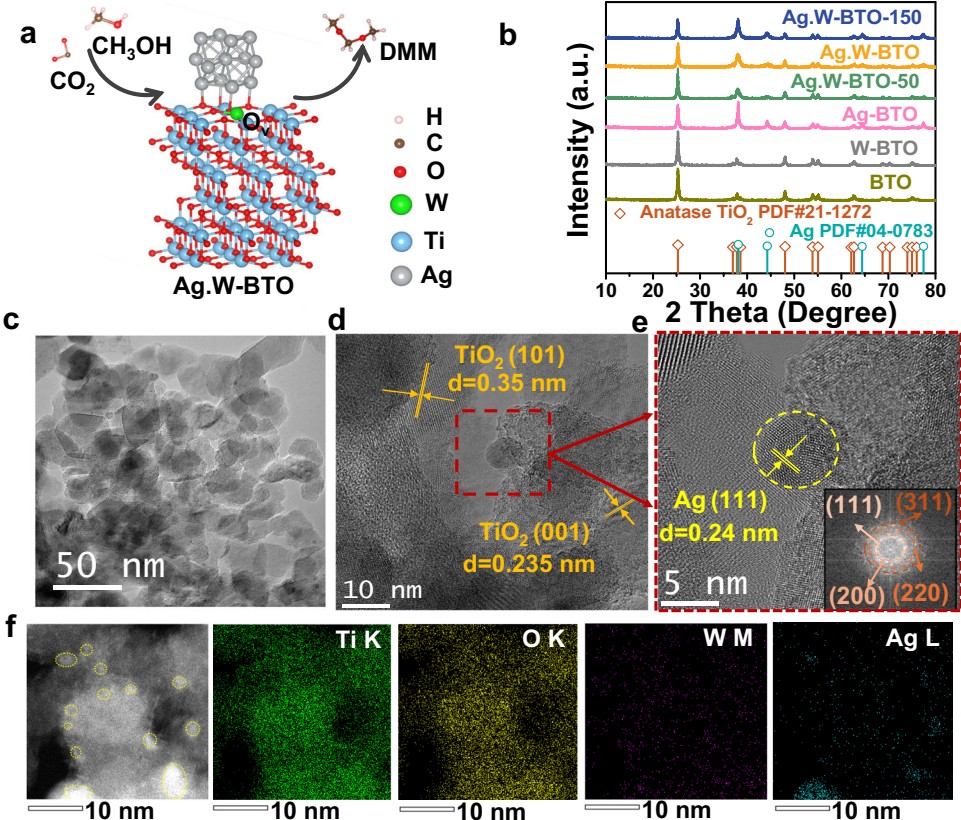

**Fig. 2 | Morphology and structural characterizations of Ag.W-BTO. a** Proposed structure of Ag.W-BTO and reaction. **b** XRD of all catalysts. **c–e** TEM and HRTEM images of Ag.W-BTO. **f** HAADF-STEM image of Ag.W-BTO and EDS elemental mappings of Ti, O, W, and Ag. (a. u.) represents arbitrary units.

(DRIFTS), isotope labeling experiments and density functional theory (DFT) calculations. The analyses showed that CH₃OH meets photo-excited holes to produce *CH₃O and H⁺ on W sites. At the same time, $CO_2$ was reduced by a multi-step proton-coupled electron transfer (PCET) process to make *CH₂O and these two intermediates finally produced DMM by asymmetric C−O coupling. This work provides a route for efficiently preparing DMM with high selectivity in mild-range photocatalytic redox conditions.

## Results

### Catalyst synthesis and structural characterization

The designed model catalysts were prepared by simply reducing WCl₆ and AgNO₃ with sodium borohydride (NaBH₄) on the BTO substrate. The tungsten (W) ions were successfully introduced to the lattice of BTO, and the resulting Ag nanoparticles were in situ formed on the BTO surface during the reduction process (Fig. 2a). More details of the preparation process are supplied in the experimental section. Comparative catalysts, including BTO, Ag-BTO and W-BTO, were synthesized using the same method (Supplementary Figs. 1–11 and Supplementary Table 1). The crystal structures of all samples are shown in the X-ray diffraction (XRD) patterns of Fig. 2b. As expected, the BTO substrates in the composite samples were identified as pure anatase TiO₂ (PDF#21-1272) after Li-EDA treatment. In this case, we expected high UV-visible light absorption and increased stability on the newly obtained BTO during the reaction[24]. The Ag nanoparticles were primarily in their metallic phase (PDF#04-0783). No prominent peak corresponding to W was evident, indicating that the W atoms were successfully doped into the BTO. The blocky Ag.W-BTO was ~40−50 nm wide, as shown in the transmission electron microscopy (TEM) image of Fig. 2c. In addition, the interplanar d spacing was 0.35 nm and 0.235 nm, shown in the Ag.W-BTO high-resolution TEM (HRTEM) image and belonging to the (101) and (001) anatase phase

TiO₂, respectively (Fig. 2d)[24]. The Ag nanoparticles were ~2–10 nm in diameter, with a clear lattice and a distance spacing of 0.24 nm, which can match Ag (111) (Fig. 2e). A fast Fourier-transform (FTT) image (inset) shows the (111), (200), (220), and (311) facets of Ag, which coincide with XRD patterns[25,26]. High-angle annular dark-field scanning TEM (HAADF-STEM) revealed Ag nanoparticles 2–10 nm in diameter (yellow circle, Fig. 2f), and energy dispersive spectrometry (EDS) was used to map the elemental distribution of Ti, O, W, and Ag. Due to the large amount of AgNO₃ added during the preparation progress (checked by ICP-OES 38.53 wt.%; Supplementary Table 2), the Ag nanoparticles showed some aggregation of Ag.W-BTO-150 (Supplementary Fig. 5).

The chemical states of the catalysts were determined by X-ray photoelectron spectroscopy (XPS). In the Ti 2p spectra (Fig. 3a), four characteristic peaks appear near 464.30, 463.31, 458.64, and 457.82 eV, corresponding to Ti 2p₁/₂ Ti⁴⁺, Ti 2p₁/₂ Ti³⁺, Ti 2p₃/₂ Ti⁴⁺, and Ti 2p₃/₂ Ti³⁺, respectively[27,28]. The presence of peaks corresponding to Ti³⁺ confirms that the synthesized catalyst contains oxygen vacancies (Oᵥ). Compared with BTO and Ag-BTO, all W-BTO and Ag.W-BTO peaks displayed a positive shift. This phenomenon is primarily caused by W species with a lower electron cloud density and a strong electron affinity that can absorb electrons from BTO, resulting in the formation of a stable structure[29]. The Ag peaks centered at 372.60 eV and 366.70 eV can be ascribed to metallic Ag (Fig. 3b). The higher shift of the Ag binding energy in Ag.W-BTO is attributed to strong heterogeneous interaction and electron transfer between Ag and W-BTO substrate[30]. In the W 4f spectrum (Fig. 3c), W⁶⁺ 4f₅/₂, W⁵⁺ 4f₅/₂, W⁶⁺ 4f₇/₂, and W⁵⁺ 4f₇/₂ can be observed at 38.38, 36.72, 35.02, and 33.24 eV, respectively[31]. To prevent photo-generated carriers from recombining under light irradiation, W⁶⁺, as a donor just below the conduction band, can be gradually transformed to unsaturated W⁵⁺, which works in W⁶⁺/W⁵⁺ pairs to improve performance (Supplementary Fig. 29c)[32]. The W of Ag.W-BTO

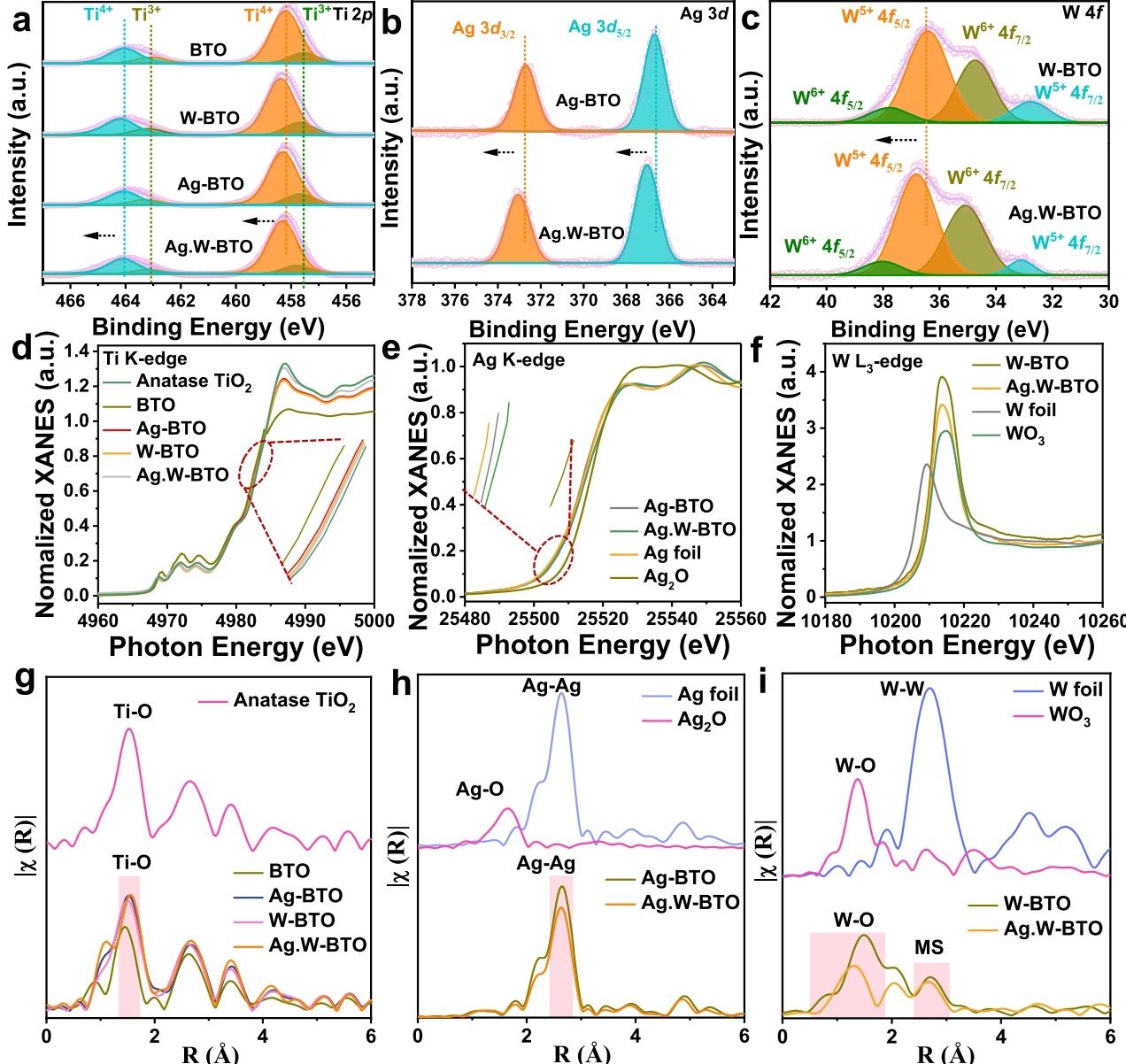

**Fig. 3 | XPS and X-ray absorption fine structure (XAFS) characterizations. a−c** Ti 2*p*, Ag 3*d*, and W 4*f* XPS of BTO, W-BTO, Ag-BTO, and Ag.W-BTO, respectively. (a. u.) represents arbitrary units. **d−f** Normalized Ti K-edge, Ag K-edge, W L$_3$-edge, and their difference X-ray absorption near edge structure (XANES) of catalysts with standard references. **g−i** Extended XAFS (EXAFS) k³ χ(k) Fourier-transform (|X (R)|) spectra of Ti, Ag, and W in *R*-space of catalysts with standard references, respectively. R (Å) represents radial distance in Angstroms.

also shifted to a positive binding energy, indicating more electron transmission from W to O after Ag modification. This is attributed to the strong electron acceptor properties of O[33,34]. All the highest peaks of the O 1*s* spectra can be ascribed to Ti–O. The other peaks correspond to $O_v$, an −OH/O−W group, and $H_2O$ (absorbed on the surface), respectively (Supplementary Fig. 12a)[35]. No boron species remained in samples during the synthesis process (Supplementary Fig. 12b). The presence of $Ti^{3+}$ species and $O_v$ were also confirmed by the Electron paramagnetic resonance (EPR) spectra (Supplementary Fig. 13). Both BTO and Ag.W-BTO show distinctive EPR signals of paramagnetic $Ti^{3+}$ (*g* = 1.96) and $O_v$ (*g* = 2.003) while $TiO_2$ (P₂₅) showed a negligible EPR signal[36–38]. This result implies that BTO with $O_v$ was successfully synthesized after reduction with Li-EDA. The peak intensities of $Ti^{3+}$ and $O_v$ in BTO are higher than those of Ag.W-BTO, meaning the doping of W may slightly replace the $Ti^{3+}$ or cover the $O_v$ in Ag.W-BTO sample.

The electronic states of Ti, Ag, and W were explored by X-ray absorption near-edge spectroscopy (XANES). The state of Ti for all prepared samples shows features similar to those of anatase $TiO_2$ (Fig. 3d). Compared with BTO, Ag-BTO, and W-BTO, the higher Ti K-edge absorption peak intensity in Ag.W-BTO is due to the collective electronic modulations of extrinsic Ag and W species. Meanwhile, the Ag in Ag-BTO and Ag.W-BTO can be traced to metallic Ag, as shown in the XRD and XPS results (Fig. 3e). However, the Ag pre-edge adsorption position of Ag.W-BTO had a positive shift compared with that of Ag-BTO, which is caused by high interaction among Ag and W-BTO support. Strong interface interactions are anticipated to promote beneficial electron transfer in the photocatalytic process. From the W L$_3$-edge absorption spectra (Fig. 3f), the white line peak positions of W in W-BTO and Ag.W-BTO are higher than those of W foil and $WO_3$ attributed to the strong electron adsorption properties of O in W-BTO and Ag.W-BTO, which agrees with the discussions of XPS spectra. The

distinctive coordination environments of Ti, Ag, and W were investigated by Fourier-transformed (FT) extended X-ray absorption fine structure (EXAFS) spectra and fitting data (Supplementary Fig. 14 and Supplementary Tables 3–5). In the Ti FT-EXAFS spectra, Ti–O bonds can be seen at -1.63, 1.65, and 1.62 Å for Ag-BTO, W-BTO, and Ag.W-BTO, respectively (Fig. 3g and Supplementary Fig. 14a). The Ti–O bonds of these samples are extended compared to those of pure BTO (located at -1.52 Å) due to the introduction of W and Ag species that can stretch the Ti–O bond length and synergistically tune the local coordination environment of Ti. The dominant peaks of Ag–Ag shown in Ag-BTO and Ag.W-BTO are nearly no difference which is consistent with that of standard Ag foil (Fig. 3h and Supplementary Fig. 14b). For $k^3 \chi(k)$-FT of the W $L_3$-edge EXAFS spectra (Fig. 3i and Supplementary Fig. 14c), the first coordination shell (W–O peak at 0.7–1.9 Å) was composed primarily of single-scattering O atoms, accompanied by multiple-scattering contributions from the second shell (peaks at 2–3 Å)[39,40]. The interatomic distances of W–O in W-BTO (-1.47 Å) and Ag.W-BTO (-1.29 Å) are different from those of standard $WO_3$ (1.36 Å) as W atoms are doped into the BTO lattice rather than the isolated oxide species. Overall, the electronic interaction of Ag and W dual-active sites is expected to be capable of guaranteeing the photo-redox synergic reaction for higher efficiency.

The adsorption abilities of light in all catalysts were measured by UV-visible light spectra (Supplementary Fig. 15a)[41]. The absorption edge of Ag.W-BTO exhibited a slight redshift after W implantation and Ag deposition, which can enable stronger UV-visible light absorption and improve hot-carrier generation and catalytic performance. The band gaps of the photocatalysts were calculated using Kubelka−Munk and Tauc equations (more details are provided in the Supporting Information)[42]. The BTO and W-BTO values were 2.91 and 2.86 eV[43], respectively (Supplementary Fig. 15b). All the BTO-related samples showed a narrower band structure than pure $TiO_2$ ($P_{25}$, 3.2 eV), which was attributed to the existence of $Ti^{3+}$ defect sites with multiple internal energy band gaps after Li-EDA treatment[22,44]. The valence band (VB) edge potentials and Femi levels of these catalysts were determined using Ultraviolet photoelectron spectroscopy (UPS) spectra[43]. In details, the positions of the secondary electron cutoff ($E_{cut\ off}$) and the valence band maximum ($E_{VBM}$) positions are determined using linear extrapolation of UPS[45]. As shown in Supplementary Fig. 15c, the $E_{cut\ off}$ values of BTO and W-BTO are 17.21 and 17.49 eV, respectively, and the $E_{VBM}$ values of these two samples are 1.79 and 2.15 eV, respectively. The work function ($\phi$) can be calculated by subtracting $E_{cut\ off}$ from the energy of the incident UV light ($h\gamma$) after measuring the width of the emitted electrons from the onset of the secondary electrons up to the Fermi edge, according to the Formula (1).

$$\phi = h\gamma - E_{cut\ off} \qquad (1)$$

Here, the energy of Helium line as a $h\gamma$ is 21.22 eV. According to formula (1), the $\phi$ of BTO and W-BTO are calculated to be 4.01 and 3.73 eV, respectively. Consequently, the Femi levels of BTO and W-BTO are −4.01 and −3.73 eV, respectively. The VB of BTO and W-BTO in Vacuum ($E_{VAB}$) were calculated using the Formula (2):

$$E_{VAB} = -(E_{VBM} + \phi) \qquad (2)$$

Resulting in −5.80 and −5.88 eV, respectively. All values corresponding to vacuum should be replaced with normal hydrogen electrode (NHE), resulting in a difference of −4.44 eV[46]. Therefore, band gap structures of BTO and Ag.W-BTO are shown in Supplementary Fig. 15d. The VB of W-BTO (1.44 V) exhibited higher positivity than BTO (1.36 V), imparting that doping W species theoretically facilitates MOR performance. Due that the Femi level ($E_f$) of metallic Ag nanoparticles ($E_f = -4.26$ eV) is more negative than that of BTO ($E_f = -4.01$ eV) and W-BTO ($E_f = -3.73$ eV) (Supplementary Fig. 16 a, b). Take Ag.W-BTO as

an example, based on the strong interfacial interaction by Mott-Schottky junction, the electrons flow from W-BTO to Ag induced by the difference in $E_f$ between Ag.W-BTO until the system reaches equilibrium, resulting in band bending and Schottky barrier formation at the interface. The suitable Schottky barrier facilitates the migration of photogenerated electrons, which also proves that CO2RR are more likely to occur on Ag species of Ag.W-BTO during DMM synthesis process.

$N_2$ adsorption/desorption analyses were carried out to determine the effect of doping W and depositing Ag on the pore structures (Supplementary Fig. 17)[47]. The presence of mesoporous structures is implied by type IV isotherms with $H_3$ hysteresis loops in all prepared catalysts[48]. The pore-structure distribution curves in Supplementary Fig. 17f are evidence of the dominant mesoporous and microporous structures of all samples. The specific surface areas (SSAs) of BTO, W-BTO, Ag-BTO, and Ag.W-BTO were 28.62, 44.68, 35.26, and 52.12 $m^2\ g^{-1}$, respectively (additional parameters are shown in Supplementary Table 6). The increased SSAs of Ag.W-BTO is likely attributed to the porous structures (Supplementary Fig. 10b).

## Photocatalytic performance by coupling CO2RR and MOR

A homemade photocatalytic reaction system (Supplementary Fig. 18) was used to verify the photocatalytic performance of the samples used as photocatalysts for photo-redox $CO_2RR$ and MOR systems. Specifically, 30 mL of $CH_3OH$ was added to the reactor, which was purged with $CO_2$ gas for 1 h without any use of sacrificial reagents in the dark. After irradiation with UV-visible light (320−780 nm) for 9 h, gas production ($H_2$, $O_2$, CO, $CH_4$) and liquid production ($CH_2O$ and DMM) were quantitively detected by gas chromatography (GC). The corresponding productivity and selectivity of each product were calculated using Eqs. (3) – (9) as detailed in the Method section. The CO production rate on BTO reached 217.71 $\mu mol\ g^{-1}$, which is -100 times greater than that on Ag.W-BTO (Fig. 4a, b). In contrast, Ag.W-BTO produced the largest amount of DMM (5702.49 $\mu mol\ g^{-1}$), but scant DMM products were seen on BTO (Fig. 4c). In addition, $CH_4$, as by-product, were produced with a minimal amount of Ag.W-BTO, confirming that it is possible to make extensive use of reaction intermediates at an Ag and W dual-redox site in an Ag.W-BTO sample. The amount of $CO_2RR$ products (CO, $CH_4$, and $CH_2O$) on Ag-BTO was significantly greater than on W-BTO, which indicates that Ag prefers to be the active reduction site. Overall, W-BTO, Ag-BTO, Ag.W-BTO-50, and Ag.W-BTO-150 achieved DMM yields of 97.69, 488.47, 3337.96, and 551.11 $\mu mol\ g^{-1}$, respectively (Fig. 4c and Supplementary Fig. 19–24). Compared with Ag-BTO and W-BTO, the greater photo-efficiency of the larger DMM yield on Ag.W-BTO was achieved because the Ag and W species individually acted as active sites to promote reductive and oxidative reactions, respectively. Ag agglomerated into large nanoparticles that covered many oxidation sites on the Ag.W-BTO-150 when excess Ag was added, resulting in less DMM and inferior selectivity (Supplementary Fig. 5). Due to producing the large amount of DMM on the Ag.W-BTO sample, the production of DMM changes was detected as time increased. As shown in Supplementary Fig. 25, the yield of DMM also exhibited a growth trend as time increased. DMM selectivity on Ag.W-BTO reached 92.08%, which is greater than that of other samples (Fig. 4d). The Apparent quantum yield (AQY) for BTO and Ag.W-BTO were measured using a range of monochromatic light band-pass filters. As depicted in Supplementary Fig. 26, the AQY of this two samples trend aligns with the UV-visible absorption spectrum, especially the AQY values of Ag.W-BTO reaching 2.15% and 1.01% at 395 nm and 420 nm, respectively. Furthermore, DMM yield and selectivity did not decrease after four catalysis cycles on Ag.W-BTO, indicating that Ag.W-BTO has excellent stability (Supplementary Fig. 27). The TEM, SEM images, XRD patterns and XPS spectrums after the cycling test were also recorded (Supplementary Fig. 28 and 29). Ag.W-BTO presents excellent sustainability of morphology and crystalline nature,

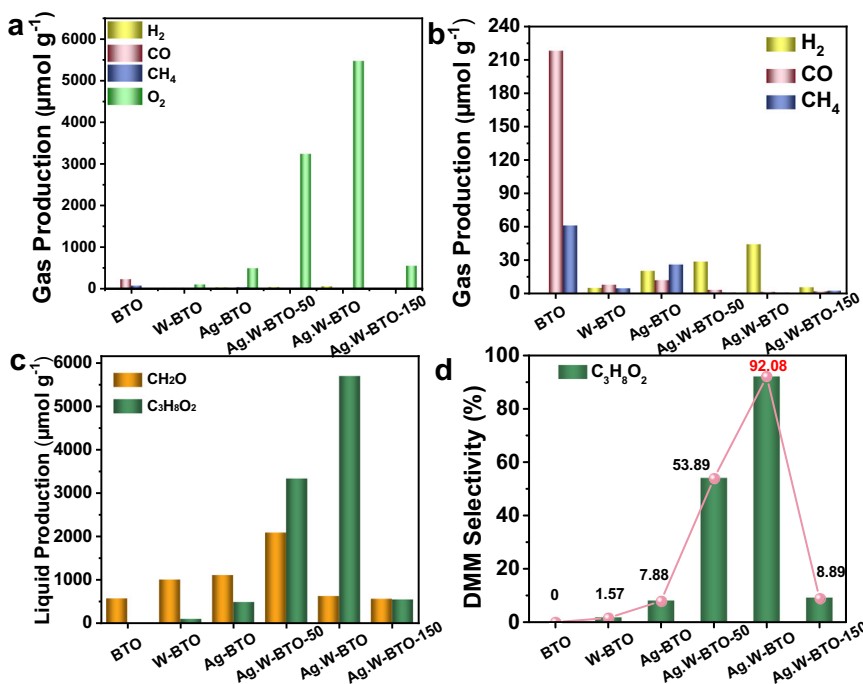

**Fig. 4 | Photocatalysis CO2RR coupling with MOR performance. a** Gas production. **b** Enlarged $H_2$, CO and $CH_4$ production. **c** Liquid production of prepared samples and **d** selectivity of DMM after 9 h of reaction time.

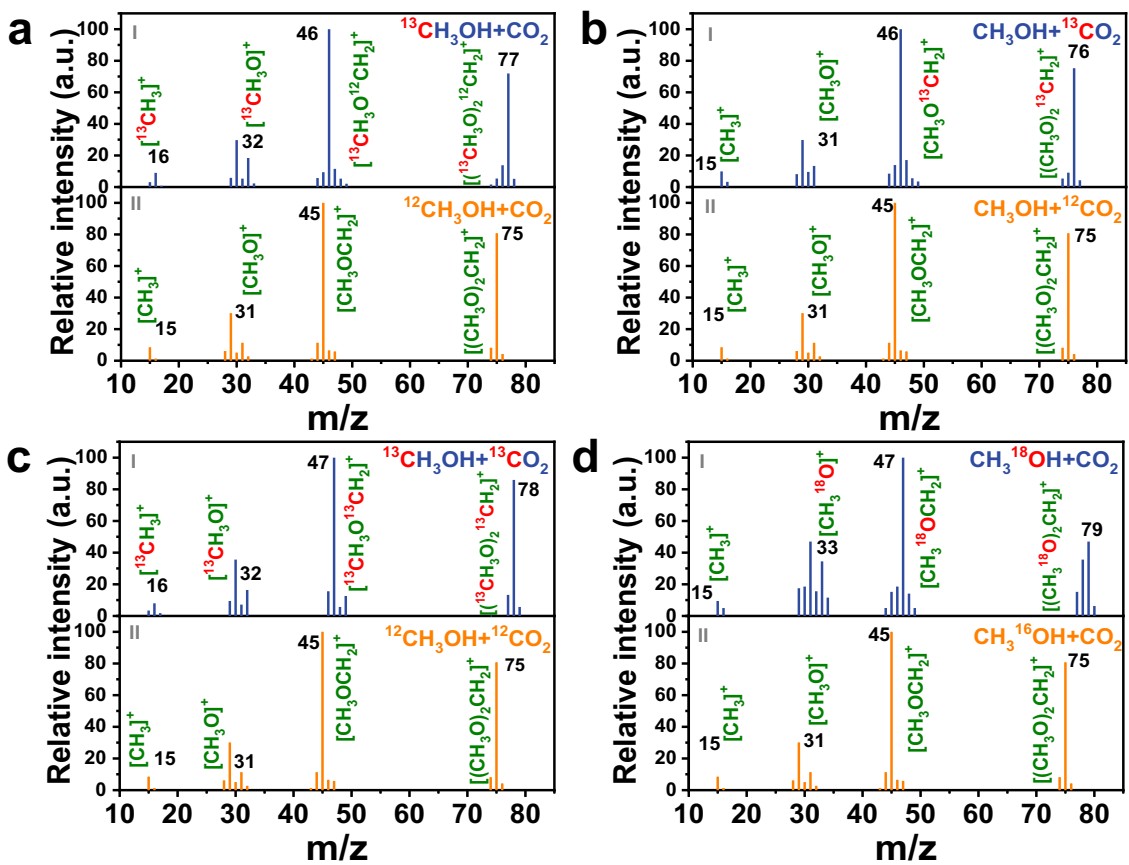

**Fig. 5 | The results of isotope labeling experiments. a** GC-MS results of I) isotope labeled $^{13}CH_3OH + CO_2$ and II) isotope non-labeled $^{12}CH_3OH + CO_2$. **b** GC-MS results of I) isotope labeled $CH_3OH + ^{13}CO_2$ and II) isotope non-labeled $CH_3OH + ^{12}CO_2$. **c** GC-MS results of I) isotope labeled $^{13}CH_3OH + ^{13}CO_2$ and II) isotope non-labeled $^{12}CH_3OH + ^{12}CO_2$. **d** GC-MS results of I) isotope labeled $CH_3^{18}OH + CO_2$ and II) isotope non-labeled $CH_3^{16}OH + CO_2$. (a. u.) represents arbitrary units.

which results in the higher stability of activity and selectivity for DMM synthesis. The valence states of Ti, Ag, W and O also showed nearly no difference before and after stability. The high stability is due to the fact that W is doped into the lattice of BTO with lattice distortion[49].

Furthermore, a series of controlled experiments were performed to examine whether our photocatalytic DMM generation integrated the advantages of $CO_2$RR and MOR (Supplementary Fig. 30). Ag.W-BTO at standard conditions produced DMM at a rate of 5702.49 $\mu$mol g$^{-1}$, which is ~7 times greater than without $CO_2$ (866.59 $\mu$mol g$^{-1}$) and ~43 times greater than without $CH_3OH$ (132.03 $\mu$mol g$^{-1}$). To explore the influence of pH on DMM synthesis, the pH of $CH_3OH$ before and after $CO_2$ exposure were tested to be 6.256 and 5.202, respectively (Supplementary Fig. 31), which indicates that the introduction of $CO_2$ can indeed decrease the reactant pH, which is essential for the initiation of the coupled reaction. Besides, the pH of the $CH_3OH$ reactant was adjusted to ~5 by using HCl ($CH_3OH$ + Ar, pH = 5) to avoid the $CO_2$ effect. It showed that the synthesis amount of DMM in $CH_3OH$ + Ar with pH = 5 condition is 533.20 $\mu$mol g$^{-1}$ after a 9 h reaction, while that of DMM in $CH_3OH$ + Ar and $CH_3OH$ + $CO_2$ without adjusting pH is about 866.59 and 5702.49 $\mu$mol g$^{-1}$, which indicates that lower pH is not the main factor affecting the production of DMM in $CH_3OH$ (Supplementary Fig. 32). Moreover, to more precisely investigate the impact of *CO intermediates on the reaction, we checked DMM production by substituting $CO_2$ with CO on Ag.W-BTO under the same reaction conditions. The marked increase in DMM production (7008.35 $\mu$mol g$^{-1}$) observed when using CO and $CH_3OH$ as reactants, compared to the utilization of $CO_2$ and $CH_3OH$ (5702.49 $\mu$mol g$^{-1}$), strongly suggests the active participation of *CO intermediates in the synthesis process of DMM (Supplementary Fig. 33). Additionally, commercial $TiO_2$ (P$_{25}$) was substituted for BTO in Ag.W-BTO using the same $NaBH_4$ reduction method to assess their catalytic activity. $TiO_2$ was chosen for its minimal oxygen vacancies ($O_v$), allowing for comparison with Ag.W-BTO to assess whether the $O_v$ are active sites in the reaction. DMM yields on $TiO_2$ and Ag.W-$TiO_2$ were found to be ~0 and 1688.35 $\mu$mol g$^{-1}$, respectively (Supplementary Fig. 34). The presence of DMM on Ag.W-$TiO_2$ (without $O_v$) suggests that $O_v$ may not be the primary reactive sites. However, the lower DMM yield on Ag.W-$TiO_2$ compared to Ag.W-BTO can be attributed to the weaker absorption of visible light, low specific surface area, and high electron and hole recombination rate of the $TiO_2$ substrate[22,36]. Humid $CO_2$ flowed instead of $CO_2$ as a controlled experiment to explore the moisture effect. The produced amount of DMM is about 2307.10 $\mu$mol g$^{-1}$. The reason why the amount decreases compared with normal conditions ($CO_2$ + $CH_3OH$) may be due to $H_2O$ oxidation in the reaction system, which competes with MOR. The high overpotential of $H_2O$ oxidation to $O_2$ (1.23 V vs. RHE) than MOR (0.58 V vs. RHE) may result in slower reaction kinetics (Supplementary Fig. 35)[17]. In addition, when the frequency of the incident photons matches the oscillation frequency of the surface free electrons in Ag nanoparticles, the localized surface plasmon resonance (LSPR) effect occurs[50]. As for exploring the LSPR effect of deposition of Ag on the Ag.W-BTO sample, the surface temperatures of the Ag.W-BTO were assessed using an infrared thermal imager. Analysis of the temperature versus time curves for the Ag.W-BTO revealed a gradual increase from about 21–31 °C in 10 min and stabilization of about 31 °C within 30 min (Supplementary Fig. 36). This stabilization phenomenon is attributed to the equilibrium reached between the heat dissipation of the sample and its surrounding medium[46].

To further explore the reaction mechanism and identify the carbon and oxygen sources origin of the produced DMM, the result of $CO_2$ and $CH_3OH$ isotope-labeling experiments were determined by Gas chromatography-mass spectrometry (GC-MS). In the non-labeled DMM, the highest intensity abundance signal (base ion peak) is m/z = 45 ([$CH_3OCH_2$]$^+$). The ion fragment of m/z = 75 ([$CH_2OCH_2OCH_3$]$^+$) is more stable than molecular ion fragment (m/z = 76, [$CH_3OCH_2OCH_3$]$^+$) after ionized and fragmented by a mass

spectrometer. The relative abundance intensity of m/z = 75 (the second highest intensity) is much higher than m/z = 76. Therefore, the second highest intensity ion fragment of m/z = 75 can determine the DMM[51]. As depicted in Fig. 5a, the second highest intensity ion peak of DMM derived from non-labeled $CH_3OH$ + $CO_2$ was m/z = 75 (Fig. 5a-II), while that of the DMM derived from labeled $^{13}CH_3OH$ + $CO_2$ reached m/z = 77 (Fig. 5a-I), evidently proving that the two carbon sources in DMM originate from $CH_3OH$ and the one carbon originates from $CO_2$. It is also observed that the m/z values of other fragments derived from $^{13}CH_3OH$ + $CO_2$ (Fig. 5a-I) such as [$^{13}CH_3$]$^+$ (m/z = 16) and [$^{13}CH_3O$]$^+$ (m/z = 32), and m/z = 46 [$^{13}CH_3O^{12}CH_2$]$^+$ are elevated by one unit higher than those of DMM derived from non-labeled $CH_3OH$ and $CO_2$ (m/z = 15 [$^{12}CH_3$]$^+$, m/z = 31 [$^{12}CH_3O$]$^+$, and m/z = 45 ([$^{12}CH_3O^{12}CH_2$]$^+$) (Fig. 5a-II), suggesting that the two terminal carbons of DMM ([$^{13}CH_3O$]$_2$$^{12}CH_2$) are derived from $^{13}CH_3OH$ and the central carbon comes from $^{12}CO_2$. In Fig. 5b, the ion fragment peak of [$CH_3$]$^+$ (m/z = 15) and [$CH_3O$]$^+$ (m/z = 31) are unchanged comparing $CH_3OH$ + $^{13}CO_2$-derived DMM (Fig. 5b-I) and $CH_3OH$ + $^{12}CO_2$-derived DMM (Fig. 5b-II). But, the $^{13}CO_2$-derived DMM shows a base ion peak of [$CH_3O^{13}CH_2$]$^+$ (m/z = 46) and the secondary intensity ion fragment peak [$CH_2O^{13}CH_2OCH_3$]$^+$ (m/z = 76) (Fig. 5b-I), which is only one m/z higher than that of the $^{12}CO_2$ labeled case (m/z = 45 [$CH_3OCH_2$]$^+$ and m/z = 75 [$CH_2OCH_2OCH_3$]$^+$) (Fig. 5b-II), which means middle position carbon in the DMM molecule originates from a $CO_2$ source. In $^{13}CH_3OH$ + $^{13}CO_2$ labeled experiments, it is clearly showed the base ion peak is m/z = 47 ([$^{13}CH_3O^{13}CH_2$]$^+$) and the second intensity fragment is m/z = 78 ([$^{13}CH_2O^{13}CH_2O^{13}CH_3$]$^+$) (Fig. 5c-I), respectively, which is two and three units higher than those of non-labeled m/z = 45 ([$CH_3OCH_2$]$^+$) and m/z = 75 ([$CH_2OCH_2OCH_3$]$^+$) (Fig. 5c-II), respectively. In addition, comparing the result of isotope labeled $^{13}CH_3OH$ + $^{12}CO_2$ experiments (Fig. 5a-I), the fragments of m/z = 47 ([$^{13}CH_3O^{13}CH_2$]$^+$) and m/z = 78 ([$^{13}CH_2O^{13}CH_2O^{13}CH_3$]$^+$) showed one unit higher than m/z = 46 ([$^{13}CH_3O^{12}CH_2$]$^+$) and m/z = 77 ([$^{13}CH_2O^{12}CH_2O^{13}CH_3$]$^+$), respectively. This result also confirmed that the middle carbon of DMM comes from $CO_2$. Furthermore, comparing the result of isotope labeled $^{12}CH_3OH$ + $^{13}CO_2$ experiments (Fig. 5b-I), the fragments of m/z = 47 ([$^{13}CH_3O^{13}CH_2$]$^+$) and m/z = 78 ([$^{13}CH_2O^{13}CH_2O^{13}CH_3$]$^+$) showed one and two unit higher than those of m/z = 46 ([$^{12}CH_3O^{13}CH_2$]$^+$) and m/z = 76 ([$^{12}CH_2O^{13}CH_2O^{12}CH_3$]$^+$), respectively, which exhibited that the two terminal carbons of DMM comes from $CH_3OH$. In the only $^{13}CH_3OH$ labeled isotope experiment (Supplementary Fig. 37), the m/z = 78 peak corresponds to the ion of the [$^{13}CH_2O^{13}CH_2O^{13}CH_3$]$^+$ structure (Supplementary Fig. 37-I), exhibiting a 3-mass unit increase compared to the non-labeled [$^{12}CH_2O^{12}CH_2O^{12}CH_3$]$^+$ ion (Supplementary Fig. 37-II). Comparing the GC-MS results of the $^{13}CH_3OH$ + $CO_2$ (Fig. 5a-I) and pure $^{13}CH_3OH$ isotope (Supplementary Fig. 37-I) experiments further confirmed not only two $CH_3OH$ oxidations but also $CO_2$RR involvement in our DMM formation pathway. The $^{18}O$-labelled $CH_3^{18}OH$ with $CO_2$ isotope experiment ($CH_3^{18}OH$ + $CO_2$) was conducted to trace the O atoms in the generated DMM (Fig.5d). The base ion peak and second intensity peak of $^{18}O$-labled DMM are m/z = 47 ([$CH_3^{18}OCH_2$]$^+$) and m/z = 79 ([$CH_2^{18}OCH_2^{18}OCH_3$]$^+$) (Fig. 5d-I), respectively, which exhibited two and four units higher than those of non-labeled DMM (m/z = 45 [$CH_3OCH_2$]$^+$ and m/z = 75 [$CH_2OCH_2OCH_3$]$^+$) (Fig. 5d-II), verifying the all the O atoms in DMM are produced by the reaction of $CH_3OH$.

## Photoinduced electrons and holes separation and transportation

The separation and transportation of electrons and holes were determined by photoluminescence (PL) measurements of the BTO, W-BTO, Ag-BTO, and Ag.W-BTO catalysts (Supplementary Fig. 38)[52,53]. The PL measurements, which assess internal charge transfer in solid samples, are excited under the monochromatic wavelength (350 nm) that may not entirely mimic the actual reaction conditions. All measured samples displayed the same spectral peaks near 498 nm, while the peak

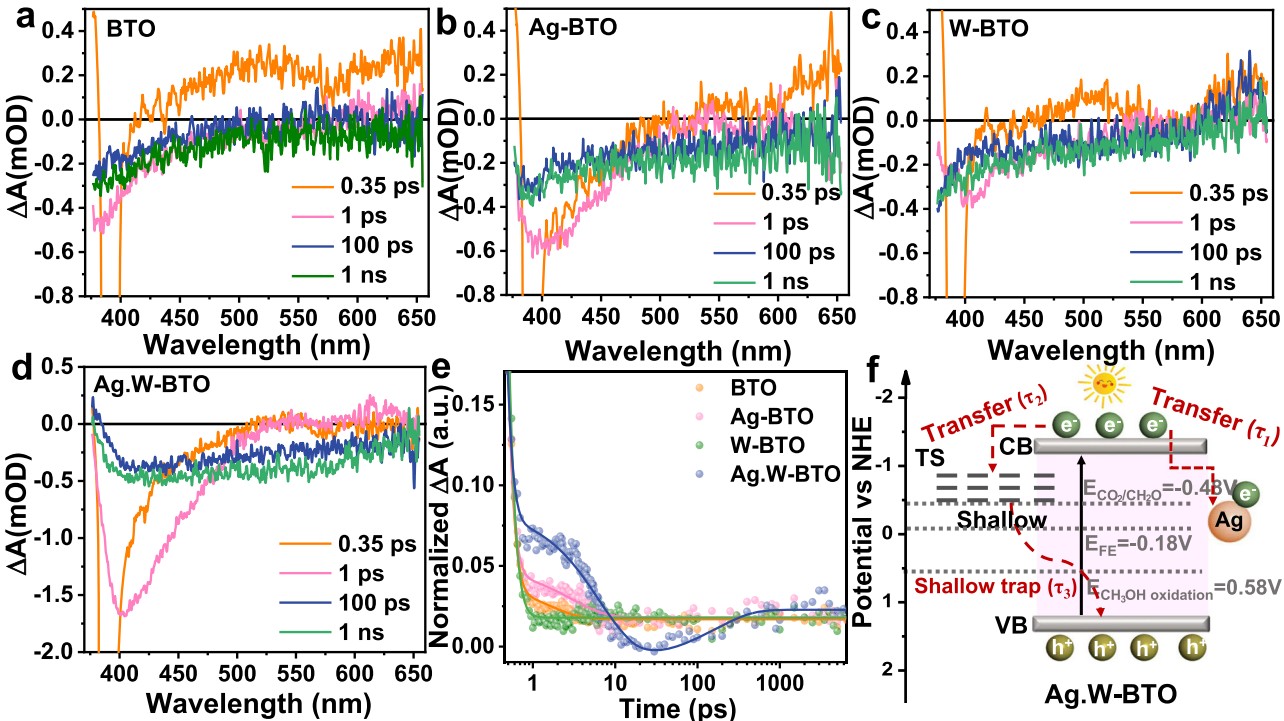

**Fig. 6 | Fs-TA spectra. a** BTO, (**b**) Ag-BTO, (**c**) W-BTO, and (**d**) Ag.W-BTO at 350 nm. **e** The normalized TA kinetics of BTO, Ag-BTO, W-BTO, and Ag.W-BTO, respectively. **f** A scheme for the charge-transfer process of Ag.W-BTO. ΔA(mOD) represents the change absorbance in milli-optical density.

intensity of Ag.W-BTO significantly declined, indicating that the Ag.W-BTO catalyst had a lower charge-carrier recombination rate. This is because Ag species tend to readily provide electrons for a $CO_2RR$, while the W species prefer to provide holes to promote MOR. Ag.W-BTO displayed the strongest photocurrent response under light irradiation (Supplementary Fig. 39a). In addition, Ag.W-BTO showed the smallest arc radius of electrochemical impedance spectra (Supplementary Fig. 39b), indicating that Ag.W-BTO had the lowest charge-transfer resistance and highest photo-charge separation rate, which favor accelerated reaction kinetics and enhanced efficiency[54].

Additionally, fs-TA measurements were conducted under 350 nm laser-flash photolysis to further elucidate electron transfer behavior at room temperature, with the checking conditions exhibiting disparities compared to actual reaction conditions such as temperature, pressure, and reactant composition. As depicted in Fig. 6a–d, the fs-TA spectra of BTO, Ag-BTO, W-BTO, and Ag.W-BTO included both broad negative and positive signals. The negative absorption signals can be ascribed to stimulated emission and bleaching of the ground state, and the positive absorption signals correspond to the presence of excited-state absorption by electrons[55]. The negative signals of Ag-BTO and W-BTO were stronger than those of BTO due to the instantaneous generation of charge carriers from the ground state and their direct excitation to the emission state by fast-electron transfer (Fig. 6a–c)[56,57]. As shown in Fig. 6d, the negative peak center (near 410 nm) of Ag.W-BTO was nearly three times higher than that of BTO, which can be attributed to the promotion of the utilization of electrons and holes to prevent charge carriers from recombination by Ag and W species, respectively. The dynamical decay traces and fitting of the photo-excited charge carriers for BTO, Ag-BTO, W-BTO, and Ag.W-BTO were probed at 388 nm (Fig. 6e and Supplementary Table 7), and a tri-exponential decay function was applied to fit the kinetics traces. The BTO, Ag-BTO, and W-BTO only show delay times of $\tau_1$ and $\tau_2$, which can be ascribed to electron transfer to the shallow trapping state (TS) and deep TS, respectively. However, Ag.W-BTO has three different trapping signals ($\tau_1 = 0.10$ ps, $\tau_2 = 6.71$ ps, and $\tau_3 = 172.05$ ps), in which $\tau_1$ and $\tau_2$ are

assigned to electron transfer to a shallow Ag species and shallow TS, respectively[58], and $\tau_3$ is due to the recombination of the shallow-trapped electrons with holes[59]. The final charge transfer process is shown in Fig. 6f. The electron transitions to an excited state on the conductive band (CB), leaving holes at the VB under irradiation. Subsequently, partial electrons quickly transfer to Ag sites ($\tau_1$) and shallow TS ($\tau_2$); the remaining electrons are combined with unconsumed holes ($\tau_3$). The longer delay time of $\tau_3$ indicates a lower speed for the recombination of charge electrons and holes during the photo-redox process.

## Study of reaction intermediates and mechanism

In-situ DRIFTS was used to identify the reaction intermediates and mechanisms in DMM synthesis. KBr served as a background (Supplementary Fig. 40) was removed before all measurements. $CO_2$ and $CH_3OH$ were purged into the system for 1 h in the dark. All peaks were marked with dotted lines, with black cases representing negligible changes in absorption bands and colorful lines emphasizing gradual increments of the bands. Due to the continuous filling of $CO_2$ gas with $CH_3OH$, a high stretching vibration bond at $2300\,cm^{-1}$ belonging to $CO_2$ adsorption is evident (Supplementary Fig. 41)[59]. The band located in $1155\,cm^{-1}$ corresponds to the spreading $-CH_3O$ group of $CH_3OH$, which confirms that $CH_3OH$ is successfully adsorbed onto the Ag.W-BTO surface (Fig. 7b and c)[60]. The signals centered at 1377, 1455, 2860, and $3030\,cm^{-1}$ were caused by the deformation of methyl groups (Fig. 7a and b). The absorption bands at 1492 and $2930\,cm^{-1}$ can be attributed to the stretching signals of C−H bonds[60,61]. The bands at 3640 and $3727\,cm^{-1}$ correspond to the isolated $-OH$ group (provided from $CH_3OH$) and the adsorbed $-OH$ group on the surface of $TiO_2$, respectively[61]. All these bands confirm the successful absorption of $CO_2$ and $CH_3OH$. After UV-visible light irradiation, new intermediates emerged. Three characteristic bands appeared at 1260, 1716, and $2060\,cm^{-1}$, corresponding to $*CH_2O$, $*COOH$, and $*CO$ intermediates, respectively[60,62]. Due to the small amount of sample added, the formation of the $*CH_3O$ intermediate for $CH_3OH$ as a

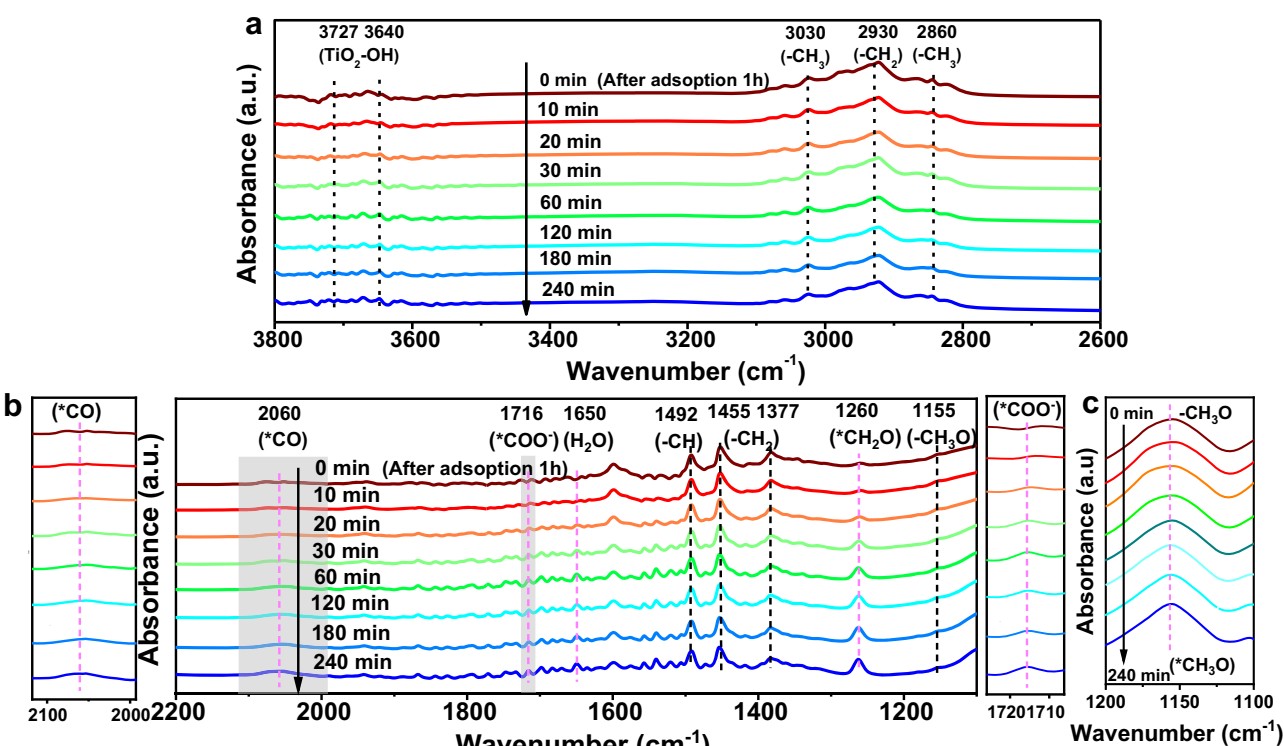

**Fig. 7 | In-situ DRIFTS spectroscopy characterization of CO2 + CH3OH on Ag.W-BTO. a, b** In-situ DRIFTS spectra at detailed reaction times (0, 10, 20, 30, 60, 120, 180, 240 min) with enlarged view of shaded area. **c** In-situ DRIFTS spectra from 1200 – 1100 upon doubling Ag.W-BTO amount. (a. u.) represents arbitrary units.

reactant of a −CH$_3$O group was not readily observed during the reaction. However, upon doubling the sample amount, a subtle increment in the −CH$_3$O peak became evident as time increased, which confirmed that it produced the *CH$_3$O (Fig. 7c). Furthermore, to elucidate the preference of CO$_2$RR and MOR in generating *CH$_2$O over *CH$_3$O intermediates on Ag and W species, respectively, controlled in-situ DRIFTS experiments were conducted by flowing CO$_2$ with H$_2$O (CO$_2$ + H$_2$O) and CH$_3$OH with Ar (CH$_3$OH + Ar) on Ag-BTO and W-BTO catalysts, respectively. During the in-situ DRIFTS reaction of flowing CO$_2$ and H$_2$O on the surface of Ag-BTO, the emergence of CO$_2$RR intermediates such as *CO, *COOH, and notably *CH$_2$O which are located about 2060, 1716, and 1260 cm$^{-1}$ was observed, suggesting Ag with sensitivity to CO$_2$RR and its tendency to produce *CH$_2$O intermediates (Supplementary Fig. 42a). Conversely, in the case of W-BTO, the production of these three intermediates, especially *CH$_2$O, did not increase with reaction time, indicating a lack of sensitivity towards *CH$_2$O generation (Supplementary Fig. 42b). This suggests that Ag is more inclined to produce *CH$_2$O intermediates. In the in-situ DRIFTS spectrum of flowing CH$_3$OH and Ar on the surface of the W-BTO catalyst, a significant increase in the peak at 1155 cm$^{-1}$ corresponding to the *CH$_3$O intermediate was observed (Supplementary Fig. 43a). However, no notable enhancement in the intensity of the *CH$_3$O peak was detected on the Ag-BTO surface when CH$_3$OH and Ar were introduced (Supplementary Fig. 43b). This indicates that W species are more effective in promoting the oxidation of CH$_3$OH to *CH$_3$O intermediates. In addition, in-situ DRIFTS experiments were conducted by flowing CO and CH$_3$OH to explore whether CO participate in the reaction. The bands of *CO (2060 cm$^{-1}$) and *CH$_3$O (1155 cm$^{-1}$) intermediates are produced immediately after the adsorption of CO and CH$_3$OH for 1 h in the dark (Supplementary Fig. 44). Subsequently, a band corresponding to the *CH$_2$O intermediate (around 1260 cm$^{-1}$) reduced by *CO, appeared after UV-visible light irradiation which is consistent with our performance result. The peak intensity of these intermediates (*CO, *CH$_2$O, and

*CH$_3$O) exhibits a smaller increase during the reaction, possibly due to the consumption of these intermediates during the synthesis of DMM at a higher reaction rate.

Although W and Ag tend to serve as oxidation and reduction sites, respectively, their roles have been confirmed by in-situ DRIFTS, band gap structures, and controlled experiments. The electronic structures and adsorption characteristics concerning CO$_2$ and CH$_3$OH were investigated through DFT calculations to further delve into the mechanisms and active sites for this reaction. The optimized structures of Ag.W-BTO were established based on TEM, XPS, and XANES analyses (Supplementary Fig. S45). The adsorption energies of CO$_2$ and CH$_3$OH on distinct active sites (W, Ag, Ti, and O$_v$) were analyzed to confirm the reduction and oxidation sites on the Ag.W-BTO sample (Supplementary Fig. 46 and 47). As depicted in Fig. 8a, all the active sites (W, Ag, Ti, and O$_v$) displayed negative values for CO$_2$ adsorption energies, indicating a higher tendency for CO$_2$ adsorption across these sites. The most negative adsorption energy for CO$_2$ (−2.413 eV) signifies a preference for CO$_2$ adsorption on the Ag sites within the Ag.W-BTO sample. This highlights the heightened propensity of the Ag sites within Ag.W-BTO for CO$_2$RR. The most negative adsorption energy (−1.375 eV) observed for W species during CH$_3$OH adsorption suggests their higher inclination towards undergoing MOR. In addition, the Charge density difference and Bader charge analyses were conducted to explore the reason for the superior performance of Ag.W-BTO. The Ag-BTO and Ag.W-BTO were chosen as an analysis model due to their better performance in CO$_2$RR (Fig. 8b). The adsorbed CO$_2$ species on Ag.W-BTO gained a notably higher quantity of electrons (1.49 e$^-$) than on Ag-BTO (1.46 e$^-$). These results confirmed more electrons can participate in the CO$_2$RR process in Ag.W-BTO. It further validates that the combined effect of Ag and W facilitates the separation and transfer of electrons and holes during the reaction process on Ag.W-BTO, which is consistent with findings from fs-TA and PL data.

To further explore the formation pathway of DMM in the coexistence of CO$_2$ and CH$_3$OH, the free energy profile and reaction

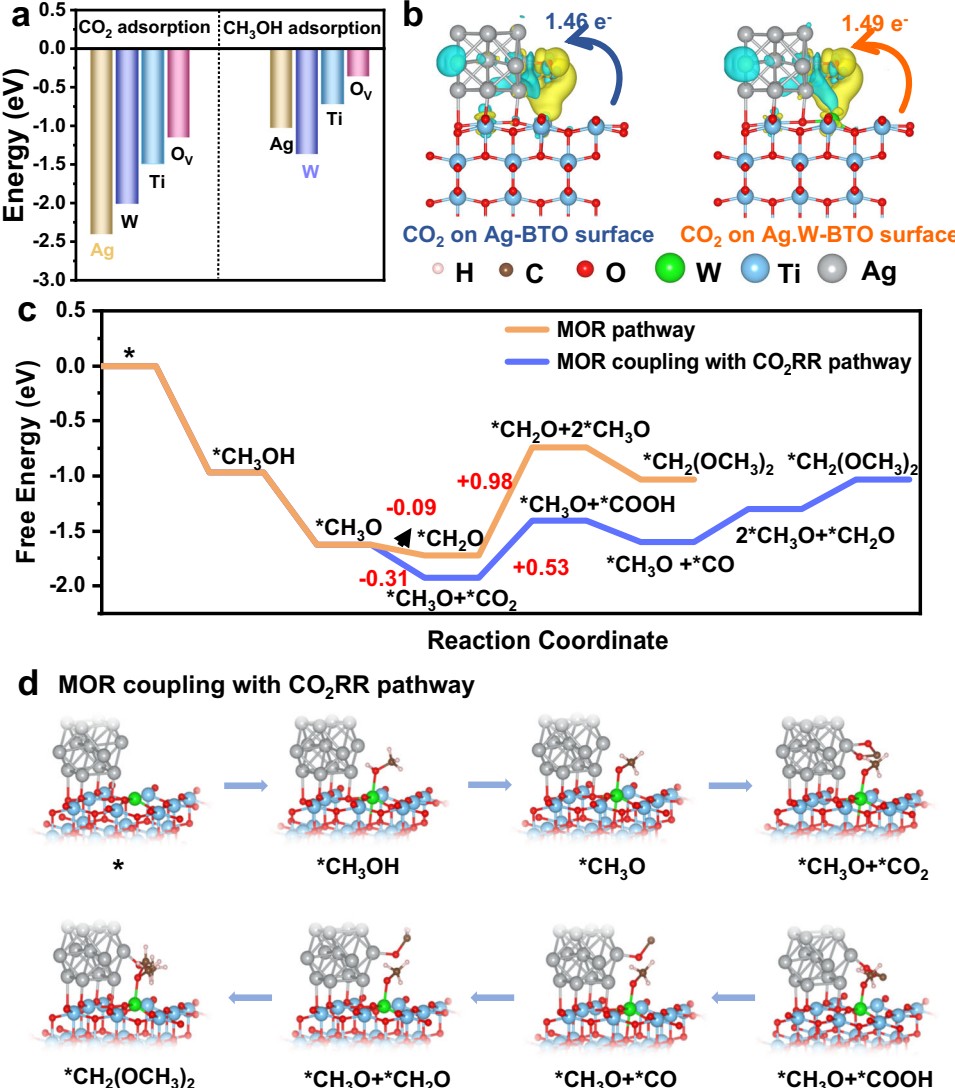

**Fig. 8 | DFT calculations. a** Calculated $CO_2$ and $CH_3OH$ adsorption energy on the different active sites (Ag, W, Ti, and $O_v$) of Ag.W-BTO, respectively. **b** Charge density difference and Bader charge analysis of $CO_2$ adsorbed on the Ag-BTO and Ag.W-BTO surface. **c** Free energy diagram of DMM production via different pathway. **d** Geometries of reaction intermediates involved in MOR coupling with $CO_2RR$ pathways.

pathway of the DMM synthesis were compared on the Ag.W-BTO in different routes (MOR pathway and MOR coupling with $CO_2RR$ (MOR + $CO_2RR$) pathway, Fig. 8c). The optimized intermediate structures corresponding to each reaction steps are displayed in Fig. 8d and Supplementary Fig. 48. The DMM formation on Ag.W-BTO through two pathways is triggered by the thermodynamical spontaneity from $CH_3OH$ to *$CH_3OH$ with an energy barrier of −0.98 eV. Then, *$CH_3OH$ is further oxidized to *$CH_3O$ with an energy barrier of −0.65 eV. Thereafter, it was found that the process from *$CH_3O$ to *$CH_3O$ + *$CO_2$ with an energy barrier of −0.31 eV is more thermodynamically favorable than the oxidation of *$CH_3O$ to *$CH_2O$ with an energy barrier of −0.09 eV, which suggests that *$CH_3O$ tends to couple with *$CO_2$ intermediates rather than to form individual *$CH_2O$ intermediates in $CH_3OH$ and $CO_2$ coexist system. In addition, the process of the next step (*$CH_2O$ + 2*$CH_3O$) in MOR pathway free energy barrier (0.98 eV) requires higher than *$CH_3O$ + *COOH (0.53 eV) in MOR + $CO_2RR$ route. That means direct oxidation of $CH_3OH$ to DMM is unfavorable. Moreover, three steps require energy input in the MOR + $CO_2RR$ pathway, including the conversion of *$CO_2$ to *COOH intermediates, the reduction of *CO to *$CH_2O$, and the coupling of *$CH_2O$ with *$CH_3O$.

Particularly, the conversion of $CO_2$ to *COOH intermediates consumes the highest energy of 0.53 eV and is considered the rate-determining step during the DMM formation process.

According to the in-situ DRIFTS analysis, systematic isotope labeling experiments, and DFT calculations, the most probable reaction paths for the catalytic system are:

**Overall**

$$2CH_3OH + CO_2 \rightarrow (CH_3O)_2CH_2(DMM) + O_2$$

**In details**

$$Ag.W-BTO \xrightarrow{h\nu} h^+ + e^-$$

**MOR:**

$$CH_3OH_{ads}(W) + h^+ \rightarrow \bullet CH_3O + H^+$$

$$CH_3OH_{ads}(W) \rightarrow \bullet CH_3O + H^+ - h^+$$

**CO₂RR:**

$$CO_{2\,ads}\,(Ag) + e^- + H^+ \rightarrow \bullet COOH$$

$$\bullet COOH + e^- + H^+ \rightarrow \bullet CO + H_2O$$

$$H_2O \rightarrow 2H^+ + 2e^- + \bullet O$$

$$\bullet CO + 2e^- + 2H^+ \rightarrow \bullet CH_2O$$

**Total of CO₂RR**

$$CO_{2\,ads}\,(Ag) + 2e^- + 2H^+ \rightarrow \bullet CH_2O + \bullet O$$

$$CO_{2\,ads}\,(Ag) \rightarrow \underline{\bullet CH_2O + \bullet O - 2e^- - 2H^+}$$

**Coupling process:**

$$2CH_3OH + CO_2 \rightarrow (CH_3O)_2CH_2 + O_2$$

$$\underline{2(\bullet CH_3O + H^+ - h^+)} + (\underline{\bullet CH_2O + \bullet O - 2e^- - 2H^+})$$

$$\rightarrow (2\bullet CH_3O + 2H^+ - 2h^+) + (\bullet CH_2O + \bullet O - 2e^- - 2H^+)$$

$$\rightarrow 2\bullet CH_3O + \bullet CH_2O + \bullet O + 2H^+ - 2H^+ \rightarrow (CH_3O)_2CH_2(DMM) + O_2$$

$CH_3OH$ and $CO_2$ are adsorbed on the surface of Ag.W-BTO in the dark, with more significant adsorption evidence on the W and Ag active sites, respectively. During UV-visible light irradiation, electrons ($e^-$) in the VB of Ag.W-BTO are excited to the CB, while the holes ($h^+$) remain in the VB. Some $e^-$ transfer to Ag sites by Schottky-junction for CO₂RR, leaving $h^+$ to oxidize $CH_3OH$ on the W sites. $CH_3OH$ is oxidized to form *$CH_3O$ and $H^+$ intermediates after binding to $h^+$ on the W sites. Simultaneously, $CO_2$ molecules obtain $e^-$ and combine with $H^+$ to produce COOH* on the active Ag sites. Furthermore, *COOH is reduced to *CO and then changed to *$CH_2O$ by a multi-step proton-coupled electron transfer (PCET) process. Finally, DMM is produced by coupling *$CH_2O$ with *$CH_3O$.

## Discussion

In summary, A photo-redox system that can simultaneously couple solar-driven CO₂RR with MOR on Ag.W-BTO dual-functional catalysts to produce a value-added chemical (DMM) was proposed. The selectivity of DMM approached 92.08% on Ag.W-BTO and was accompanied by a record-high yield of 5702.49 μmol g⁻¹ after 9 h UV-visible irradiation without sacrificial agents. Validation of the synergistically coupled multi-step PCET mechanism for DMM formation was achieved through systematic isotope labeling experiments, in-situ DRIFTS analysis, and DFT calculations. The Ag species were largely responsible for the facile $CO_2$ adsorption and reduction to *$CH_2O$, while the W dopant promoted $CH_3OH$ oxidation to *$CH_3O$. The two obtained intermediates couple to synthesize DMM. This work provides a concept for green photochemical synthesis of high-value chemicals by coupling $CO_2$ reduction with another small molecular conversion and the fine design of photocatalysts.

## Methods

### Reagents

All chemicals were used as received without further purification. Pure titanium dioxide ($TiO_2$, $P_{25}$) was obtained from Degussa (Korea), lithium (Li), ethylenediamine (EDA), hydrochloric acid (HCl), silver nitrate ($AgNO_3$), tungsten hexachloride ($WCl_6$), sodium borohydride ($NaBH_4$), ethanol ($C_2H_5OH$), methanol ($CH_3OH$), ¹³C labeled $CH_3OH$

(¹³$CH_3OH$), ¹⁸O labeled $CH_3OH$ ($CH_3$¹⁸OH), formaldehyde solution ($CH_2O$), and dimethoxymethane solution (DMM, $C_3H_8O_2$) were purchased from Sigma-Aldrich (Korea). Carbon dioxide ($CO_2$) gas (99.99%), hydrogen ($H_2$) gas (5%), oxygen ($O_2$) gas (99.99%), carbon monoxide (CO) gas (99.99%), and methane ($CH_4$) gas (99.99%) were provided by Deokyang Co., LTD (Korea), ¹³$CO_2$-labeled gas (99.99%, 2.16 L) was obtained by Korea noble gas Co., LTD. The ID water used in all experiments was purified using a Millipore system.

### Samples synthesis

**Synthesis of blue TiO₂ (BTO).** The synthesis of BTO by using the Li-treated method. Specifically, type $P_{25}$ (1 g) and 694 mg of metallic Li were gradually put into a rubber-closed system and left for 30 min in a vacuum state. Then, 100 ml EDA was injected into this closed system under Ar gas condition with ice. These mixed reactions were stirred for 3 days. Next, 1 mol L⁻¹ HCl was slowly dropped into the mixture to form Li salts and quench the excess electrons. Finally, the resulting blue material was washed with DI water several times and dried in a vacuum oven at 50 °C.

**Synthesis of Ag.W-BTO-related samples.** First, 250 mg of BTO and 500 mg of $NaBH_4$ were vacuumed for 15 min, then placed in an ice bath and added to 30 mL ethanol while stirring until completely dispersed. 100 mg of $WCl_6$ was dispersed in 5 mL ethanol, and 100 mg of $AgNO_3$ was dispersed in 5 mL DI water, respectively. Then, slowly inject these two dispersed solutions into the BTO solution. The mixed solution was then stirred for 90 min in an ice bath before being washed many times with DI water, and ethanol and finally dried in a vacuum oven. The powder was named Ag.W-BTO. The different amounts of Ag by the same method (50 mg and 150 mg) were signaled as Ag.W-BTO-50 and Ag.W-BTO-150, respectively. W-BTO and Ag-BTO were produced using the same method except for not additionally adding the $AgNO_3$ and $WCl_6$.

### In-situ DRIFTS measurements

The surface of Ag.W-BTO was analyzed by in-situ DRIFTS during the CO₂RR + MOR experiment under UV-visible light to explore the reaction mechanism. Initially, a mixture was prepared by combining 10 mg of the Ag.W-BTO sample with 190 mg of KBr powder, maintaining a weight ratio of 1:19 between the sample and KBr. The mixture was then loaded into a sample cup within the reaction cell, positioned beneath the center of three windows, comprising one quartz window and two infrared-transparent windows. Subsequently, the high-purity $CO_2$ gas (99.99%) bubbling with $CH_3OH$ was flowed in the reaction cell for adsorbing on the catalyst for 1 h under dark conditions. Hereafter, the $CO_2$ reduction reaction coupling with the $CH_3OH$ oxidation reaction began under the UV-visible light (320–780 nm) during continuous $CO_2$ flow, and in-situ DRIFTS spectra were collected at specific reaction times. It is worth highlighting that background IR spectra were obtained by measuring KBr powder alone under identical experimental conditions. Subsequently, each background spectrum was subtracted from the corresponding DRIFTS spectrum acquired from the samples (Ag.W-BTO and KBr powder) to eliminate the influence of KBr powder in the analysis. Other reaction conditions are the same as photocatalytic reaction conditions. For checking in-situ DRIFTS spectra from 1200 to 1100 cm⁻¹ on Ag.W-BTO, the process was the same as the previous method except that the Ag.W-BTO sample (20 mg), which had 10 mg of more sample, was mixed with 180 mg KBr powder (the weight ratio of the sample and KBr was 1:9).

### Photo-reaction measurements

**Standard testing.** The direct photogeneration of DMM was performed from the reaction of $CO_2$ and $CH_3OH$ in a 50 mL stainless-steel autoclave. The system of photocatalytic experiments was set up as shown in Supplementary Fig. 18. Specifically, add 25 mg catalysts and 30 ml

$CH_3OH$ in the stainless-steel inlet, followed by sonication for 30 min. Then, after adding the stirrer bar inside, flowing $CO_2$ gas (99.99%) was used for 1 h to purge the air in the reactor in dark conditions. The autoclave was finally pressurized with $CO_2$ pressure to 0.2 MPa. Finally, the reaction was performed at 373 K with stirring at 400 r.p.m. for 9 h. The light source for the photocatalysis was a 300 W Xenon lamp (320–780 nm, PLS-SXE300, Beijing Perfectlight Technology Co., Ltd). The GC system detected the amounts of products. For in-operando checking, 250 uL product was introduced to the GC at different time intervals (0, 3, 6, and 9 h). In the stability test, the $CO_2$ reduction was repeated for 4 cycles under the same condition.

**Comparison testing.** For comparison, several other reaction conditions were also carried out under identical conditions to evaluate the role of the $CO_2$ gas and $CH_3OH$: (a) $CO_2$ gas + $H_2O$ + Ag.W-BTO sample (Without $CH_3OH$); (b) $CH_3OH$ + Ar + Ag.W-BTO sample (Without $CO_2$); (c) $CO_2$ gas + $CH_3OH$ + Ag.W-BTO sample under dark condition (Dark); (d) $CO_2$ gas + $CH_3OH$ (Without catalyst). The amount of production was also checked by CG.

**Isotopic $^{13}CO_2$ experiments.** $^{13}$C-labeled $CO_2$ ($^{13}CO_2$) gas was used as the feed gas in the labeling experiment. The reaction condition was the same as the common $CO_2$ reaction conditions except for using Ar to purge the air in the reactor due to the limited supply and expense of $^{13}CO_2$ gas. The final production was checked by GC-MS.

**Isotopic $CH_3OH$ experiments.** $^{13}$C-labeled $CH_3OH$ ($^{13}CH_3OH$) and $^{18}$O-labeled $CH_3OH$ ($^{13}CH_3OH$) react with $CO_2$, respectively. The final production was checked by GC-MS.

### Determination of the production
**Quantitative test by GC.** The hydrogen ($H_2$) calibration data were collected and plotted. A predetermined amount of $H_2$ gas (0, 0.0446, 0.0670, 0.0892, 0.111 μmol) was injected into the GC. The concentration-area curves were calibrated, and a fitted curve for which the $R^2$ value was determined to be 0.999, indicating a highly strong linear correlation (Supplementary Fig. 19a). y (area) = 15114.35338x (μmol) is the linear fitting equation. After a 9 h $CO_2$ reduction reaction, the produced gas (250 μL) was injected into the GC to calculate the amount.

The calibration data of carbon monoxide (CO) were obtained and plotted. Certain amounts of CO gas (0, 0.0514, 0.103, 0.154, 0.206, 0.257 μmol) were injected into GC, the concentration-area curves were calibrated, and a fitted curve exhibited the $R^2$ value of 0.99907, indicating a strong liner relationship (Supplementary Fig. 19b). The linear fitting equation is y (area) = 180921.20423x (μmol). After the $CO_2$ reduction reaction (9 h), the production (250 μL) gas was injected into the GC to calculate the amount.

The calibration data of methane ($CH_4$) were obtained and plotted. A certain amount of $CH_4$ gas (0, 0.102, 0.204, 0.306, 0.408, 0.510 μmol) was injected into GC. The concentration-area curves were calibrated, and a fitted curve for which the $R^2$ value was determined to be 0.99909, indicating a highly strong linear correlation (Supplementary Fig. 19c). The linear fitting equation is y (area) = 174925.83406x (μmol). After the $CO_2$ reduction reaction (9 h), the produced gas (250 μL) was injected into the GC to calculate the amount.

The calibration data of methanol ($CH_3OH$) were obtained and plotted. A certain concentration of $CH_3OH$ liquid (4.88, 9.76, 29.28, 39.05 μmol) was injected into GC, the concentration-area curves were calibrated, and a fitted curve exhibited the $R^2$ value of 0.9996 indicating a strong liner relationship (Supplementary Fig. 19d). The linear fitting equation is y (area) = 34812.349x (μmol). Before and after the $CO_2$ reduction reaction (9 h), the liquid (5 μL) was injected into the GC to calculate the amount.

The calibration data of oxygen ($O_2$) were obtained and plotted. A certain amount of $O_2$ gas (0, 0.438, 0.875, 1.313, 1.750, 2.188 μmol) was injected into GC. The concentration-area curves were calibrated, and a fitted curve for which the $R^2$ value was determined to be 0.999, indicating a highly strong linear correlation (Supplementary Fig. 19e). The linear fitting equation is y (area) = 1097.052x (μmol). After the $CO_2$ reduction reaction (9 h), the produced gas (250 μL) was injected into the GC to calculate the amount.

The calibration data of formaldehyde ($CH_2O$) were obtained and plotted. Certain amounts of $CH_2O$ were dissolved in DI water to obtain a series of concentrated solutions (0.3125, 0.625, 1.25, 2.5, 5 mmol L$^{-1}$). These different concentration solutions (5 uL) were injected into GC, and the rotation time of $CH_2O$ was about 14.31 min, resulting in a calibration curve. The concentration-area curves were calibrated, and a fitted curve exhibited the $R^2$ value of 0.99917, indicating a strong liner relationship (Supplementary Fig. 20). The linear fitting equation is y(area) = 659.14525x (mmol L$^{-1}$). After the $CO_2$ reduction reaction (9 h), the production (5 uL) liquid was injected into the GC to calculate the amount.

The calibration data of dimethoxymethane ($C_3H_8O_2$) were obtained and plotted. A certain amount of $C_3H_8O_2$ was dissolved in DI water to obtain a series of concentrated solutions (0.02, 0.04, 0.08, 0.125, 0.25, 0.5, 1 mol L$^{-1}$). These different concentration solutions (5 uL) were injected into GC, and the rotation time of $C_3H_8O_2$ was about 22.89 min, resulting in a calibration curve. The concentration-area curves were calibrated, and a fitted curve exhibited the $R^2$ value of 0.99984, indicating a strong liner relationship (Supplementary Fig. 21). The linear fitting equation is y(area) = 2.45663x (mol L$^{-1}$). After the $CO_2$ reduction reaction (9 h), the production (5 uL) gas was injected into the GC to calculate the amount.

**Qualitative analysis by GC-MS.** After the $CO_2$ reduction reaction, the production was injected into GC-MS (5 μL).

### Calculation of Selectivity
The gas products ($H_2$, CO, $CH_4$ and $O_2$) amount (μmol g$^{-1}$), liquid products ($CH_2O$, $C_3H_8O_2$) amount (μmol g$^{-1}$) after 9 h reaction were calculated using equations presented in the following:

$$H_2 = \frac{Vr*S}{15114.35338*W*Vn*1000} \tag{3}$$

$$CO = \frac{Vr*S}{180921.20423*W*Vn*1000} \tag{4}$$

$$CH_4 = \frac{Vr*S}{174925.83406*W*Vn*1000} \tag{5}$$

$$O_2 = \frac{Vr*S}{1097.052*W*Vn*1000} \tag{6}$$

$$CH_2O = \frac{VL*S}{659.14525*1000*W}*1000 \tag{7}$$

$$C_3H_8O_2 = \frac{VL*S}{2.45663*10^6*W}*1000 \tag{8}$$

$$\text{Selectivity} = \text{Yield DMM}*2/\text{Consumption } CH_3OH*100\% \tag{9}$$

Where $Vr$ (mL) is the volume of the reactor, $VL$ (mL) is the volume of liquid, $Vn$ (μL) represents product volume injected into GC, $S$ is the area of the product obtained by GC, and $W$ (g) is the mass of catalyst.

## Data availability

The data that support the conclusions of this study are available within the paper and supplementary information. Source data are provided with this paper. Figshare https://doi.org/10.6084/m9.figshare.25713021. Source data are provided with this paper.

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

## Acknowledgements

This work was supported by the National Research Foundation of Korea (NRF) grant (NRF-2022R1A2C2093415), the Circle Foundation (Republic of Korea) for 1 year since December 2023 as CO$_2$ to Multi-carbon Production Center selected as the '2023 The Circle Foundation Innovative Science Technology Center' (Grant Number: 2023 TCF Innovative Science Project-03), and partially supported by the Korea Basic Science Institute (National Research Facilities and Equipment Center) grant funded by the Ministry of Education (2022R1A6C101A751). The authors thank the BL10C beamline of the Pohang Light Source (PLS-II, Korea) Facility for providing synchrotron beam time. The authors thank the China Scholarship Council (CSC: 202208260012) supported.

## Author contributions

Y.W. carried out the experiments, conceptualization and writing draft. Y.L. gave direction, supervision, and co-writing draft. L.W., S.P., H.W., H.K., J.Z., and Y.H. helped in the data discussion. T.N. and C.-L.D. fitted EXAFS data. P.Z. and S.W. helped with DFT calculations. Y.K., Y.J, and S.Z. helped in collecting the in-situ DRIFTS data. J.-H.K. and D.-Y.Y. helped in collecting the fs-TA data. M.G.K. conducted an XAS experiment. H.L. provided direction, supervision, and editing of this paper.

## Competing interests

The authors declare no competing interests.
