## [Peer Review File · Nature Communications]

Coupling photocatalytic of CO₂ reduction and CH₃OH oxidation for selective dimethoxymethane productionREVIEWER COMMENTS

Reviewer #1 (Remarks to the Author):

In this submission, a novel photo-redox system is reported that can simultaneously couple solar-driven CO₂RR with methanol oxidation reaction (MOR) on a Ag.W-BTO dual-functional catalysts to produce dimethoxymethane (DMM). Alternative routes to generate DMM are certainly of high importance, as this molecule provides routes to both clean fuels and chemicals, as noted by the authors. The submitted work is of interest, especially due to the high selectivity to DMM at more than 90%. The use of isotopic experiments is a great addition to this work and is a critical piece to put together the mechanistic discussion. Questions remain about the proposed 4 PCET steps to get from CO₂ to formaldehyde, as noted below. The following group of both minor and major points must be addressed:

(1) in the introduction: The authors should note the work of Palkovits (Aachen, Germany: Sustainable Energy Fuels, 2021, 5, 117) and the group at NREL (Colorado, US: ACS Sustain Chem Eng, 2020, 8, 12151) who are also investigating alternative routes to DMM via methanol oxidation with good results for comparison to the submitted work. These should be summarized in the opening paragraph with the other early references.

(2) Line 38: The first sentence is incorrect. Oxymethylene dimethyl ether has middle units of -O-CH₂, not DMM. Better to just give the DMM structure, CH₃-OCH₂-OCH₃, if that is needed.

(3) Lines 95-96: should this read “The designed model catalysts were prepared by simply reducing [WCl₆ and AgNO₃ with] sodium borohydride.”? In general, the description of the materials synthesis is subpar and needs improvement to be clear and reproducible to the reader.

(4) Lines 159-162: To support the claim that a difference of 0.02 Å in the Ag-Ag bond is real, the authors must report the error in these measurements. This is especially important considering the bulk nature of XAS (where the data represents the average of all Ag-Ag bonds, not just those near the W-BTO) and the claim that a large number of electrons were transferred from Ag to the W-BTO to result in a shortened bond distance. At this point in the paper, this seems like an overly analyzed piece of data, and the more simple explanation that the Ag-Ag bonds are within error of each other, is more apparent to the reader when looking at the plots.

(5) Lines 192-195: as plotted, the N₂ physisorption data do not clearly show type IV isotherms. The plots have too much overlapping data on them to be readable. This data is not clearly presented. If more porosity is generated through the incorporation of the Ag and W on BTO, it should be evident in the TEM images. Comparing this N₂ phys data in light of the TEM images is needed to support the explanation of higher porosity.

(6) Lines 214-215: the statement “This phenomenon implies that *CO intermediates participated in producing DMM on the Ag.W-BTO surface under the synergistic effect of Ag and W.” is highly speculative,

and not based on the data presented at this point in the manuscript. One could easily argue that CO production is greatly inhibited on the Ag.W-BTO catalyst in favor of CH₂O, which is why it generates high selectivity to DMM rather than gaseous products. The DRIFTS data presented later informs this statement, but I understand that 4 PCET steps are needed to generate formaldehyde (the critical intermediate to form DMM) from CO₂. It may be that this statement appears too early in the manuscript, and the explanation is best left when the elementary reactions are presented in Lines 351-355. The authors should perform specific experiments to support this claim, such as feeding CO instead of CO₂ and monitoring catalytic performance and DRIFTS speciation. One would expect the rate of DMM production to be significantly higher if fewer PCET steps are needed to use CO instead of CO₂. If the same rate is observed, it may be that there is a different rate-limiting step, which would also be a great addition to this report.

(7) Finally, The 'Discussion' section is perhaps best noted as a summary, needing no header, and beginning with 'In summary,...'.

Reviewer #2 (Remarks to the Author):

This work proposes a dual-functional catalyst system employing a silver and tungsten co-modified blue titanium dioxide catalyst (Ag.W-BTO) for the efficient generation of DMM through the synergistic coupling of CO₂ reduction and CH₃OH oxidation under mild reaction conditions. The selectivity of DMM reached 92.08% on Ag.W-BTO, accompanied by a yield of 5702.49 μmol g⁻¹ after 9 hours of UV-visible irradiation without the use of sacrificial agents. The author considers that the Ag and W species serve as catalysts for the conversion of CO₂ to *CH₂O and the oxidation of CH₃OH to *CH₃O, respectively. Nevertheless, this view lacks sufficient evidence and the content of the manuscript is incomplete. This work is not suitable for acceptance by Nature communications. The following concerns regarding this manuscript should be addressed:

1. It is well known that Ag exhibits localized surface plasmon resonance (LSPR) effect, which can enhance the surface temperature of a catalyst upon visible light irradiation. Given the high Ag content in the Ag.W-BTO catalyst, careful consideration must be given to the impact of elevated temperatures during the reaction process.
2. Figure 3a shows that there are two characteristic peaks, Ti³⁺ and Ti⁴⁺, and the content of Ti³⁺ accounts for the vast majority. In reported works, the oxidation state of titanium in the blue TiO₂ is predominantly +4, so the XPS spectra of Ti 2p need to be reconsidered (Advanced Functional Materials 2017, 27: 1700856; ACS Applied Materials & Interfaces, 2016, 8(40): 26851-26859; Advanced Energy Materials, 2020, 10(10): 1903107). Additionally, EPR characterization is required to verify the presence of both Ti³⁺ and oxygen vacancies.
3. The authors provided a detailed characterization of Ti³⁺ in blue TiO₂. However, its correlation with the reaction remains unestablished. Further clarification is requested.
4. The article mentioned that "W⁶⁺ can be gradually transformed to unsaturated W⁵⁺", thus relevant

characterization should be provided as evidence.

5. Typically, in XAS analysis, metal K-edge compares near-edge absorption positions, while metal L3-edge requires comparison based on white line peak positions. Therefore, it is necessary to revise descriptions related to W L3-edge accordingly. The results in Figure g-i should be obtained by the Fourier transform of EXAFS spectrum, but the processing results seem to be wrong. In the processing process, attention should be paid to adjusting parameters, removing bad points and K-space values. All R-space data needs to be reprocessed and discussed. In addition, the Ti-O, Ag-O, W-O and W-O-Ti bonds mentioned in the paper need to provide fitting data and charts to verify whether the fitting is reasonable.

6. In Figure 4f, the m/z value for CH₃18O was indicated as 33 which is inconsistent with the position shown as 32 in the figure.

7. The author needs to explain why *CH₃O exists at 0 minutes (no light), as shown in Figure 6c.

8. Theoretical calculations alone cannot offer sufficient evidence regarding the active sites of Ag and W species in CO₂ conversion to *CH₂O and CH₃OH oxidation to *CH₃O. Therefore, it is imperative to provide compelling evidence.

9. Do Ag and W have an equivalent impact on other substrates?

Reviewer #3 (Remarks to the Author):

The manuscript " Photocatalytic coupling of CO₂ reduction and CH₃OH oxidation for over 90 % selective dimethoxymethane production" presents a novel method or selective dimethoxymethane (DMM) production by photocatalytic coupling of CO₂ reduction and CH₃OH oxidation. The photo-redox catalysis system is based on the a silver and tungsten co-modified blue titanium dioxide (Ag.W-BTO) photocatalyst. The research includes comprehensive investigation of the Ag.W-BTO structure, electron and hole transfer behaviors, and a systematic exploration of the photocatalytic activity, selectivity, and mechanisms involved in DMM production. The content of the manuscript is generally systematic and logical, and the elucidation of the mechanism makes sense. This manuscript can be further considered by Nat. Commun. subject to a revision, along the lines suggested hereafter:

1. The fitting of the XPS spectra in your study appears to be inaccurate, particularly in the representation of peak widths at half height for the same species. Notably, in Figure 3a, there are inconsistencies in the widths of Ti³⁺ peaks. It is crucial to maintain uniformity in peak widths for the same species to ensure accuracy in spectral analysis. I recommend a thorough re-fitting of your XPS spectra to address these discrepancies and improve the reliability of your findings.

2. The current approach of determining the band position of the photocatalyst using XPS-VB and absorption spectra has some methodological inaccuracies. It's important to note that XPS-VB spectra only reveal the relative position from the valence band to the Fermi level. Since your article does not establish the Fermi level's position in the catalyst, this method falls short of providing a complete picture. To accurately ascertain the band positions, Ultraviolet Photoelectron Spectroscopy (UPS) should be employed. UPS offers a more direct and reliable measurement for determining the band positions in

photocatalytic materials.

3. In fact, the process of preparing thin films on FTO significantly influences the photocurrent and impedance. Therefore, it is crucial for the authors to provide a detailed description of the experimental procedures used for sample preparation on FTO.

4. The authors should provide the Apparent Quantum Yield (AQY) of the photocatalysts, which is a crucial metric in evaluating the performance of photocatalytic materials.

5. The authors should provide a more rigorous explanation regarding the experiments on charge separation, such as PL, photocurrent, and transient absorption measurements. It is important to note that these tests do not replicate actual reaction conditions. Therefore, attributing observations from these tests directly to the actual reaction processes can be misleading. A more accurate interpretation of the experimental results in the context of the actual reaction conditions is needed to validate the findings related to charge separation in the photocatalytic process.

6. XPS spectra of the Ag₂W-BTO photocatalysts before and after the stability test should be provided.

Reviewer #4 (Remarks to the Author):

This study investigates the production of dimethoxymethane (DMM) using a photo-redox catalytic process involving a catalyst made from silver and tungsten-modified blue titanium dioxide (BTO). The authors suggest that this process combines CO₂ reduction with methanol oxidation to produce DMM. They conducted several experiments, including control tests and using isotopic reagents, to support their concept.

Although the concept of the study is stimulating, the results have some notable shortcomings. A key challenge is determining the exact source of the DMM as could be a result of both methanol oxidation and CO₂ reduction, or solely methanol oxidation.

The authors conducted two main sets of experiments to back their claim of producing DMM from a combination of methanol and CO₂. First, they compared results under CO₂ with those under argon, finding seven times higher activity under CO₂. Second, they used different labeled molecules (like methanol-d₄, methanol-C₁₃, and CO₂-C₁₃) in their tests. While I appreciate the effort in conducting these detailed experiments, I disagree with how they interpreted the results.

The increased activity under CO₂ could also be due to a lower pH in the solution, which might promote methanol oxidation and its subsequent reaction with formaldehyde. This implies that DMM could form without CO₂, but not without methanol (43 times less). Adjusting the pH in the solution under argon conditions could further clarify this point. Additionally, the main mass spectrometry signal for DMM should be at $m/z=75$ (m/z (DMM) - m/z (H⁺)), and not at 76 (much less intense) as the authors suggest. This is further supported by the observation of DMM-d₇ showing a main signal at m/z 82 (Figure S27; not at 84 which is the last small peak on the figure) as one might expect. This discrepancy calls for a

revision of their interpretations of the labeled experiments as following:

In the experiment using $^{13}\text{CH}_3\text{OH}$ and CO_2 , the mass-to-charge ratio (m/z) of 78 for DMM indicates the presence of three ^{13}C ($75+3$) and not only two as expected. This finding strongly suggests that the primary source of DMM is the oxidation of methanol.

For the experiment performed using ($\text{CH}_3^{18}\text{OH}+\text{CO}_2$) we would expect to see $^{16}\text{O}^{18}\text{O}$ -DMM with a m/z of 77 if the reaction involved CO_2 as well as methanol. Similarly, $^{18}\text{O}^{18}\text{O}$ -DMM should appear at m/z 79 and not at 80 as previously assumed. This lack of detection challenges the hypothesis that CO_2 is playing a significant role in the formation of DMM under these conditions.

The experiment performed using unlabeled methanol and $^{13}\text{CO}_2$ show the main m/z peak at 75 (Figure S27) in agreement with the formation of unlabeled DMM without excluding a minority of C^{13} -DMM.

Furthermore, the production of hydrogen over the catalyst contradicts the expected reaction stoichiometry, which is another point of concern. The origin of the two protons (4H^+) and two electrons (4e^-) as outlined on page 21, line 354 of the study, doesn't align with the proposed reaction formula which should involve 2 protons and 2 electrons ($2\text{CH}_3\text{OH} + \text{CO}_2 = \text{CH}_3\text{OCH}_2\text{OCH}_3$). This discrepancy raises questions about the mechanism proposed by the authors and suggests a need for a more thorough examination of the reaction pathway and the role of the catalyst.

Less critical, but still noteworthy, are concerns about their choice of catalyst (BTO over other types of titanium dioxide) and the misclassification of the isotherms in their analysis (type III and not IV, no mesoporosity but textural inter-particle porosity) and the IR assignments (COOH is confused with H_2O rotational peaks, miss of a band around 1700 assigned to carboxylate of adsorbed formaldehyde, etc.). It's important to note that DMM can also be produced through the photocatalytic oxidation of methanol vapor in the presence of Brønsted acidity (like zeolites) under aerated conditions, without any oxidants or solvents (e.g. Ftouni et al. *TiO₂/Zeolite Bifunctional (Photo)Catalysts for a Selective Conversion of Methanol to Dimethoxymethane: On the Role of Brønsted Acidity*, *J. Phys. Chem. C* 2018 122 (51)). This raises questions about the novelty of the study as DMM.

Given these issues, I cannot recommend the publication of this study in Nature Communications.

Point-by-point Response for Nature Communications manuscript

(Manuscript ID: NCOMMS-23-63082A)

Manuscript Type: Article

Title: Photocatalytic coupling of CO₂ reduction and CH₃OH oxidation for over 90% selective dimethoxymethane production

Author(s): Yixuan Wang, Yang Liu, Lingling Wang, Silambarasan Perumal, Hongdan Wang, Hyun Ko, **Chung-Li Dong**, Panpan Zhang, Shuaijun Wang, **Ta Thi Thuy Nga**, Young Dok Kim, **Yujing Ji**, Shufang Zhao, Ji-Hee Kim, Dong-Yub Yee, Yosep Hwang, Jinqiang Zhang, Min Gyu Kim and Hyoyoung Lee

We are grateful to the editor, editorial staff, and reviewers for their critical comments and valuable suggestions. The manuscript has been modified after addressing all the suggestions listed below:

(Explanations of the reviewers' comments are marked in **BLUE**, and the changes made in the revision are shown with **YELLOW** highlight accordingly).

Author (s)' Points-by-points responses to Reviewers:

Reviewer: 1

Comments:

In this submission, a novel photo-redox system is reported that can simultaneously couple solar-driven CO₂RR with methanol oxidation reaction (MOR) on a Ag.W-BTO dual-functional catalysts to produce dimethoxymethane (DMM). Alternative routes to generate DMM are certainly of high importance, as this molecule provides routes to both clean fuels and chemicals, as noted by the authors. The submitted work is of interest, especially due to the high selectivity to DMM at more than 90%. The use of isotopic experiments is a great addition to this work and is a critical piece to put together the

mechanistic discussion. Questions remain about the proposed 4 PCET steps to get from CO₂ to formaldehyde, as noted below. The following group of both minor and major points must be addressed:

Response: We appreciate your positive comments and valuable suggestions to improve our manuscript. We have made a point-by-point response to your comments and carefully revised the manuscript as you suggested.

Q1-1. In the introduction: The authors should note the work of Palkovits (Aachen, Germany: Sustainable Energy Fuels, 2021, 5, 117) and the group at NREL (Colorado, US: ACS Sustain Chem Eng, 2020, 8, 12151) who are also investigating alternative routes to DMM via methanol oxidation with good results for comparison to the submitted work. These should be summarized in the opening paragraph with the other early references.

Answer 1-1) We sincerely thank you for carefully reviewing our manuscript and providing valuable information for improving its quality. We carefully read these excellent works and added them to our introduction part.

In addition, undesirable peroxide products and carbon oxides are easily produced due to the strong oxidation ability of O₂.^{6, 7} To avoid by-product generation associated with the CH₃OH oxidation route, the CH₃OH dehydrogenation to DMM was explored. The synthesis of DMM via dehydrogenation entails the non-oxidative conversion of methanol into the formaldehyde intermediate, followed by subsequent acetalization reaction of formaldehyde with methanol to yield DMM. For example, Palkovits et al. reached over 80% selectivity of DMM over the Cu/zeolite catalyst under a gas-phase reactor.⁸ Most recently, To et al. achieved 40% of the DMM equilibrium-limited yield under mild conditions (200 °C, 1.7atm) based on Cu-zirconia-alumina (Cu/ZrAlO) catalyst.⁹ Nonetheless, the primary challenge is the requirement for harsh conditions, such as high reaction temperatures, to overcome the thermodynamic constraints of gas-phase CH₃OH dehydrogenation. Since carbon dioxide (CO₂) emissions contribute to global warming^{10, 11}, the conversion of CO₂ to a high-value chemical with a lower global warming potential could make a substantial contribution to the global effort to mitigate climate change.¹² Recently, CO₂ has been considered a feedstock for DMM production through a more environmentally friendly process, one of the most popular approaches is direct synthesis by

reacting CO₂ and CH₃OH. (Revised Manuscript on pages 2 and 3)

8. Sun R, *et al.* Hydrogen-efficient non-oxidative transformation of methanol into dimethoxymethane over a tailored bifunctional Cu catalyst. *Sustainable Energy & Fuels* **5**, 117-126 (2021).
9. To AT, *et al.* Dehydrogenative coupling of methanol for the gas-phase, one-step synthesis of dimethoxymethane over supported copper catalysts. *ACS Sustain. Chem. Eng.* **8**, 12151-12160 (2020).

Q1-2. Line 38: The first sentence is incorrect. Oxymethylene dimethyl ether has middle units of -O-CH₂, not DMM. Better to just give the DMM structure, CH₃-OCH₂-OCH₃, if that is needed.

Answer 1-2) We thank the referee's great comments on our work. We change this sentence as below:

Dimethoxymethane (DMM, CH₃O-CH₂-OCH₃) is an attractive compound for numerous applications, including its use as a fuel additive that can enhance diesel fuel yields and as a precursor of oxymethylene dimethyl ether. (Revised Manuscript on page 2)

Q1-3. Lines 95-96: should this read "The designed model catalysts were prepared by simply reducing [WCl₆ and AgNO₃ with] sodium borohydride."? In general, the description of the materials synthesis is subpar and needs improvement to be clear and reproducible to the reader.

Answer 1-3) We sincerely thank the referee for valuable suggestions and apologize for not providing a precise description. We revised this sentence as below:

The designed model catalysts were prepared by simply reducing WCl₆ and AgNO₃ with sodium borohydride (NaBH₄) on the BTO substrate. The tungsten (W) ions were successfully introduced to the lattice of BTO, and the resulting Ag nanoparticles were in situ formed on the

BTO surface during the reduction process (Fig. 2a). More details of the preparation process are supplied in the experimental section. (Revised Manuscript on pages 5 and 6)

Q1-4. Lines 159-162: To support the claim that a difference of 0.02 Å in the Ag-Ag bond is real, the authors must report the error in these measurements. This is especially important considering the bulk nature of XAS (where the data represents the average of all Ag-Ag bonds, not just those near the W-BTO) and the claim that a large number of electrons were transferred from Ag to the W-BTO to result in a shortened bond distance. At this point in the paper, this seems like an overly analyzed piece of data, and the more simple explanation that the Ag-Ag bonds are within error of each other, is more apparent to the reader when looking at the plots.

Answer 1-4) Thank you for your professional advice. And we completely agree with your opinions. Based on your suggestion, we re-fitted the EXAFS data and observed the disappearance of the 0.02 Å error (Figure 3h and Supplementary Figure 14b). Therefore, we revised the description as below:

The dominant peaks of Ag–Ag shown in Ag-BTO and Ag.W-BTO are nearly no difference, which is consistent with that of standard Ag foil (Fig. 3h and Supplementary Fig. 14b). (Revised Manuscript on page 9)

Figure 3. h Extended X-ray absorption fine structure (EXAFS) $k^3 \chi(k)$ Fourier-transform (FT)

spectra of Ag in R-space of catalysts with standard references, respectively. (Revised Manuscript on page 10)

Supplementary Figure 14. b Extended X-ray absorption fine structure (EXAFS) $k^3 \chi(k)$ Fourier-transform (FT) and fitting spectra of Ag in R-space of catalysts with standard references, respectively. (Revised Supporting Information on page 18)

Q1-5. Lines 192-195: as plotted, the N_2 physisorption data do not clearly show type IV isotherms. The plots have too much overlapping data on them to be readable. This data is not clearly presented. If more porosity is generated through the incorporation of the Ag and W on BTO, it should be evident in the TEM images. Comparing this N_2 phys data in light of the TEM images is needed to support the explanation of higher porosity.

Answer 1-5) We thank the reviewer for carefully reviewing our manuscript and apologize for this inappropriate information. We separately displayed these results (Supplementary Figure 17a-d) and we can more clearly notice the type IV isotherms with H_3 hysteresis loops of BTO based samples. In addition, we totally agree with that porosity is more easily and directly confirmed by the TEM images. Then we carefully studied TEM images but could not find clear pore structures due to thick structure of BTO. Interestingly, we identified some pore structures

and highlighted them with yellow circles in the SEM images (Supplementary Figure 10b). We revised relative figures and description are shown below:

Supplementary Figure 17. N₂ sorption isotherms. a BTO, b Ag-BTO, c W-BTO and d Ag.W-BTO. e Comparison N₂ sorption isotherms and f pore sizes of BTO, Ag-BTO, W-BTO, and Ag.W-BTO. (Revised Supporting Information on page 19)

Supplementary Figure 10. b SEM images of Ag.W-BTO nano-bulks. (Revised Supporting Information on page 15)

N₂ adsorption/desorption analyses were carried out to determine the effect of doping W and depositing Ag on the pore structures (Supplementary Fig. 17).⁴⁸ The presence of mesoporous structures is implied by type IV isotherms with H₃ hysteresis loops in all prepared catalysts.⁴⁹

The pore-structure distribution curves in Supplementary Fig. 17f are evidence of the dominant mesoporous and microporous structures of all samples. The specific surface areas (SSAs) of BTO, W-BTO, Ag-BTO, and Ag.W-BTO were 28.62, 44.68, 35.26, and 52.12 m² g⁻¹, respectively (additional parameters are shown in Supplementary Table 6). **The increased specific surface area (SSA) of Ag.W-BTO is likely attributed to the porous structures (Supplementary Fig. 10b).** (Revised Manuscript on page 12)

Q1-6. Lines 214-215: the statement “This phenomenon implies that *CO intermediates participated in producing DMM on the Ag.W-BTO surface under the synergistic effect of Ag and W.” is highly speculative, and not based on the data presented at this point in the manuscript. One could easily argue that CO production is greatly inhibited on the Ag.W-BTO catalyst in favor of CH₂O, which is why it generates high selectivity to DMM rather than gaseous products. The DRIFTS data presented later informs this statement, but and I understand that 4 PCET steps are needed to generate formaldehyde (the critical intermediate to form DMM) from CO₂. It may be that this statement appears too early in the manuscript, and the explanation is best left when the elementary reactions are presented in Lines 351-355. The authors should perform specific experiments to support this claim, such as feeding CO instead of CO₂ and monitoring catalytic performance and DRIFTS speciation. One would expect the rate of DMM production to be significantly higher if fewer PCET steps are needed to use CO instead of CO₂. if the same rate is observed, it may be that there is a different rate-limiting step, which would also be a great addition to this report.

Answer 1-6) Thank you very much for your kind suggestions and professional comments. We totally agree your points and newly added the additional experiments.

We deleted this sentence “This phenomenon implies that *CO intermediates participated in producing DMM on the Ag.W-BTO surface under the synergistic effect of Ag and W. ” in Lines 214-215. The relevant description was relocated after lines 351-355 and the final modifications in the Manuscript are shown at the end of this question.

Then, to more clearly explore the influence of *CO intermediate on the reaction, we check the

performance of DMM production and in-situ DRIFTS analysis by replacing CO₂ with CO on Ag.W-BTO under the same reaction conditions as suggested. A significant increase in DMM production (7008.35 μmol g⁻¹) was observed when CO and CH₃OH were utilized as reactants, compared to the case of CO₂ and CH₃OH (5702.49 μmol g⁻¹), which strongly implies the active involvement of *CO intermediates during the DMM synthesis process (Supplementary Figure 33). According to the reaction pathway that we proposed in the manuscript, as CO was used to replace CO₂ gas, *CH₂O intermediate can be formed from the easier conversion of CO, followed by the direct coupling of *CH₂O with CH₃OH-derived *CH₃O intermediates, leading to the production of DMM.

Therefore, for the in-situ DRIFTS spectrum, we first feed the CO and CH₃OH within 1h under the dark condition, and it is clearly showed *CO (2060 cm⁻¹) and *CH₃O (1155 cm⁻¹) intermediates in the 0 min, confirming successfully adsorbed CO and CH₃OH in the dark (Supplementary Figure 44). Subsequently, the characteristic peak of *CH₂O intermediate reduced by *CO located about 1260 cm⁻¹ appeared after UV-visible light irradiation, and finally the *CH₂O coupling with *CH₃O form DMM. The peak intensity of these intermediates (*CO, *CH₂O, and *CH₃O) displays a smaller increase, which may be due to the consumption of intermediates to synthesize DMM with a higher reaction rate.

Finally, for DFT calculations, we added which step is a rate-determine step and confirmed *CH₂O produced by CO₂ reduction (CO₂RR) rather than CH₃OH oxidation (MOR) during the DMM formation process. The calculated DMM formation free energies through two reaction paths (the origin line and purple line exhibited the MOR process and the other MOR coupling with the CO₂RR process, respectively) are shown in Figure 7c. In the third step, it is determined that the transition from *CH₃O to *CH₃O + *CO₂ with an energy barrier of -0.31 eV is more thermodynamically preferable compared to the oxidation of *CH₃O to *CH₂O with an energy barrier of -0.09 eV. This indicates that CH₃OH hardly tends to form *CH₂O intermediates, and the MOR coupling with the CO₂RR pathway is more favored in the system where CH₃OH and CO₂ coexist. Furthermore, three steps need energy input in MOR coupling with the CO₂RR pathway, including the conversion of *CO₂ to *COOH intermediate, the reduction of *CO to *CH₂O, and the coupling of *CH₂O with *CH₃O process. Especially the step of CO₂ to *COOH intermediates consumes the highest energy barrier, which is 0.53 eV and the rate-determine

step. Hence, by substituting CO for CO₂, two steps with a higher energy barrier during *CO₂ to *CO were eliminated, and the whole reaction energy input decreased, leading to an increased production of DMM.

Based on your suggestion and the above analysis, we revised the Manuscript and Supporting Information, shown below:

Supplementary Figure 33. The comparison of DMM production amount of CO + CH₃OH and CO₂ + CH₃OH. (Revised Supporting Information on page 29)

Supplementary Figure 44. In-situ DRIFTS spectroscopy characterization on the Ag₃W-BTO by flowing CO and CH₃OH gas under UV-visible irradiation at detailed reaction times (0, 10, 20, 30, 60, 120, 180, 240 min) after adoption 1 hour in the dark. (Revised Supporting Infor

mation on page 35)

Figure 7. DFT calculations. a Calculated CO₂ and CH₃OH adsorption energy on the different active sites (Ag, W, Ti, and O_v) of Ag.W-BTO, respectively. b Charge density difference and Bader charge analysis of CO₂ adsorbed on the Ag-BTO and Ag.W-BTO surface. c Free energy diagram of DMM production via different pathway. d Geometries of reaction intermediates involved in MOR coupling with CO₂RR pathways. (Revised Manuscript on page 24)

Moreover, to more precisely investigate the impact of *CO intermediates on the reaction, we checked DMM production by substituting CO₂ with CO on Ag.W-BTO under the same reaction conditions. The marked increase in DMM production (7008.35 μmol g⁻¹) observed when using CO and CH₃OH as reactants, compared to the utilization of CO₂ and CH₃OH (5702.49 μmol g⁻¹), strongly suggests the active participation of *CO intermediates in the synthesis process of DMM (Supplementary Fig. 33). (Revised Manuscript on page 15)

In addition, in-situ DRIFTS experiments were conducted by flowing CO and CH₃OH to explore whether CO participate in the reaction. The bands of *CO (2060 cm⁻¹) and *CH₃O (1155 cm⁻¹) intermediates are produced immediately after the adsorption of CO and CH₃OH for 1 hour in the dark (Supplementary Fig. 44). Subsequently, a band corresponding to the *CH₂O intermediate (around 1260 cm⁻¹) reduced by *CO, appeared after UV-visible light irradiation which is consistent with our performance result. The peak intensity of these intermediates (*CO, *CH₂O, and *CH₃O) exhibits a smaller increase during the reaction, possibly due to the consumption of these intermediates during the synthesis of DMM at a higher reaction rate. (Revised Manuscript on pages 21 and 22)

To further explore the formation pathway of DMM in the coexistence of CO₂ and CH₃OH, the free energy profile and reaction pathway of the DMM synthesis were compared on the Ag.W-BTO in different routes (MOR pathway and MOR coupling with CO₂RR (MOR + CO₂RR) pathway, **Fig. 7c**). The optimized intermediate structures corresponding to each reaction steps are displayed in **Fig. 7d** and Supplementary Fig. 48. The DMM formation on Ag.W-BTO through two pathways is triggered by the thermodynamical spontaneity from CH₃OH to *CH₃OH with an energy barrier of -0.98 eV. Then, *CH₃OH is further oxidated to *CH₃O with an energy barrier of -0.65 eV. Thereafter, it was found that the process from *CH₃O to *CH₃O + *CO₂ with an energy barrier of -0.31 eV is more thermodynamically favorable than the oxidation of *CH₃O to *CH₂O with an energy barrier of -0.09 eV, which suggests that *CH₃O tends to couple with *CO₂ intermediates rather than to form individual *CH₂O intermediates in CH₃OH and CO₂ coexist system. In addition, the process of the next step (*CH₂O + *CH₃O) in MOR pathway free energy barrier (0.98 eV) requires higher than *CH₃O + *COOH (0.53 eV) in MOR + CO₂RR route. That means direct oxidation of CH₃OH to DMM is unfavorable. Moreover, three steps require energy input in the MOR + CO₂RR pathway,

including the conversion of $^*\text{CO}_2$ to $^*\text{COOH}$ intermediates, the reduction of $^*\text{CO}$ to $^*\text{CH}_2\text{O}$, and the coupling of $^*\text{CH}_2\text{O}$ with $^*\text{CH}_3\text{O}$. Particularly, the conversion of CO_2 to $^*\text{COOH}$ intermediates consumes the highest energy of 0.53 eV and is considered the rate-determining step during the DMM formation process. (Revised Manuscript on pages 23)

Simultaneously, CO_2 molecules obtain e^- and combine with H^+ to produce COOH^* on the active Ag sites. Furthermore, $^*\text{COOH}$ is reduced to $^*\text{CO}$ and then changed to $^*\text{CH}_2\text{O}$ by a multi-step proton-coupled electron transfer (PCET) process. (Revised Manuscript on pages 25)

Q1-7. Finally, The ‘Discussion’ section is perhaps best noted as a summary, needing no header, and beginning with ‘In summary,...’

Answer 1-7) We thank the reviewer for the excellent suggestion and revised as blow:

In summary, a novel photo-redox system that can simultaneously couple solar-driven CO_2RR with MOR on Ag.W-BTO dual-functional catalysts to produce a value-added chemical (DMM) was proposed. The selectivity of DMM approached 92.08% on Ag.W-BTO and was accompanied by a record-high yield of $5702.49 \mu\text{mol g}^{-1}$ after 9 h UV-visible irradiation without sacrificial agents. Validation of the synergistically coupled multi-step PCET mechanism for DMM formation was achieved through systematic isotope labeling experiments, in-situ DRIFTS analysis, and DFT calculations. The Ag species were largely responsible for the facile CO_2 adsorption and reduction to $^*\text{CH}_2\text{O}$, while the W dopant promoted CH_3OH oxidation to $^*\text{CH}_3\text{O}$. The two obtained intermediates couple to synthesize DMM. This work provides a novel concept for green photochemical synthesis of high-value chemicals by coupling CO_2 reduction with another small molecular conversion and the fine design of photocatalysts. (Revised Manuscript on pages 25)

We truly thank Reviewer 1 for the insightful comments and kind suggestions.

Reviewer #2 (Remarks to the Author):

This work proposes a dual-functional catalyst system employing a silver and tungsten co-modified blue titanium dioxide catalyst (Ag.W-BTO) for the efficient generation of DMM through the synergistic coupling of CO₂ reduction and CH₃OH oxidation under mild reaction conditions. The selectivity of DMM reached 92.08% on Ag.W-BTO, accompanied by a yield of 5702.49 μmol g⁻¹ after 9 hours of UV-visible irradiation without the use of sacrificial agents. The author considers that the Ag and W species serve as catalysts for the conversion of CO₂ to *CH₂O and the oxidation of CH₃OH to *CH₃O, respectively. Nevertheless, this view lacks sufficient evidence and the content of the manuscript is incomplete. This work is not suitable for acceptance by Nature communications. The following concerns regarding this manuscript should be addressed:

Response: We are grateful for the time and effort Reviewer 2 has spent reviewing our manuscript. The review comments are very professional and helpful for further strengthening the manuscript. We have responded to your comments point-by-point and carefully revised the manuscript as you suggested.

Q2-1. It is well known that Ag exhibits localized surface plasmon resonance (LSPR) effect, which can enhance the surface temperature of a catalyst upon visible light irradiation. Given the high Ag content in the Ag.W-BTO catalyst, careful consideration must be given to the impact of elevated temperatures during the reaction process.

Answer Q2-1) We thank the referee for carefully reviewing our manuscript and bringing valuable suggestions. We agree that the presentation in the original version did not explain this issue. As the definition of the LSPR effect, when the incident photon's frequency aligns with the oscillation frequency of the surface free electron in a noble metal like Ag, the localized surface plasmon resonance (LSPR) effect will occur between them (*Nano Lett.* 2018, 18, 6, 3752–3758). Then, the LSPR effect Ag in Ag.W-BTO was additionally confirmed by the increased surface temperature under irradiation (Supplementary Figure 35). The temperature of the Ag.W-BTO surface increased from about 21 to 31°C within 10 minutes and remained unchanged thereafter. Since we keep the reaction temperature at 100°C during the reaction, the surface temperature of all catalysts will rise to 100°C. The LSPR does not have a great impact

on the surface temperature. The relative changes in the revised manuscript are shown as follows:

Supplementary Figure 35. Localized surface plasmon resonance (LSPR) effect of Ag.W-BTO. **a** Temperature vs time curves of Ag.W-BTO within 60 min. **b** Representative infrared thermal image. (Revised Supporting Information on page 30)

The surface temperatures of the Ag.W-BTO were assessed using an infrared thermal imager (Teledyne FLIR, FLIR TG165). 25 mg of photocatalyst was positioned at a distance of 40 cm from the 300 W Xenon lamp cap (wavelength, 320-780 nm). (Revised Supporting Information on page 4)

In addition, when the frequency of the incident photons matches the oscillation frequency of the surface free electrons in Ag nanoparticles, the localized surface plasmon resonance (LSPR) effect occurs.⁵¹ As for exploring the LSPR effect of deposition of Ag on the Ag.W-BTO sample, the surface temperatures of the Ag.W-BTO were assessed using an infrared thermal imager. Analysis of the temperature versus time curves for the Ag.W-BTO revealed a gradual increase from about 21 to 31 °C in 10 minutes and stabilization of about 31 °C within 30 minutes (Supplementary Fig. 35). This stabilization phenomenon is attributed to the equilibrium reached between the heat dissipation of the sample and its surrounding medium.⁴⁷ (Revised Manuscript on pages 15 and 16)

51. Biggins JS, Yazdi S, Ringe E. Magnesium nanoparticle plasmonics. *Nano Lett.* **18**, 3752-3758 (2018).

47. Guo Y, *et al.* Efficient interfacial electron transfer induced by hollow-structured ZnIn₂S₄ for extending hot electron lifetimes. *Energy Environ. Sci.* **16**, 3462-3473 (2023).

Q2-2. Figure 3a shows that there are two characteristic peaks, Ti³⁺ and Ti⁴⁺, and the content of Ti³⁺ accounts for the vast majority. In reported works, the oxidation state of titanium in the blue TiO₂ is predominantly +4, so the XPS spectra of Ti 2p need to be reconsidered (*Advanced Functional Materials* 2017, 27: 1700856; *ACS Applied Materials & Interfaces*, 2016, 8(40): 26851-26859; *Advanced Energy Materials*, 2020, 10(10): 1903107). Additionally, EPR characterization is required to verify the presence of both Ti³⁺ and oxygen vacancies.

Answer Q2-2) We appreciate the reviewer for the careful review and insightful comment and apologize for the error encountered during the fitting of the XPS data of Ti 2p. The equipment expert refitted the Ti 2p (Figure 3) and newly conducted the EPR (Supplementary Figure 13) to confirm the existence of Ti³⁺ and oxygen vacancies in our BTO samples. Both BTO and Ag.W-BTO exhibit distinct EPR signals from paramagnetic Ti³⁺ (g=1.96) and O_v (g=2.003), whereas TiO₂ (P₂₅) shows a negligible EPR signal. The modifications in the manuscript are as follows:

Figure 3. a Ti 2p XPS of BTO, W-BTO, Ag-BTO, and Ag.W-BTO, respectively. (Revised

Supplementary Figure 13. Electron paramagnetic resonance (EPR) spectra of TiO₂ (P₂₅), BTO and Ag.W-BTO. (Revised Supporting Information on page 17)

The chemical states of the catalysts were determined by X-ray photoelectron spectroscopy (XPS). In the Ti 2p spectra (**Fig. 3a**), four characteristic peaks appear near 464.30, 463.31, 458.64, and 457.82 eV, corresponding to Ti 2p_{1/2} Ti⁴⁺, Ti 2p_{1/2} Ti³⁺, Ti 2p_{3/2} Ti⁴⁺, and Ti 2p_{3/2} Ti³⁺, respectively.^{28, 29} The presence of peaks corresponding to Ti³⁺ confirms that the synthesized catalyst contains oxygen vacancies (O_v). (Revised Manuscript on page 7)

28. Zhang S, *et al.* Laser-assisted rapid synthesis of anatase/rutile TiO₂ heterojunction with Function-specified micro-zones for the effective photo-oxidation of sulfamethoxazole. *Chem. Eng. J.* **453**, 139702 (2023).

29. Zhang Y, *et al.* Ti³⁺ self-doped blue TiO₂(B) single-crystalline nanorods for efficient solar-driven photocatalytic performance. *ACS Appl. Mater. Inter.* **8**, 26851-26859 (2016).

The presence of Ti^{3+} species and O_v were also confirmed by the Electron paramagnetic resonance (EPR) spectra (Supplementary Fig. 13). Both BTO and Ag.W-BTO show distinctive EPR signals of paramagnetic Ti^{3+} ($g=1.96$) and O_v ($g=2.003$) while TiO_2 (P_{25}) showed a negligible EPR signal.³⁷⁻³⁹ This result implies that BTO with O_v was successfully synthesized after reduction with Li-EDA. The peak intensities of Ti^{3+} and O_v in Ag.W-BTO are higher than those of BTO, meaning the doping of W may slightly replace the Ti^{3+} or cover the O_v in Ag.W-BTO samples. (Revised Manuscript on page 8)

37. Lee J, *et al.* Phase-selective active sites on ordered/disordered titanium dioxide enable exceptional photocatalytic ammonia synthesis. *Chem. Sci.* **12**, 9619-9629 (2021).
38. Hao Z, *et al.* Oxygen-deficient blue TiO_2 for ultrastable and fast lithium storage. *Adv. Energy Mater.* **10**, 1903107 (2020).
39. Zhang Y, Ding Z, Foster CW, Banks CE, Qiu X, Ji X. Oxygen vacancies evoked blue $TiO_2(B)$ nanobelts with efficiency enhancement in sodium storage behaviors. *Adv. Funct. Mater.* **27**, 1700856 (2017).

Q2-3. The authors provided a detailed characterization of Ti^{3+} in blue TiO_2 . However, its correlation with the reaction remains unestablished. Further clarification is requested.

Answer Q2-3) We appreciate the reviewer for raising the valuable suggestion. The Blue TiO_2 (BTO) with Ti^{3+} states were obtained from the commercial TiO_2 (P_{25}) by treatment of the Lithium-in-Ethylenediamine solution (Li-EDA). As a strong reductant, Li-EDA was already evidenced that only selectively reduced the rutile phase of P_{25} , leaving crystalline anatase phase BTO, which has Ti^{3+} and O vacancies (O_v) (*Materials Today*, 35 (2020) 25-33). In this work, compared with TiO_2 (3.2 eV), the existence of Ti^{3+} defect sites with multiple internal energy levels results in a narrow band gap of BTO (about 2.91 eV), leading to **high visible light absorption** (*Materials Today Chemistry*, 24 (2022) 100827). We added the explanation in the band gap structure section in the revised manuscript.

In addition, DFT calculations were additionally conducted on the new catalyst structures with

Ti³⁺ and O_v to further explore the reaction mechanism. The results show that the adsorption energy of CO₂ and CH₃OH is relatively negative on Ag and W than that of Ti³⁺ and O_v, respectively (Figure 7a). Therefore, the dominant active sites of CO₂RR and CH₃OH oxidation are W and Ag, respectively. The Ti³⁺ and O vacancies in catalysts mainly play a role in **narrowing the band gap and improving visible light absorption.**

All the BTO-related samples showed a narrower band structure than pure TiO₂ (P₂₅, 3.2 eV), which was attributed to the existence of Ti³⁺ defect sites with multiple internal energy band gaps after Lithium-in-Ethylenediamine solution (Li-EDA) treatment.^{23,45} (Revised Manuscript on page 11)

23. Nguyen CT, *et al.* Highly efficient nanostructured metal-decorated hybrid semiconductors for solar conversion of CO₂ with almost complete CO selectivity. *Mater. Today* **35**, 25-33 (2020).
45. Kumar D, *et al.* Nanocavity-assisted single-crystalline Ti³⁺ self-doped blue TiO₂ (B) as efficient cocatalyst for high selective CO₂ photoreduction of g-C₃N₄. *Materials Today Chemistry* **24**, 100827 (2022).

The adsorption energies of CO₂ and CH₃OH on distinct active sites (W, Ag, Ti, and O_v) were analyzed to confirm the reduction and oxidation sites on the Ag.W-BTO sample (Supplementary Fig. 46 and 47). As depicted in **Fig. 7a**, all the active sites (W, Ag, Ti, and O_v) displayed negative values for CO₂ adsorption energies, indicating a higher tendency for CO₂ adsorption across these sites. The most negative adsorption energy for CO₂ (−2.413 eV) signifies a preference for CO₂ adsorption on the Ag sites within the Ag.W-BTO sample. This highlights the heightened propensity of the Ag sites within Ag.W-BTO for CO₂RR. The most negative adsorption energy (−1.375 eV) observed for W species during CH₃OH adsorption suggests their higher inclination towards undergoing MOR. (Revised Manuscript on page 22)

Figure 7. DFT calculations. **a** Calculated CO₂ and CH₃OH adsorption energy on the different active sites (Ag, W, Ti, and O_v) of Ag.W-BTO, respectively. **b** Charge density difference and Bader charge analysis of CO₂ adsorbed on the Ag-BTO and Ag.W-BTO surface. **c** Free energy diagram of DMM production via different pathway. **d** Geometries of reaction intermediates involved in MOR coupling with CO₂RR pathways. (Revised Manuscript on page 24)

Q2-4. The article mentioned that “W⁶⁺ can be gradually transformed to unsaturated W⁵⁺”,

thus relevant characterization should be provided as evidence.

Answer Q2-4) We appreciate the reviewer for the careful review and insightful comment. To confirm this point, we newly measured the W 4f XPS of Ag.W-BTO before and after the reaction. The result is shown in Supplementary Figure 29c below. Compared with Ag.W-BTO, the area ratio of W^{6+} and W^{5+} decreased. It means that W^{6+} can be gradually transformed into unsaturated W^{5+} . However, there are no obvious changes for other elements (Ti, Ag, and O), which can prove that our catalyst is rather stable during the reaction. We added Supplementary Figure 29 and revised the Manuscript as below:

Supplementary Figure 29. XPS of Ag.W-BTO before and after the stability. a Ti 2p, b Ag 3d, c W 4f, and d O 1s. (Revised Supporting Information on page 27)

To prevent photo-generated carriers from recombining under light irradiation, W^{6+} , as a donor just below the conduction band, can be gradually transformed to unsaturated W^{5+} , which works in W^{6+}/W^{5+} pairs to improve performance (Supplementary Fig. 29c).³³ (Revised Manuscript on page 8)

33. Huang M, *et al.* Highly Selective Photocatalytic Aerobic Oxidation of Methane to Oxygenates with Water over W-doped TiO_2 . *ChemSusChem* **15**, e202200548 (2022).

Q2-5. Typically, in XAS analysis, metal K-edge compares near-edge absorption positions, while metal L_3 -edge requires comparison based on white line peak positions. Therefore, it is necessary to revise descriptions related to W L_3 -edge accordingly. The results in Figure g-i should be obtained by the Fourier transform of EXAFS spectrum, but the processing results seem to be wrong. In the processing process, attention should be paid to adjusting parameters, removing bad points and K-space values. All R-space data needs to be reprocessed and discussed. In addition, the Ti-O, Ag-O, W-O, and W-O-Ti bonds mentioned in the paper need to provide fitting data and charts to verify whether the fitting is reasonable.

Answer 2-5) We sincerely thank the referee for valuable suggestions. Based on your suggestions, we revised descriptions related to W L_3 -edge accordingly, re-plotted the EXAFS spectrum of R-space in Ti, Ag, and W (Figure 3g-i) by adjusting parameters, and removed bad points and K-space values. Finally, we newly fitted all the EXAFS data, including references and samples (Supplementary Figure 14). The content is shown below:

Figure 3. **d-f** Normalized Ti K-edge, Ag K-edge, W L_3 -edge, and their difference X-ray absorption spectra (XANES) of catalysts with standard references. **g-i** Extended X-ray absorption fine structure (EXAFS) $k^3 \chi(k)$ Fourier-transform (FT) spectra of Ti, Ag, and W in R-space of catalysts with standard references, respectively. (Revised Manuscript on page 10)

Supplementary Figure 14. Fitting EXAFS spectrum. **a-c** Extended X-ray absorption fine structure (EXAFS) $k^3 \chi(k)$ Fourier-transform (FT) and fitting spectra of Ti, Ag, and W in R-space of catalysts with standard references, respectively. (Revised Supporting Information on page 18)

Supplementary Table 3. EXAFS fitting parameters at the Ti K-edge for Anatase TiO₂, BTO, Ag-BTO, W-BTO, and Ag.W-BTO

Sample	Bond	$CN = N \times \frac{S_0^2}{S_0}$	R (Å)	σ^2 (Å ⁻²)	ΔE_0 (eV)	R-factor
Anatase TiO ₂	Ti-O	6	1.956	0.0091	-1.98	0.0511
	Ti-Ti (1)	4	3.05	0.0089	-9.57	
	Ti-Ti (2)	4	3.842	0.0077	-16.15	
BTO	Ti-O	2.4	1.91	0.0025	-9.55	0.0576
	Ti-O	5.4	1.933	0.0083	-4.58	
Ag-BTO	Ti-Ti (1)	1.92	3.01	0.0036	-11.53	0.0246
	Ti-Ti (2)	4	3.565	0.0156	4.8	
	Ti-O	5.22	1.928	0.0081	-5.1	
W-BTO	Ti-Ti (1)	1.8	3.003	0.0029	-12.2	0.0186
	Ti-Ti (2)	4	3.567	0.0155	4.85	
	Ti-O	5.57	1.954	0.0088	-3.6	
Ag.W-BTO	Ti-Ti (1)	2.2	3.012	0.0040	-13	0.0178
	Ti-Ti (2)	4	3.597	0.0143	5.27	

CN: coordination numbers; R: bond distance; σ^2 : Debye-Waller factors; ΔE^0 : the inner potential correction; R-factor: goodness of fit.

Supplementary Table 4. EXAFS fitting parameters at the Ag K-edge for Ag foil, Ag₂O, Ag-BTO, and Ag.W-BTO

Sample	Bond	CN = $N \times S_0^2$	R (Å)	σ^2 (Å ⁻²)	ΔE_0 (eV)	R-factor
Ag foil	Ag–Ag	12	2.863	0.0081	–1	0.0007
Ag ₂ O	Ag–O	2	2.062	0.0057	2.12	0.0495
	Ag–Ag	12	3.151	0.0416	–8.08	
Ag-BTO	Ag–Ag	10.46	2.860	0.0081	0.63	0.0010
Ag.W-BTO	Ag–Ag	8.41	2.857	0.0080	–1.48	0.0022

CN: coordination numbers; R: bond distance; σ^2 : Debye-Waller factors; ΔE_0 : the inner potential correction; R-factor: goodness of fit.

Supplementary Table 5. EXAFS fitting parameters at the W L₃-edge for W foil, WO₃, W-BTO, and Ag.W-BTO

Sample	Bond	CN = $N \times S_0^2$	R (Å)	σ^2 (Å ⁻²)	ΔE_0 (eV)	R-factor
W foil	W–W (1)	8	2.756	0.004	7.93	0.0199
	W–W (2)	6	3.177	0.007	9.84	
WO ₃	W–O (1)	4	1.781	0.0072	1.1	0.0118
	W–O (2)	2	2.097	0.0048	3.04	
W-BTO	W–O (1)	3.06	1.887	0.0034	14.9	0.0391
	W–O (2)	2.77	2.245	0.0054	15.69	
Ag.W-BTO	W–O (1)	3.88	1.77	0.009	–8.48	0.047
	W–O (2)	1	2.098	0.0005	–1.47	

CN: coordination numbers; R: bond distance; σ^2 : Debye-Waller factors; ΔE_0 : the inner potential correction; R-factor: goodness of fit.

(Above revised Supporting Information on pages 40 and 41)

From the W L₃-edge absorption spectra (**Fig. 3f**), the white line peak positions of W in W-BTO and Ag.W-BTO are higher than those of W foil and WO₃ attributed to the strong electron adsorption properties of O in W-BTO and Ag.W-BTO, which agrees with the discussions of XPS spectra. The distinctive coordination environments of Ti, Ag, and W were investigated by Fourier-transformed (FT) extended X-ray absorption fine structure (EXAFS) spectra and fitting data (Supplementary Fig. 14 and Supplementary Tables 3–5). In the Ti FT-EXAFS spectra, Ti–O bonds can be seen at approximately 1.63, 1.65, and 1.62 Å for Ag-BTO, W-BTO, and Ag.W-BTO, respectively (**Fig. 3g** and **Supplementary Fig. 14a**). The Ti–O bonds of these samples are extended compared to those of pure BTO (located at approximately 1.52 Å) due to the introduction of W and Ag species that can stretch the Ti–O bond length and synergistically tune the local coordination environment of Ti. The dominant peaks of Ag–Ag shown in Ag-BTO and Ag.W-BTO are nearly no difference which is consistent with that of standard Ag foil (**Fig. 3h** and **Supplementary Fig. 14b**). For k³ χ(k)-FT of the W L₃-edge EXAFS spectra (**Fig. 3i** and **Supplementary Fig. 14c**), the first coordination shell (W–O peak at 0.7–1.9 Å) was composed primarily of single-scattering (SS) O atoms, accompanied by multiple-scattering (MS) contributions from the second shell (peaks at 2–3 Å).^{40, 41} The interatomic distances of W–O in W-BTO (approximately 1.47 Å) and Ag.W-BTO (approximately 1.29 Å) are different from those of standard WO₃ (1.36 Å) as W atoms are doped into the BTO lattice rather than the isolated oxide species. Overall, the electronic interaction of Ag and W dual-active sites is expected to be capable of guaranteeing the photo-redox synergic reaction for higher efficiency. (**Revised Manuscript on page 9**)

Q2-6. In Figure 4f, the m/z value for CH₃¹⁸O was indicated as 33 which is inconsistent with the position shown as 32 in the figure.

Answer 2-6) We thank the reviewer for carefully reviewing our manuscript and sincerely apologize for the erroneous labeling of ions in the original m/z value. We re-conducted the ¹⁸O-labeled CH₃OH + CO₂ labeling experiment. Specifically, the m/z value of [CH₃¹⁸O]⁺ ion is 33, which is the same as we checked (Figure 4f), but we mistakenly marked it as 32. Thus, we have

rectified this error by replacing the number 32 with 33 in Figure 4f, as indicated below:

Figure 4. f GC-MS results of I) isotope labeled $\text{CH}_3^{18}\text{OH} + \text{CO}_2$ and II) isotope non-labeled $\text{CH}_3^{16}\text{OH} + \text{CO}_2$. (Revised Manuscript on page 13)

Q2-7. The author needs to explain why $^*\text{CH}_3\text{O}$ exists at 0 minutes (no light), as shown in Figure 6c.

Answer 2-7) We sincerely thank the referee for valuable suggestions. Before checking the in-situ DRIFTS, high-purity CO_2 gas (99.99%) bubbling with CH_3OH was flowed in the reaction cell for adsorbing on the catalyst for 1h under dark conditions. Hereafter, the CO_2 reduction reaction coupling with the CH_3OH oxidation reaction was initiated. When the UV-visible light was irradiated (320-780 nm), the in-situ DRIFTS spectra were started to collect at specific reaction times (0, 10, 20, 30, 60, 120, 180, 240 min). With the progression of reaction time, the intensity of the $^*\text{CH}_3\text{O}$ peak steadily rises, signifying the generation of $^*\text{CH}_3\text{O}$ intermediates throughout the reaction. DFT calculations prove that the decomposition of CH_3OH into $^*\text{CH}_3\text{O}$ is a thermodynamic spontaneous reaction, **so this process is very rapid, and a small amount of $^*\text{CH}_3\text{O}$ can be detected even at 0 minutes.** It also confirming CH_3OH is successfully

adsorption onto the Ag.W-BTO surface. We revised the relative contents in the revised Manuscript as below:

Due to the continuous filling of CO₂ gas with CH₃OH, a high stretching vibration bond at 2300 cm⁻¹ belonging to CO₂ adsorption is evident (Supplementary Fig. 41).⁵⁹ The band located in 1155 cm⁻¹ corresponds to the spreading -CH₃O group of CH₃OH, which confirms that CH₃OH is successfully adsorbed onto the Ag.W-BTO surface (Fig. 6b and c).⁶⁰ (Revised Manuscript on page 20)

Q2-8. Theoretical calculations alone cannot offer sufficient evidence regarding the active sites of Ag and W species in CO₂ conversion to *CH₂O and CH₃OH oxidation to *CH₃O. Therefore, it is imperative to provide compelling evidence.

Answer 2-8) We sincerely thank the referee for carefully reviewing our manuscript and providing constructive suggestions for improving its quality. We only calculated the adsorption energies of CO₂ and CH₃OH on Ag, W, and Ti, respectively in the original Manuscript, which could not fully explain this point. However, we newly conducted a controlled in-situ DRIFT of flowing CO₂ and CH₃OH on Ag-BTO and W-BTO, respectively this version. The results identified that the Ag and W species mainly contribute to the conversion of CO₂ to *CH₂O and the oxidation of CH₃OH to *CH₃O, respectively. Moreover, the controlled experiments and band structure analysis of Ag.W-BTO can also be used as a supplementary explanation. Finally, we added DFT calculations and theoretically confirmed that *CH₂O was produced by CO₂ reduction (CO₂RR) rather than CH₃OH oxidation (MOR) in the DMM formation process. We explain all of these point by point as below:

First, we newly add and introduce the result of in-situ DRIFTS, which provided more reliable evidence:

In the in-situ DRIFTS reaction of CO₂ and H₂O on the surface of Ag-BTO, we observed the generation of CO₂RR intermediates such as *CO, *COOH, and *CH₂O (Supplementary Figure 42a). However, in the case of W-BTO, these three intermediates, especially the *CH₂O intermediate, were not produced with increasing time during the reaction (Supplementary Figure 42b). This suggests that Ag is more sensitive to the CO₂RR process and tends to produce *CH₂O intermediates.

For the in-situ DRIFTS spectrum of flowing CH₃OH and Ar on the W-BTO surface of the W-BTO catalyst (Supplementary Figure 43a), a noticeable increase can be found for the peak of 1155 cm⁻¹ corresponding to the *CH₃O intermediate. However, no obvious intensity enhancement of the *CH₃O peak was detected on the Ag-BTO surface when introducing CH₃OH and Ar (Supplementary Figure 43b). This indicates that W species mainly contribute to promoting the oxidation of CH₃OH to *CH₃O intermediates.

Supplementary Figure 42. The comparison of in-situ DRIFTS spectra by flowing C O₂ + H₂O. a Ag-BTO with enlarged view of shaded area. b W-BTO under UV-visible light irradiation in different reaction times. (Revised Supporting Information on page 34)

Supplementary Figure 43. The comparison of in-situ DRIFTS spectra by flowing CH₃OH + Ar. a W-BTO and b Ag-BTO under UV-visible light irradiation in different reaction times. (Revised Supporting Information on page 34)

Then, the supporting evidence was shown in controlled experiments and band gap structure:

In the controlled experiment, the amount of CO₂RR products (like CO, CH₄, and CH₂O) on Ag-BTO was significantly larger than those on W-BTO, which indicates that Ag nanoparticles prefer to be the CO₂RR active sites (Figure 4b). Furthermore, the band gap structure showed that the valence band of W-BTO is more positive than BTO, which means doping of W is advantageous for promoting oxidation reactions, implying that W species can act as CH₃OH oxidation sites (Supplementary Figure 15). Due to the formation of Mott-Schottky heterojunctions on Ag.W-BTO, it is observed that electrons were excited and tended to flow toward the Ag nanoparticles, facilitating the in-situ CO₂RR (Supplementary Figure 16).

Supplementary Figure 15. Band gap structures. a UV-vis of BTO, W-BTO, Ag-BTO, and Ag.W-BTO. **b** estimation of band gap, **c** UPS and **d** band gap structure of BTO, W-BTO. (Revised Supporting Information on page 18)

Supplementary Figure 16. Band gap structures. a Ag-BTO and **b** Ag.W-BTO. (Revised Supporting Information on page 19)

Finally, the theoretical explanations were provided by DFT calculations:

The calculated DMM formation free energies of the two reaction paths (origin line and purple line exhibited the pure MOR process and the MOR and CO₂RR coupling process (MOR + CO₂RR), respectively) are shown in Figure 7c. First, the DMM synthesis through these two pathways on Ag.W-BTO is triggered by the thermodynamically spontaneous process from CH₃OH to *CH₃OH with the free energy barrier of -0.98 eV. Then, *CH₃OH is further oxidated to *CH₃O with a free energy barrier of -0.65 eV. In the third step, it determined that the coupling of *CH₃O + *CO₂ with an energy barrier of -0.31 eV is thermodynamically preferable compared to the oxidation of *CH₃O to *CH₂O with an energy barrier of -0.09 eV. This indicates that CH₃OH hardly tends to form *CH₂O intermediates, and the MOR coupling with the CO₂RR pathway is much more favorable when CH₃OH and CO₂ coexist on Ag.W-BTO surface.

Figure 7. DFT calculations. **a** Calculated CO₂ and CH₃OH adsorption energy on the different active sites (Ag, W, Ti, and O_v) of Ag.W-BTO, respectively. **b** Charge density difference and Bader charge analysis of CO₂ adsorbed on the Ag-BTO and Ag.W-BTO surface. **c** Free energy diagram of DMM production via different pathway. **d** Geometries of reaction intermediates involved in MOR coupling with CO₂RR pathways. (Revised Manuscript on page 24)

Based on the above analysis, we revised the manuscript as blow:

Furthermore, to elucidate the preference of CO₂RR and MOR in generating *CH₂O over *CH₃O intermediates on Ag and W species, respectively, controlled in-situ DRIFTS experiments were conducted by flowing CO₂ with H₂O (CO₂ + H₂O) and CH₃OH with Ar (CH₃OH + Ar) on Ag-BTO and W-BTO catalysts, respectively. During the in-situ DRIFTS reaction of flowing CO₂ and H₂O on the surface of Ag-BTO, the emergence of CO₂RR intermediates such as *CO, *COOH, and notably *CH₂O which are located about 2060, 1716, and 1260 cm⁻¹ was observed, suggesting Ag with sensitivity to CO₂RR and its tendency to produce *CH₂O intermediates (Supplementary Fig. 42a). Conversely, in the case of W-BTO, the production of these three intermediates, especially *CH₂O, did not increase with reaction time, indicating a lack of sensitivity towards *CH₂O generation (Supplementary Fig. 42b). This suggests that Ag is more inclined to produce *CH₂O intermediates. In the in-situ DRIFTS spectrum of flowing CH₃OH and Ar on the surface of the W-BTO catalyst, a significant increase in the peak at 1155 cm⁻¹ corresponding to the *CH₃O intermediate was observed (Supplementary Fig. 43a). However, no notable enhancement in the intensity of the *CH₃O peak was detected on the Ag-BTO surface when CH₃OH and Ar were introduced (Supplementary Fig. 43b). This indicates that W species are more effective in promoting the oxidation of CH₃OH to *CH₃O intermediates. (Revised Manuscript on page 21)

The amount of CO₂RR products (CO, CH₄, and CH₂O) on Ag-BTO was significantly greater than on W-BTO, which indicates that Ag prefers to be the active reduction site. (Revised Manuscript on page 14)

Therefore, band gap structures of BTO and Ag.W-BTO are shown in Supplementary Fig. 15d. The VB of W-BTO (1.44 V) exhibited higher positivity than BTO (1.36 V), imparting that doping W species theoretically facilitates MOR performance. (Revised Manuscript on pages 11 and 12)

To further explore the formation pathway of DMM in the coexistence of CO₂ and CH₃OH, the free energy profile and reaction pathway of the DMM synthesis were compared on the Ag.W-BTO in different routes (MOR pathway and MOR coupling with CO₂RR (MOR + CO₂RR) pathway, **Fig. 7c**). The optimized intermediate structures corresponding to each reaction steps are displayed in **Fig. 7d** and Supplementary Fig. 48. The DMM formation on Ag.W-BTO

through two pathways is triggered by the thermodynamical spontaneity from CH₃OH to *CH₃OH with an energy barrier of -0.98 eV. Then, *CH₃OH is further oxidated to *CH₃O with an energy barrier of -0.65 eV. Thereafter, it was found that the process from *CH₃O to *CH₃O + *CO₂ with an energy barrier of -0.31 eV is more thermodynamically favorable than the oxidation of *CH₃O to *CH₂O with an energy barrier of -0.09 eV, which suggests that *CH₃O tends to couple with *CO₂ intermediates rather than to form individual *CH₂O intermediates in CH₃OH and CO₂ coexist system. In addition, the process of the next step (*CH₂O + *CH₃O) in MOR pathway free energy barrier (0.98 eV) requires higher than *CH₃O + *COOH (0.53 eV) in MOR + CO₂RR route. That means direct oxidation of CH₃OH to DMM is unfavorable. Moreover, three steps require energy input in the MOR + CO₂RR pathway, including the conversion of *CO₂ to *COOH intermediates, the reduction of *CO to *CH₂O, and the coupling of *CH₂O with *CH₃O. Particularly, the conversion of CO₂ to *COOH intermediates consumes the highest energy of 0.53 eV and is considered the rate-determining step during the DMM formation process. (Revised Manuscript on page 23)

Q2-9. Do Ag and W have an equivalent impact on other substrates?

Answer 2-9) We appreciate and are thankful for your insightful questions. Ag and W are active sites for CO₂RR and MOR, respectively. We used the same synthesis method as Ag.W-BTO to prepare Ag.W-TiO₂ by replacing BTO with commercial TiO₂ (P₂₅). TiO₂ (P₂₅) with minimal oxygen vacancies (O_v), enabling comparison with Ag.W-BTO to examine whether O_v serves as an active site for the reaction. In comparison to Ag.W-BTO, the yield of DMM on TiO₂ and Ag.W-TiO₂ is approximately 0 and 1688.35 μmol g⁻¹, respectively (Supplementary Figure 34). The presence of DMM on Ag.W-TiO₂ (without O_v) also suggests that O_v may not be the primary reactive site. However, the lower yield of DMM on Ag.W-TiO₂ compared to Ag.W-BTO can be attributed to the weaker absorption of visible light, low specific surface area, and high electron and hole recombination rate of the TiO₂ substrate, as reported in our previous work (*Materials Today*, 35 (2020) 25-33, *Chemical Science*, 12 (2021), 9619–9629).

Supplementary Figure 34. The comparison of DMM production amount on TiO₂, BTO, Ag.W-TiO₂ and Ag.W-BTO, respectively. (Revised Supporting Information on page 30)

Additionally, commercial TiO₂ (P₂₅) was substituted for BTO in Ag.W-BTO using the same NaBH₄ reduction method to assess their catalytic activity. TiO₂ was chosen for its minimal oxygen vacancies (O_v), allowing for comparison with Ag.W-BTO to assess whether the O_v are active sites in the reaction. DMM yields on TiO₂ and Ag.W-TiO₂ were found to be approximately 0 and 1688.35 $\mu\text{mol g}^{-1}$, respectively (Supplementary Fig. 34). The presence of DMM on Ag.W-TiO₂ (without O_v) suggests that O_v may not be the primary reactive sites. However, the lower DMM yield on Ag.W-TiO₂ compared to Ag.W-BTO can be attributed to the weaker absorption of visible light, low specific surface area, and high electron and hole recombination rate of the TiO₂ substrate.^{23,37} (Revised Manuscript on pages 15 and 16)

23. Nguyen CT, *et al.* Highly efficient nanostructured metal-decorated hybrid semiconductors for solar conversion of CO₂ with almost complete CO selectivity. *Mater. Today* **35**, 25-33 (2020).
37. Lee J, *et al.* Phase-selective active sites on ordered/disordered titanium dioxide enable exceptional photocatalytic ammonia synthesis. *Chem. Sci.* **12**, 9619-9629 (2021).

We truly thank Reviewer 2 for the insightful comments and kind suggestions.

Reviewer #3 (Remarks to the Author):

The manuscript " Photocatalytic coupling of CO₂ reduction and CH₃OH oxidation for over 90 % selective dimethoxymethane production" presents a novel method or selective dimethoxymethane (DMM) production by photocatalytic coupling of CO₂ reduction and CH₃OH oxidation. The photo-redox catalysis system is based on the a silver and tungsten co-modified blue titanium dioxide (Ag.W-BTO) photocatalyst. The research includes comprehensive investigation of the Ag.W-BTO structure, electron and hole transfer behaviors, and a systematic exploration of the photocatalytic activity, selectivity, and mechanisms involved in DMM production. The content of the manuscript is generally systematic and logical, and the elucidation of the mechanism makes sense. This manuscript can be further considered by Nat. Commun. subject to a revision, along the lines suggested hereafter:

Response: We deeply appreciate your positive feedback and insightful suggestions for enhancing our manuscript. We have meticulously addressed each of your comments and diligently revised the manuscript accordingly.

Q3-1. The fitting of the XPS spectra in your study appears to be inaccurate, particularly in the representation of peak widths at half height for the same species. Notably, in Figure 3a, there are inconsistencies in the widths of Ti³⁺ peaks. It is crucial to maintain uniformity in peak widths for the same species to ensure accuracy in spectral analysis. I recommend a thorough re-fitting of your XPS spectra to address these discrepancies and improve the reliability of your findings.

Answer 3-1) We deeply appreciate your valuable recommendations and sincerely apologize for the inaccuracy in our fitting method of fitting XPS data. We refitted all XPS spectra in Figure 3a-c and Supplementary Figure 12 as below:

Figure 3. a-c Ti 2p, Ag 3d, and W 4f XPS of BTO, W-BTO, Ag-BTO, and Ag.W-BTO, respectively. (Revised Manuscript on page 10)

Supplementary Figure 12. XPS spectra. a O 1s XPS of BTO, W-BTO, Ag-BTO, and Ag.W-BTO, respectively. **b** B 1s XPS of Ag.W-BTO. (Revised Supporting Information on page 17)

The chemical states of the catalysts were determined by X-ray photoelectron spectroscopy (XPS). In the Ti 2p spectra (**Fig. 3a**), four characteristic peaks appear near 464.30, 463.31, 458.64, and 457.82 eV, corresponding to Ti 2p_{1/2} Ti⁴⁺, Ti 2p_{1/2} Ti³⁺, Ti 2p_{3/2} Ti⁴⁺, and Ti 2p_{3/2}

Ti³⁺, respectively.^{28, 29} The presence of peaks corresponding to Ti³⁺ confirms that the synthesized catalyst contains oxygen vacancies (O_v). Compared with BTO and Ag-BTO, all W-BTO and Ag.W-BTO peaks displayed a positive shift. This phenomenon is primarily caused by W species with a lower electron cloud density and a strong electron affinity that can absorb electrons from BTO, resulting in the formation of a stable structure.³⁰ The Ag peaks centered at 372.60 eV and 366.70 eV can be ascribed to metallic Ag (**Fig. 3b**). The higher shift of the Ag binding energy in Ag.W-BTO is attributed to strong heterogeneous interaction and electron transfer between Ag and W-BTO substrate.³¹ In the W 4f spectrum (**Fig. 3c**), W⁶⁺ 4f_{5/2}, W⁵⁺ 4f_{5/2}, W⁶⁺ 4f_{7/2}, and W⁵⁺ 4f_{7/2} can be observed at 38.38, 36.72, 35.02, and 33.24 eV, respectively.³² To prevent photo-generated carriers from recombining under light irradiation, W⁶⁺, as a donor just below the conduction band, can be gradually transformed to unsaturated W⁵⁺, which works in W⁶⁺/W⁵⁺ pairs to improve performance (Supplementary Fig. 29c).³³ The W of Ag.W-BTO also shifted to a positive binding energy, indicating more electron transmission from W to O after Ag modification. This is attributed to the strong electron acceptor properties of O.^{34,35} All the highest peaks of the O 1s spectra can be ascribed to Ti–O. The other peaks correspond to O_v, an –OH/O–W group, and H₂O (absorbed on the surface), respectively (Supplementary Fig. 12a).³⁶ No boron species remained in samples during the synthesis process (Supplementary Fig. 12b).

Q3-2. The current approach of determining the band position of the photocatalyst using XPS-VB and absorption spectra has some methodological inaccuracies. It's important to note that XPS-VB spectra only reveal the relative position from the valence band to the Fermi level. Since your article does not establish the Fermi level's position in the catalyst, this method falls short of providing a complete picture. To accurately ascertain the band positions, Ultraviolet Photoelectron Spectroscopy (UPS) should be employed. UPS offers a more direct and reliable measurement for determining the band positions in photocatalytic materials.

Answer 3-2) We thank the referee for this professional comment. We newly measured the UPS of BTO, Ag-BTO, W-BTO, and Ag.W-BTO and accurately determined the band structure of the catalyst based on the UPS spectrum. Accordingly, we revised the corresponding figures and descriptions in the Manuscript and Supporting Information.

Supplementary Figure 15. Band gap structures. a UV-vis of BTO, W-BTO, Ag-BTO, and Ag.W-BTO. **b** estimation of band gap, **c** UPS and **d** band gap structure of BTO, W-BTO. (Revised Supporting Information on page 18)

Supplementary Figure 16. Band gap structures. a Ag-BTO and **b** Ag.W-BTO. (Revised Supporting Information on page 19)

The valence band (VB) edge potentials and Fermi levels of these catalysts were determined using Ultraviolet photoelectron spectroscopy (UPS) spectra.⁴⁴ In details, the positions of the secondary electron cutoff ($E_{\text{cut off}}$) and the valence band maximum (E_{VBM}) positions are determined using linear extrapolation of UPS.⁴⁶ As shown in Supplementary Fig. 15c, the $E_{\text{cut off}}$ values of BTO and W-BTO are 17.21 and 17.49 eV, respectively, and the E_{VBM} values of these two samples are 1.79 and 2.15 eV, respectively. The work function (ϕ) can be calculated by subtracting $E_{\text{cut off}}$ from the energy of the incident UV light ($h\nu$) after measuring the width of the emitted electrons from the onset of the secondary electrons up to the Fermi edge, according to the Formula (1).

$$\phi = h\nu - E_{\text{cut off}} \quad (1)$$

Here, the energy of HeI as a UV source ($h\nu$) is 21.22 eV. According to formula (1), the ϕ of BTO and W-BTO are calculated to be 4.01 and 3.73 eV, respectively. Consequently, the Fermi levels of BTO and W-BTO are -4.01 and -3.73 eV, respectively. The VB of BTO and W-BTO in Vacuum (E_{VAB}) were calculated using the Formula (2):

$$E_{\text{VAB}} = -(E_{\text{VBM}} + \phi) \quad (2)$$

Resulting in -5.80 and -5.88 eV, respectively. According to the E_{VAB} and band gap, the conduction band (CB) of BTO and W-BTO in Vacuum is determined at -5.99 and -5.86 eV. All values corresponding to vacuum should be replaced with NHE, resulting in a difference of -4.44 eV.⁴⁷ Therefore, Band gap structures of BTO and Ag.W-BTO are shown in Supplementary Fig. 15d. The VB of W-BTO (1.44 V) exhibited higher positivity compared with BTO (1.36 V), imparting that doping W species theoretically facilitates MOR performance. Due that the Fermi level (E_f) of metallic Ag nanoparticles ($E_f = -4.26$ eV) is more negative than that of BTO ($E_f = -4.01$ eV) and W-BTO ($E_f = -3.73$ eV) (Supplementary Fig. 16 a, b). Take Ag.W-BTO as an example, based on the strong interfacial interaction by Mott-Schottky junction, the electrons flow from W-BTO to Ag induced by the difference in E_f between Ag.W-BTO until the system reaches equilibrium, resulting in band bending and Schottky barrier formation at the interface. The suitable Schottky barrier facilitates the migration of photogenerated electrons, which also proves that CO₂RR are more likely to occur on Ag species of Ag.W-BTO during DMM synthesis process. (Revised Manuscript on pages 11 and 12)

44. An HJ, Baek SD, Kim DH, Myoung JM. Energy and charge dual transfer engineering for high-performance green perovskite light-emitting diodes. *Adv. Funct. Mater.* **32**, 2112849 (2022).
46. Wang C, *et al.* Probing effective photocorrosion inhibition and highly improved photocatalytic hydrogen production on monodisperse PANI@ CdS core-shell nanospheres. *Appl. Catal. B Environ.* **188**, 351-359 (2016).

Q3-3. In fact, the process of preparing thin films on FTO significantly influences the photocurrent and impedance. Therefore, it is crucial for the authors to provide a detailed description of the experimental procedures used for sample preparation on FTO.

Answer 3-3) We thank you for your professional comment and are pleased to clarify this issue. We added the part for preparing the samples on FTO and checking photocurrent and impedance procedure in Supporting Information:

The electrochemical photo-current and impedance (EIS) were obtained using an electrochemical workstation (CHI-660E, USA) with a three-electrode system. For working electrode preparation, 5 mg of catalyst and 20 μL of Nafion solution (5%) were dispersed into a 230 μL mixture solution including DI water (100 μL) and IPA (isopropanol, 130 μL) by sonication for 30 min. 250 μL of suspension was dropped onto the conductive side of FTO glass with the size of $1 \times 1 \text{ cm}^2$. After the sample is thoroughly dried at room temperature on the FTO glass, the mass loading of all catalysts is determined as 5 mg cm^{-2} . The Ag/AgCl electrode and Pt mesh were used as the reference electrode, and counter electrode, respectively. The 15 mL 0.1 M Na_2SO_4 with 5 mL CH_3OH aqueous solution is the electrolyte. Before the electrochemical test, the CO_2 gas flowed in the electrolyte for 30 min then kept the system closed. (Revised Supporting Information on page 3)

Q3-4. The authors should provide the Apparent Quantum Yield (AQY) of the photocatalysts, which is a crucial metric in evaluating the performance of photocatalytic materials.

Answer 3-4) We sincerely thank the referee for carefully reviewing our manuscript and providing valuable suggestions for improving its quality. We measured the performance with

different monochromatic wavelengths and calculated the Apparent Quantum Yield (AQY). The revised contents of the manuscript and Supporting Information as below:

The Apparent quantum yield (AQY) for BTO and Ag.W-BTO were measured using a range of monochromatic light band-pass filters. As depicted in Supplementary Fig. 26, the AQE of this two samples trend aligns with the UV-visible absorption spectrum, especially the AQE values of Ag.W-BTO reaching 2.15% and 1.01% at 395 nm and 420 nm, respectively. (Revised Manuscript on page 14)

Supplementary Figure 26. UV-vis absorption spectrum and wavelength-dependent AQY of BTO and Ag.W-BTO. (Revised Supporting Information on page 24)

Apparent quantum yield (AQY) calculation method

In terms of AQY, the Xe lamp was also replaced by the monochromatic LED light (CEL-LEDS35, Beijing Perfectlight Technology Co., Ltd, the wavelengths (λ) are 395, 420, 500, and 595 nm, respectively). Other experimental parameters are the same as the photocatalytic DMM production process. The light intensity was monitored by an optical power meter

(CEL-NP2000-2, Beijing Perfectlight Technology Co., Ltd). The number of incident photons (N) is calculated by equation (3), and AQY is then calculated in equation (4).

$$N = \frac{E\lambda}{hc} = \frac{I \times S \times t \times \lambda}{hc} \quad (3)$$

$$\begin{aligned} \text{AQY} &= \frac{\text{the number of reacted electrons}}{\text{the number of incident photons}} \times 100\% \\ &= \frac{[\text{CO}]_n \times 2 + [\text{CH}_4]_n \times 8 + [\text{CH}_2\text{O}]_n \times 4 + [\text{C}_3\text{H}_8\text{O}_2]_n \times 6}{N} \times 100\% \\ &= \frac{2 \times 6.02 \times 10^{23} \times n(\text{CO}) + 8 \times 6.02 \times 10^{23} \times n(\text{CH}_4) + 4 \times 6.02 \times 10^{23} \times n(\text{CH}_2\text{O}) + 6 \times 6.02 \times 10^{23} \times n(\text{C}_3\text{H}_8\text{O}_2)}{N} \times 100\% \quad (4) \end{aligned}$$

In which I was the light intensity ($\text{W} \cdot \text{m}^{-2}$), S was the irradiation area (m^2), t was the DMM production time (s), λ was the wavelength of monochromatic LED light (395, 420, 500, and 595 nm), h was Planck's constant ($6.63 \times 10^{-34} \text{ J} \cdot \text{s}$), and c was the speed of light ($3.8 \times 10^8 \text{ m} \cdot \text{s}^{-1}$). $[\text{CO}]_n$, $[\text{CH}_4]_n$, $[\text{CH}_2\text{O}]_n$ and $[\text{C}_3\text{H}_8\text{O}_2]_n$ are the number of evolved CO, CH₄, CH₂O and C₃H₈O₂ molecules, respectively. (Revised Supporting Information on pages 4 and 5)

Q3-5. The authors should provide a more rigorous explanation regarding the experiments on charge separation, such as PL, photocurrent, and transient absorption measurements. It is important to note that these tests do not replicate actual reaction conditions. Therefore, attributing observations from these tests directly to the actual reaction processes can be misleading. A more accurate interpretation of the experimental results in the context of the actual reaction conditions is needed to validate the findings related to charge separation in the photocatalytic process.

Answer 3-5) We sincerely thank the referee for valuable suggestions and apologize for the lack of rigor in the description of the charge separation experiments. Since photoluminescence (PL) and femtosecond transient absorption spectroscopy (fs-TA) measurements were conducted in different specific instruments with monochromatic wavelength (350 nm, PL and fs-TA), the great challenge remained in ensuring complete alignment with the real experiment conditions

(such as the wavelength of irradiated light, temperature, and pressure). We rigorously state that these test conditions are not actual reaction conditions. But in future work, we will pay attention to more rigorous descriptions. Therefore, we added more accurate descriptions in the manuscript as below:

The separation and transportation of electrons and holes were determined by photoluminescence (PL) measurements of the BTO, W-BTO, Ag-BTO, and Ag.W-BTO catalysts (Supplementary Fig. 38).^{52, 53} The PL measurements, which assess internal charge transfer in solid samples, are excited under the monochromatic wavelength (350 nm) that may not entirely mimic the actual reaction conditions. (Revised Manuscript on page 17)

Additionally, femtosecond transient absorption spectroscopy (fs-TA) measurements were conducted under 350 nm laser-flash photolysis to further elucidate electron transfer behavior at room temperature, with the checking conditions exhibiting disparities compared to actual reaction conditions such as temperature, pressure, and reactant composition (Revised Manuscript on page 19)

In addition, as you suggested, we retested the electrochemical photocurrent and impedance performance of catalysts under the near-actual conditions. For example, we added CH₃OH and flowed CO₂ gas in the electrolyte. The test methods and results are shown as follows:

The electrochemical photo-current and impedance (EIS) were obtained using an electrochemical workstation (CHI-660E, USA) with a three-electrode system. For working electrode preparation, 5 mg of catalyst and 20 μ L of Nafion solution (5%) were dispersed into a 230 μ L mixture solution including DI water (100 μ L) and IPA (isopropanol, 130 μ L) by sonication for 30 min. 250 μ L of suspension was dropped onto the conductive side of FTO glass with the size of 1 \times 1cm². After the sample is thoroughly dried at room temperature on the FTO glass, the mass loading of all catalysts is determined as 5 mg cm⁻². The Ag/AgCl electrode and Pt mesh were used as the reference electrode, and counter electrode, respectively. The 15 mL 0.1 M Na₂SO₄ with 5 mL CH₃OH aqueous solution is the electrolyte. Before the electrochemical test, the CO₂ gas flowed in the electrolyte for 30 min then kept the system closed. (Revised Supporting Information on page 3)

Supplementary Figure 39. Photoelectrochemical characterization. a Photo-current and **b** Electrochemical impedance spectroscopy (EIS) of BTO, Ag-BTO, W-BTO, and Ag.W-BTO. (Revised Supporting Information on page 32)

Q3-6. XPS spectra of the Ag.W-BTO photocatalysts before and after the stability test should be provided.

Answer 3-6) We sincerely thank the referee for carefully reviewing our manuscript and providing valuable suggestions for improving its quality. In Supplementary Figure 29, we added the XPS spectra of the Ag.W-BTO before and after the stability test.

Supplementary Figure 29. XPS of Ag.W-BTO before and after the stability. a Ti 2p, b Ag 3d, c W 4f, and d O 1s. (Revised Supporting Information on page 27)

The TEM, SEM images, XRD patterns and XPS spectrums after the cycling test were also recorded (Supplementary Fig. 28 and 29). Ag.W-BTO presents excellent sustainability of morphology and crystalline nature, which results in the higher stability of activity and selectivity for DMM synthesis. The valence states of Ti, Ag, W and O also showed nearly no difference before and after stability. (Revised Manuscript on page 14)

We truly thank Reviewer 3 for the insightful comments and kind suggestions.

Reviewer #4 (Remarks to the Author):

This study investigates the production of dimethoxymethane (DMM) using a photo-redox catalytic process involving a catalyst made from silver and tungsten-modified blue titanium dioxide (BTO). The authors suggest that this process combines CO₂ reduction with methanol oxidation to produce DMM. They conducted several experiments, including control tests and using isotopic reagents, to support their concept.

Although the concept of the study is stimulating, the results have some notable shortcomings. A key challenge is determining the exact source of the DMM as could be a result of both methanol oxidation and CO₂ reduction, or solely methanol oxidation. The authors conducted two main sets of experiments to back their claim of producing DMM from a combination of methanol and CO₂. First, they compared results under CO₂ with those under argon, finding seven times higher activity under CO₂. Second, they used different labeled molecules (like methanol-d₄, methanol-C¹³, and CO₂-C¹³) in their tests. While I appreciate the effort in conducting these detailed experiments, I disagree with how they interpreted the results.

Response: We sincerely appreciate your valuable suggestions, which have substantially improved the quality of our article. We have meticulously addressed each point raised, re-experimented accordingly, and provided responses and revisions as outlined below:

Q4-1. The increased activity under CO₂ could also be due to a lower pH in the solution, which might promote methanol oxidation and its subsequent reaction with formaldehyde. This implies that DMM could form without CO₂, but not without methanol (43 times less). Adjusting the pH in the solution under argon conditions could further clarify this point.

Answer 4-1) We are grateful for the time and effort Reviewer 4 has spent reviewing our manuscript. The review comments are very helpful for further strengthening the manuscript.

First, the impact of the reactants' pH on DMM formation was newly investigated according to your suggestions. In detail, we measured the pH values of two comparative samples (pure CH₃OH, and CO₂ flowing in CH₃OH), which are approximately 5-6 with minimal discernible differences by comparison of standard colors in the pH test paper (Figure R1a and b). To ensure

the authority of these results, we further measured the pH values of these two samples by using a pH meter. The pH values of the CH_3OH before and after flowing CO_2 gas are 6.256 and 5.202, respectively (Supplementary Figure 31a and b). The pH paper and meter results indicate that the flow of CO_2 can decrease the pH of the solvents but not significantly to turn it into a strongly acidic condition. The decrease in pH of the solution also confirmed that the CO_2 successfully dissolved in CH_3OH , which also serves as a prerequisite for the occurrence of the coupled reaction. Based on this, we used HCl to adjust the pH of CH_3OH to about 5 ($\text{CH}_3\text{OH} + \text{Ar}$, pH = 5) to avoid the CO_2 effect. It showed that the synthesis amount of DMM in $\text{CH}_3\text{OH} + \text{Ar}$ with pH = 5 condition is $533.20 \mu\text{mol g}^{-1}$ after a 9h reaction, while that of DMM in $\text{CH}_3\text{OH} + \text{Ar}$ and $\text{CH}_3\text{OH} + \text{CO}_2$ without adjusting pH is about 866.59 and $5702.49 \mu\text{mol g}^{-1}$, which indicates that lower pH is not the main factor affecting the production of DMM in CH_3OH (Supplementary Figure 32).

Figure R1. (a) The standard color of pH paper. (b) The pH test results of CH_3OH before and after flowing CO_2 gas.

Supplementary Figure 31. The pH values comparison of CH₃OH by a pH meter. a before and b after flowing CO₂ gas. (Revised Supporting Information on page 28)

Supplementary Figure 32. The result of pH impact on DMM yield. (Revised Supporting Information on page 29)

Based on the pH test, CO₂ can dissolve into CH₃OH in CO₂ and CH₃OH co-existence systems. Also, the formation of DMM of MOR coupling with CO₂RR (MOR + CO₂RR) is more thermodynamically than the pure MOR process by energy barrier comparison (Figure 7c). Specifically, the transition from *CH₃O to *CH₃O + *CO₂ (MOR + CO₂RR), yielding an energy barrier of -0.31 eV, is thermodynamically more favorable compared to the oxidation of *CH₃O to *CH₂O (MOR) with an energy barrier of -0.09 eV. This signifies a higher propensity for the MOR to couple with CO₂RR when CH₃OH and CO₂ coexist. Moreover, the subsequent step in the MOR pathway (*CH₂O + *CH₃O) exhibits a higher free energy barrier (0.98 eV) than *CH₃O + *COOH (0.53 eV) in the MOR + CO₂RR pathway. This further diminishes the feasibility of synthesizing DMM through the single MOR pathway. Importantly, we re-conducted the isotope labeling experiment to further support our mechanism (the results described in detail in subsequent questions).

We sincerely thank you again for your constructive suggestions. Based on your suggestions,

the experiments' results, and the DFT analysis, we revised the manuscript and supporting information as below:

To explore the influence of pH on DMM synthesis, the pH of CH₃OH before and after CO₂ exposure were tested to be 6.256 and 5.202, respectively (Supplementary Fig. 31), which indicates that the introduction of CO₂ can indeed decrease the reactant pH, which is essential for the initiation of the coupled reaction. Besides, the pH of the CH₃OH reactant was adjusted to approximately 5 by using HCl (CH₃OH + Ar, pH = 5) to avoid the CO₂ effect. It showed that the synthesis amount of DMM in CH₃OH + Ar with pH = 5 condition is 533.20 μmol g⁻¹ after a 9h reaction, while those of DMM in CH₃OH + Ar and CH₃OH + CO₂ without adjusting pH are about 866.59 and 5702.49 μmol g⁻¹, respectively, which indicates that lower pH is not the main factor affecting the production of DMM in CH₃OH (Supplementary Fig. 32). (Revised Manuscript on page 15)

To further explore the formation pathway of DMM in the coexistence of CO₂ and CH₃OH, the free energy profile and reaction pathway of the DMM synthesis were compared on the Ag.W-BTO in different routes (MOR pathway and MOR coupling with CO₂RR (MOR + CO₂RR) pathway, **Fig. 7c**). The optimized intermediate structures corresponding to each reaction steps are displayed in **Fig. 7d** and Supplementary Fig. 48. The DMM formation on Ag.W-BTO through two pathways is triggered by the thermodynamical spontaneity from CH₃OH to *CH₃OH with an energy barrier of -0.98 eV. Then, *CH₃OH is further oxidated to *CH₃O with an energy barrier of -0.65 eV. Thereafter, it was found that the process from *CH₃O to *CH₃O + *CO₂ with an energy barrier of -0.31 eV is more thermodynamically favorable than the oxidation of *CH₃O to *CH₂O with an energy barrier of -0.09 eV, which suggests that *CH₃O tends to couple with *CO₂ intermediates rather than to form individual *CH₂O intermediates in CH₃OH and CO₂ coexist system. In addition, the process of the next step (*CH₂O + *CH₃O) in MOR pathway free energy barrier (0.98 eV) requires higher than *CH₃O + *COOH (0.53 eV) in MOR + CO₂RR route. That means direct oxidation of CH₃OH to DMM is unfavorable. Moreover, three steps require energy input in the MOR + CO₂RR pathway, including the conversion of *CO₂ to *COOH intermediates, the reduction of *CO to *CH₂O, and the coupling of *CH₂O with *CH₃O. Particularly, the conversion of CO₂ to *COOH intermediates consumes the highest energy of 0.53 eV and is considered the rate-determining step during the DMM formation process. (Revised Manuscript on page 23)

Figure 7. DFT calculations. **a** Calculated CO₂ and CH₃OH adsorption energy on the different active sites (Ag, W, Ti, and O_v) of Ag.W-BTO, respectively. **b** Charge density difference and Bader charge analysis of CO₂ adsorbed on the Ag-BTO and Ag.W-BTO surface. **c** Free energy diagram of DMM production via different pathway. **d** Geometries of reaction intermediates involved in MOR coupling with CO₂RR pathways. (Revised Manuscript on page 24)

Supplementary Figure 48. Geometries of selected reaction intermediates involved in MOR pathways in the generation of DMM product. (Revised Supporting Information on page 38)

Q4-2. Additionally, the main mass spectrometry signal for DMM should be at $m/z=75$ (m/z (DMM) - m/z (H^+)), and not at 76 (much less intense) as the authors suggest. This is further supported by the observation of DMM-d7 showing a main signal at m/z 82 (Figure S27; not at 84 which is the last small peak on the figure) as one might expect. This discrepancy calls for a revision of their interpretations of the labeled experiments as following: In the experiment using $^{13}CH_3OH$ and CO_2 , the mass-to-charge ratio (m/z) of 78 for DMM indicates the presence of three ^{13}C ($75 + 3$) and not only two as expected. This finding strongly suggests that the primary source of DMM is the oxidation of methanol.

Answer 4-2) We sincerely thank the referee for the valuable suggestions. In the GC-MS of non-labeled DMM ($CH_3OCH_2OCH_3$), the molecular ion peak is $m/z=76$ which corresponds with the molecular weight of DMM (76) and the $m/z=45$ ($[CH_2OCH_3]^+$) is the base ion peak with highest relative intensity (100%). Relative to the base ion peak, the relative abundance of $m/z=75$ is the second highest, which is higher than $m/z=76$, indicating that this ion fragment

($m/z=75$, $[\text{CH}_2\text{OCH}_2\text{OCH}_3]^+$) is more stable than molecular ions fragments ($m/z=76$, $[\text{CH}_3\text{OCH}_2\text{OCH}_3]^+$) after ionized and fragmented by a mass spectrometer (Figure R2a). In our work, the carbon in the central position of the DMM originates from CO_2 , while the two carbons at the terminal positions are derived from CH_3OH . Therefore, the result of the $^{13}\text{CH}_3\text{OH} + \text{CO}_2$ isotope labeling experiment is expected to be a molecular mass of 78 ($76 + 2$) (Figure R2b). Besides, only $^{13}\text{CH}_3\text{OH}$ labeling experiment is expected to be 79 ($76 + 3$). The second intensity ion peaks of these two reactions should be 77 ($75 + 2$) for $[\text{CH}_3\text{O}^{12}\text{CH}_2\text{O}^{13}\text{CH}_2]^+$ and 78 ($75 + 3$) for $[\text{CH}_3\text{O}^{13}\text{CH}_2\text{O}^{13}\text{CH}_2]^+$, respectively, and the corresponding base ion peak should be 46 ($45 + 1$) for $[\text{CH}_3\text{O}^{12}\text{CH}_2]^+$ and 47 ($46 + 1$) for $[\text{CH}_3\text{O}^{13}\text{CH}_2]^+$, respectively (Figures R2b and R2c).

Based on the above analysis, we re-conducted $^{13}\text{CH}_3\text{OH} + \text{CO}_2$ and only $^{13}\text{CH}_3\text{OH}$ isotope labeling experiments, respectively. The result is shown in Figure 4e and Supplementary Figure 37.

Reaction	Molecular ion peak	Base ion peak	Second intensity ion peak
a $\text{CH}_3\text{OH} + \text{CO}_2 \longrightarrow \text{CH}_3\text{-O-CH}_2\text{-O-CH}_3$	76	45	75
b $^{13}\text{CH}_3\text{OH} + \text{CO}_2 \longrightarrow \text{CH}_3\text{-O-CH}_2\text{-O-CH}_3$	78	46	77
c $^{13}\text{CH}_3\text{OH} \longrightarrow \text{CH}_3\text{-O-CH}_2\text{-O-CH}_3$	79	47	78

Figure R2. The expected GC-MS (m/z) values of molecular ion peak, base ion peak, and second intensity peak in (a) $\text{CH}_3\text{OH} + \text{CO}_2$, (b) $^{13}\text{CH}_3\text{OH} + \text{CO}_2$, and (c) only $^{13}\text{CH}_3\text{OH}$ as reactants, respectively.

Figure 4. e GC-MS results of I) isotope labeled $^{13}\text{CH}_3\text{OH} + \text{CO}_2$ and II) isotope non-labeled $^{12}\text{CH}_3\text{OH} + \text{CO}_2$. (Revised Manuscript on page 13)

Supplementary Figure 37. GC-MS result of I) only isotope labeled $^{13}\text{CH}_3\text{OH}$ and II) isotope non-labeled $^{12}\text{CH}_3\text{OH} + \text{CO}_2$. (Revised Supporting information on page 31)

In the $^{13}\text{CH}_3\text{OH} + \text{CO}_2$ isotope experiment (Figure 4e-I), the $m/z=78$ ($76 + 2$) peak represents the molecular ion peak of the $[\text{}^{13}\text{CH}_3\text{O}^{12}\text{CH}_2\text{O}^{13}\text{CH}_3]^+$ structure. The base ion peak and a second intensity ion fragment peak are observed at $m/z=46$ ($45 + 1$) for $[\text{}^{13}\text{CH}_3\text{O}^{12}\text{CH}_2]^+$ and $m/z=77$ ($75 + 2$) for $[\text{}^{13}\text{CH}_3\text{O}^{12}\text{CH}_2\text{O}^{13}\text{CH}_2]^+$, respectively. Additionally, ion fragments corresponding to $[\text{}^{13}\text{CH}_3\text{O}]^+$ and $[\text{}^{13}\text{CH}_3\text{O}^{12}\text{CH}_2]^+$ exhibit m/z values of 32 and 46, respectively. These values are observed to be 1 unit higher than their counterparts from non-labeled DMM ($[\text{}^{12}\text{CH}_3\text{O}]^+$ and $[\text{}^{12}\text{CH}_3\text{O}^{12}\text{CH}_2]^+$) (Figure 4e-II), respectively, thereby confirming that the two terminal carbons originate from CH_3OH and the central carbon originates from CO_2 but not from CH_3OH .

In the $^{13}\text{CH}_3\text{OH}$ isotope experiment (Supplementary Figure 37-I), the $m/z=79$ peak corresponds to the molecular ion of the $[\text{}^{13}\text{CH}_3\text{O}^{13}\text{CH}_2\text{O}^{13}\text{CH}_3]^+$ structure, exhibiting a 3-mass unit increase compared to the $[\text{}^{12}\text{CH}_3\text{O}^{12}\text{CH}_2\text{}^{12}\text{OCH}_3]^+$ ion (Supplementary Figure 37-II). Remarkably, the base ion peak and a secondary high-intensity ion fragment peak are observed at $m/z=47$ ($45 + 2$) $[\text{}^{13}\text{CH}_3\text{O}^{13}\text{CH}_2]^+$ and $m/z=78$ ($75 + 3$) $[\text{}^{13}\text{CH}_3\text{O}^{13}\text{CH}_2\text{O}^{13}\text{CH}_2]^+$, respectively. Comparing the GC-MS results of the $^{13}\text{CH}_3\text{OH} + \text{CO}_2$ and pure $^{13}\text{CH}_3\text{OH}$ isotope experiments further confirms not only two CH_3OH oxidations but also CO_2RR involvement in our DMM formation pathway.

Based on the above analysis, the revised manuscript is shown below:

As depicted in **Fig. 4e**, the total molecular ion peak of DMM derived from non-labeled $\text{CH}_3\text{OH} + \text{CO}_2$ was 76 ($m/z=76$) (**Fig. 4e-II**), while that of the DMM derived from labeled $^{13}\text{CH}_3\text{OH} + \text{CO}_2$ reached 78 (**Fig. 4e-I**), evidently proving that the two carbon sources in DMM originate from CH_3OH and the central carbon originates from CO_2 . It is also observed that the m/z values of other fragments derived from $^{13}\text{CH}_3\text{OH} + \text{CO}_2$ (**Fig. 4e-I**), such as $[\text{}^{13}\text{CH}_3]^+$ ($m/z=16$) and $[\text{}^{13}\text{CH}_3\text{O}]^+$ ($m/z=32$), are elevated by one unit compared to DMM derived from non-labeled CH_3OH and CO_2 ($m/z=15$ $[\text{}^{12}\text{CH}_3]^+$ and $m/z=31$ $[\text{}^{12}\text{CH}_3\text{O}]^+$) (**Fig. 4e-II**). Additionally, a base ion peak and a secondary intensity ion fragment peak are detected at $m/z=46$ ($[\text{}^{13}\text{CH}_3\text{O}^{12}\text{CH}_2]^+$) and $m/z=77$ $[\text{}^{13}\text{CH}_2\text{O}^{12}\text{CH}_2\text{O}^{13}\text{CH}_3]^+$ in ^{13}C -labeled DMM (**Fig. 4e-I**) which are one and two units higher than their counterparts from non-labeled DMM $m/z=45$ ($[\text{}^{12}\text{CH}_3\text{O}^{12}\text{CH}_2]^+$ and $m/z=75$ $[\text{}^{12}\text{CH}_2\text{O}^{12}\text{CH}_2\text{O}^{12}\text{CH}_3]^+$) (**Fig. 4e-II**), respectively, suggesting that the two terminal carbons of DMM ($[\text{}^{13}\text{CH}_3\text{O}]_2^{12}\text{CH}_2$) are derived from $^{13}\text{CH}_3\text{OH}$ and the central carbon comes

from $^{12}\text{CO}_2$. In the only $^{13}\text{CH}_3\text{OH}$ labeled isotope experiment (Supplementary Fig. 37), the $m/z=79$ peak corresponds to the molecular ion of the $[\text{}^{13}\text{CH}_3\text{O}^{13}\text{CH}_2\text{O}^{13}\text{CH}_3]^+$ structure (Supplementary Fig. 37-I), exhibiting a 3-mass unit increase compared to the non-labeled $[\text{}^{12}\text{CH}_3\text{O}^{12}\text{CH}_2\text{O}^{12}\text{CH}_3]^+$ ion (Supplementary Fig. 37- II). Remarkably, the base ion peak and a secondary high-intensity ion fragment peak are observed at $m/z=47$ ($45 + 2$) $[\text{}^{13}\text{CH}_3\text{O}^{13}\text{CH}_2]^+$ and $m/z=78$ ($75 + 3$) $[\text{}^{13}\text{CH}_3\text{O}^{13}\text{CH}_2\text{O}^{13}\text{CH}_2]^+$, respectively (Supplementary Fig. 37-I). Comparing the GC-MS results of the $^{13}\text{CH}_3\text{OH} + \text{CO}_2$ (Fig. 4e-I) and pure $^{13}\text{CH}_3\text{OH}$ isotope (Supplementary Fig. 37-I) experiments further confirms not only two CH_3OH oxidations but also CO_2RR involvement in our DMM formation pathway. (Revised Manuscript on page 16)

Q 4-3) For the experiment performed using $(\text{CH}_3^{18}\text{OH} + \text{CO}_2)$ we would expect to see $^{16}\text{O}^{18}\text{O}$ -DMM with a m/z of 77 if the reaction involved CO_2 as well as methanol. Similarly, $^{18}\text{O}^{18}\text{O}$ -DMM should appear at m/z 79 and not at 80 as previously assumed. This lack of detection challenges the hypothesis that CO_2 is playing a significant role in the formation of DMM under these conditions.

Answer 4-3) Thank you very much for your professional advice, and we believe to clarify this point with your suggestions. In our $\text{CO}_2\text{RR} + \text{MOR}$ system, the two oxygens in the DMM originate from CH_3OH , not CO_2 . Therefore, the m/z of molecular ion peak of $[\text{CH}_3^{18}\text{OCH}_2^{18}\text{OCH}_3]^+$ is expected to show in 80 in the $\text{CH}_3^{18}\text{OH} + \text{CO}_2$ labeling experiment. The expected base ion peak and second highest peak should also be displayed at 47 and 79, respectively (Figure R3b). We re-checked the GC-MS of the DMM product after the $\text{CH}_3^{18}\text{OH} + \text{CO}_2$ labeling experiment (Figure 4f-I). The m/z of molecular ion peak, base ion peak, and second intensity peak of ^{18}O -labeled DMM are 80 ($[\text{CH}_3^{18}\text{OCH}_2^{18}\text{OCH}_3]^+$), 47 ($[\text{CH}_3^{18}\text{OCH}_2]^+$), and 79 ($[\text{CH}_2^{18}\text{OCH}_2^{18}\text{OCH}_3]^+$), respectively, which exhibit m/z of 4, 2, and 4 unit higher than non-labeled DMM (Figure 4f-II) ($m/z=76$ $[\text{CH}_3\text{OCH}_2\text{OCH}_3]^+$, $m/z=45$ $[\text{CH}_3\text{OCH}_2]^+$, $m/z=75$ $[\text{CH}_2\text{OCH}_2\text{OCH}_3]^+$), confirming two oxygens of DMM is sourced from CH_3OH .

Reaction		Molecular ion peak	Base ion peak	Second intensity ion peak
a	$\text{CH}_3\text{OH} + \text{CO}_2 \longrightarrow \text{CH}_3\text{-O-CH}_2\text{-O-CH}_3$	76	45	75
b	$\text{CH}_3^{18}\text{OH} + \text{CO}_2 \longrightarrow \text{CH}_3^{18}\text{-O-CH}_2\text{-O-CH}_3$	80	47	79

Figure R3. The expected GC-MS (m/z) values of molecular ion peak, base ion peak and second intensity peak in (a) $\text{CH}_3\text{OH} + \text{CO}_2$, (b) $\text{CH}_3^{18}\text{OH} + \text{CO}_2$, respectively.

Figure 4. f GC-MS results of I) isotope labeled $\text{CH}_3^{18}\text{OH} + \text{CO}_2$ and II) isotope non-labeled $\text{CH}_3^{16}\text{OH} + \text{CO}_2$. (Revised Manuscript on page 13)

The ^{18}O -labelled $\text{CH}_3^{18}\text{OH}$ with CO_2 isotope experiment ($\text{CH}_3^{18}\text{OH} + \text{CO}_2$) was conducted to trace the O atoms in the generated DMM. The m/z of molecular ion peak of the ^{18}O -labeled DMM was determined to be 80, corresponding to the molecular ion ($[(\text{CH}_3^{18}\text{O})_2\text{CH}_2]^+$) (**Fig. 4f-I**). The base ion peak and second intensity peak of ^{18}O -labeled DMM are $m/z=47$ ($[\text{CH}_3^{18}\text{OCH}_2]^+$) and $m/z=79$ ($[\text{CH}_2^{18}\text{OCH}_2^{18}\text{OCH}_3]^+$), respectively, which exhibited 2 and 4 units higher than non-labeled DMM ($m/z=45$ $[\text{CH}_3\text{OCH}_2]^+$ and $m/z=75$ $[\text{CH}_2\text{OCH}_2\text{OCH}_3]^+$) (**Fig. 4f-II**), verifying the all the O atoms in DMM are produced by the reaction of CH_3OH (**Fig. 4f**). (Revised Manuscript on pages 16 and 17)

Q 4-4) The experiment performed using unlabeled methanol and $^{13}\text{CO}_2$ show the main

m/z peak at 75 (Figure S27) in agreement with the formation of unlabeled DMM without excluding a minority of C¹³-DMM.

Answer 4-4) We thank the referee for carefully reviewing our manuscript and are pleased to clarify this issue. We re-checked the m/z of molecular ion peak ($[(\text{CH}_3\text{O})_2^{13}\text{CH}_2]^+$), the base ion peak ($[\text{CH}_3\text{O}^{13}\text{CH}_2]^+$), and the second intensity ion peak ($[\text{CH}_3\text{O}^{13}\text{CH}_2\text{OCH}_2]^+$), which are located in 77, 46, and 76 (Figure 4g-I) which are 1 unit higher than 76, 45, and 75 (Figure 4g-II), respectively (Figure R4). In addition, the m/z=15 of $[\text{CH}_3]^+$ and m/z=31 $[\text{CH}_3\text{O}]^+$ are the same as unlabeled DMM, which proves the middle position carbon originates from CO₂.

Reaction		Molecular ion peak	Base ion peak	Second intensity ion peak
a	$\text{CH}_3\text{OH} + \text{CO}_2 \longrightarrow \text{CH}_3-\text{O}-\text{CH}_2-\text{O}-\text{CH}_3$	76	45	75
b	$\text{CH}_3\text{OH} + {}^{13}\text{CO}_2 \longrightarrow \text{CH}_3-\text{O}-{}^{13}\text{CH}_2-\text{O}-\text{CH}_3$	77	46	76

Figure R4. The expected GC-MS (m/z) values of molecular ion peak, base ion peak, and second intensity peak in (a) $\text{CH}_3\text{OH} + \text{CO}_2$ and (b) $\text{CH}_3\text{OH} + {}^{13}\text{CO}_2$, respectively.

Figure 4. g The GC-MS results of I) isotope labeled $\text{CH}_3\text{OH} + {}^{13}\text{CO}_2$ and II) non-isotope

labeled $\text{CH}_3\text{OH} + {}^{12}\text{CO}_2$. (Revised Manuscript on page 13)

In **Fig. 4g**, the ion fragment peak of $[\text{CH}_3]^+$ ($m/z=15$) and $[\text{CH}_3\text{O}]^+$ ($m/z=31$) are unchanged comparing $\text{CH}_3\text{OH} + {}^{13}\text{CO}_2$ -derived DMM and $\text{CH}_3\text{OH} + {}^{12}\text{CO}_2$ -derived DMM. But, the ${}^{13}\text{CO}_2$ -derived DMM shows a base ion peak of $[\text{CH}_3\text{O}^{13}\text{CH}_2]^+$ ($m/z=46$) and the secondary intensity ion fragment peak $[\text{CH}_2\text{O}^{13}\text{CH}_2\text{OCH}_3]^+$ ($m/z=76$) (**Fig. 4g-I**), which only one m/z higher than that of the ${}^{12}\text{CO}_2$ labeled case ($m/z=45$ $[\text{CH}_2\text{CH}_3\text{O}]^+$ and $m/z=75$ $[\text{CH}_2\text{OCH}_2\text{OCH}_3]^+$) (**Fig. 4g-II**), which means middle position carbon in the DMM molecule originates from a CO_2 source. (Revised Manuscript on page 17)

Therefore, based on the results of the control experiments, isotope labeling experiments, and DFT theoretical calculations, we believe that we can more clearly explain DMM synthesis by CO_2 and CH_3OH coupling pathway.

Q4-5) Furthermore, the production of hydrogen over the catalyst contradicts the expected reaction stoichiometry, which is another point of concern. The origin of the two protons (4H^+) and two electrons (4e^-) as outlined on page 21, line 354 of the study, doesn't align with the proposed reaction formula which should involve 2 protons and 2 electrons ($2\text{CH}_3\text{OH} + \text{CO}_2 = \text{CH}_3\text{OCH}_2\text{OCH}_3$). This discrepancy raises questions about the mechanism proposed by the authors and suggests a need for a more thorough examination of the reaction pathway and the role of the catalyst.

Answer 4-5) Thank you for your professional advice. We completely agree with your opinions and offer an honest apology for the error in DMM production pathway in our reaction. We re-checked the amount of H_2 and O_2 of all prepared samples, which exhibit almost O_2 but very little H_2 production of DMM production (Figure 4a and b).

Figure 4. Photocatalysis CO_2RR coupling with MOR performance. a Gas production. b Enlarged H_2 , CO and CH_4 production. (Revised Manuscript on page 13)

In addition, according to the in-situ DRIFTS analysis, systematic isotope labeling experiments, and DFT calculations, the most probable reaction paths for the catalytic system are:

Mechanism of MOR and CO_2RR coupling pathway:

In details:

MOR:

CO_2RR :

Coupling process:

In addition, due to little H₂ being detected, our previous catalyst may have adsorbed H₂O on the surface, and/or a little CH₃OH may have undergone a self-oxidation reaction to generate DMM, which following: $3\text{CH}_3\text{OH} \rightarrow (\text{CH}_3\text{O})_2\text{CH}_2 \text{ (DMM)} + \text{H}_2 + \text{H}_2\text{O}$. Furthermore, in comparison of isotope labeling experiments of ¹³CH₃OH + CO₂ (Figure 4e-I) and ¹³CH₃OH (Supplementary Figure 37-I), the major base ion peak, major molecular ion peak, and major second intensity fragment in ¹³CH₃OH + CO₂ labeling experiment are m/z=46, 78, and 77 (Figure 4e-I), respectively, which are one unit lower than that in ¹³CH₃OH labeling experiment result (Supplementary Figure 37-I). It means that DMM originates mainly by coupling MOR and CO₂RR process ((Figure 4e-I), not by MOR (Supplementary Figure 37-I).

Figure 4. e GC-MS results of I) isotope labeled ¹³CH₃OH + CO₂ and II) isotope non-labeled CH₃OH + CO₂. (Revised Manuscript on page 13)

Supplementary Figure 37. GC-MS result of labeling experiments. I) only isotope labeled ^{13}C and II) isotope non-labeled ^{12}C + CO_2 . (Revised Supporting information on page 31)

Q 4-6) Less critical, but still noteworthy, are concerns about their choice of catalyst (BTO over other types of titanium dioxide) and the misclassification of the isotherms in their analysis (type III and not IV, no mesoporosity but textural inter-particle porosity).

Answer 4-6) We thank the reviewer for carefully reviewing our manuscript. We separately displayed these isotherms plots (Supplementary Figure 17a-d). It clearly shows the type IV isotherms with H_3 hysteresis loops of all samples from Supplementary Figure 17a-d. In addition, some pore structures are identified and highlighted with yellow circles in the SEM images (Supplementary Figure 10b). We revised the relative figures and description as shown below:

Supplementary Figure 17. N₂ sorption isotherms. a BTO, b Ag-BTO, c W-BTO and d Ag.W-BTO. e Comparison N₂ sorption isotherms and f pore sizes of BTO, Ag-BTO, W-BTO, and Ag.W-BTO. (Revised Supporting Information on page 19)

Supplementary Figure 10. b SEM images of Ag.W-BTO nano-bulks. (Revised Supporting Information on page 15)

N₂ adsorption/desorption analyses were carried out to determine the effect of doping W and depositing Ag on the pore structures (Supplementary Fig. 17).⁴⁸ The presence of mesoporous structures is implied by type IV isotherms with H₃ hysteresis loops in all prepared catalysts.⁴⁹ The pore-structure distribution curves in Supplementary Fig. 17f are evidence of the dominant mesoporous and microporous structures of all samples. The specific surface areas (SSAs) of

BTO, W-BTO, Ag-BTO, and Ag.W-BTO were 28.62, 44.68, 35.26, and 52.12 m² g⁻¹, respectively (additional parameters are shown in Supplementary Table 6). The increased specific surface area (SSA) of Ag.W-BTO is likely attributed to the porous structures (Supplementary Fig. 10b). (Revised Manuscript on page 12)

Q4-7. The IR assignments (COOH is confused with H₂O rotational peaks, miss of a band around 1700 assigned to carboxylate of adsorbed formaldehyde, etc.).

Answer 4-7) We thank you for your specialized advice and sincerely apologize for providing the wrong peak position of H₂O and carboxylate of adsorbed formaldehyde. We revised as follows:

Figure 6. In-situ DRIFTS spectroscopy characterization of CO₂ + CH₃OH on Ag.W-BTO.
b In-situ DRIFTS spectra at detailed reaction times (0, 10, 20, 30, 60, 120, 180, 240 min) with enlarged view of shaded area. (Revised Manuscript on page 20)

Q4-8. It's important to note that DMM can also be produced through the photocatalytic oxidation of methanol vapor in the presence of Bronsted acidity (like zeolites) under aerated conditions, without any oxidants or solvents (e.g. Ftouni et al. TiO₂/Zeolite Bifunctional (Photo) Catalysts for a Selective Conversion of Methanol to Dimethoxymethane: On the Role of Brønsted Acidity, *J. Phys. Chem. C* 2018 122 (51)). This raises questions about the novelty of the study as DMM. Given these issues, I cannot recommend the publication of this study in Nature Communications.

Answer 4-8) We deeply appreciate your recommendation paper and attention to the issue of innovation in our work. We carefully studied this article, Ftouni et al. have done outstanding work in which larger than 90% selective DMM was directly photocatalytic produced on TiO₂/zeolite with Brønsted acid sites by using CH₃OH as reactant under room temperature.

But comparing this work, our innovation is based on using **CO₂ for carbon neutral 2050** to prepare a C₃ product (DMM) with a selectivity of higher than 90%. Based on our new isotope labeling experiment, in-situ-FT-IR, and DFT calculations, DMM is produced by coupling CO₂RR with the MOR pathway without any sacrificial agents.

Also, to our knowledge, no previous work of DMM synthesis by photo-redox coupling scheme with CO₂ and CH₃OH has been reported. Furthermore, the mechanism and intermediate's behavior have not been investigated during the photo-redox carbon-oxygen (C–O) coupling reaction process. This study fundamentally unravels the new route for synthesizing the C₃₊ or above products by coupling CO₂ reduction integrated with organic reaction. We hope our work suggests an effective approach that allows the maximized utilization of photo-generated electrons and holes simultaneously to meet the objectives of “carbon peak and neutralization” world strategy development.

We truly thank Reviewer 4 for the insightful comments and kind suggestions.

REVIEWER COMMENTS

Reviewer #1 (Remarks to the Author):

The authors have successfully addressed the comments, and have improved the quality of the manuscript for publication.

Reviewer #2 (Remarks to the Author):

The revision has been greatly improved and the authors have addressed fully my concerns. Now it can be accepted for publication.

Reviewer #3 (Remarks to the Author):

All of my concerns have been properly addressed and I would like to recommend the acceptance of the revised manuscript in its current version.

Reviewer #4 (Remarks to the Author):

I appreciate the authors' efforts in responding to the comments. However, I remain unconvinced by their response regarding the main MS signal of the DMM being at 76. Our personal data, along with most of the reported papers on DMM, confirm that the main signal is at 45 (often not used due to other contributions), with 75 representing the second peak (used for quantification and DMM identification), while 76 is a trace (<2%). The authors can verify this information by consulting general websites reporting the MS spectrum of DMM, beyond other characteristics, such as PubChem (<https://pubchem.ncbi.nlm.nih.gov/compound/Dimethoxymethane#section=Other-MS>) and NISTbook (<https://webbook.nist.gov/cgi/cbook.cgi?ID=C109875&Units=SI&Mask=200#Mass-Spec>).

I invite the authors to investigate whether the observed 'better' activity under CO₂ might be due to impurities such as O₂ or moisture. Given the critical nature of this point and its potential to alter the interpretations reported by the authors, I cannot recommend the publication of this work.

Point-by-point Response for Nature Communications manuscript

(Manuscript ID: NCOMMS-23-63082A)

Manuscript Type: Article

Title: Photocatalytic coupling of CO₂ reduction and CH₃OH oxidation for over 90% selective dimethoxymethane production

Author(s): Yixuan Wang, Yang Liu, Lingling Wang, Silambarasan Perumal, Hongdan Wang, Hyun Ko, Chung-Li Dong, Panpan Zhang, Shuaijun Wang, Ta Thi Thuy Nga, Young Dok Kim, Yujing Ji, Shufang Zhao, Ji-Hee Kim, Dong-Yub Yee, Yosep Hwang, Jinqiang Zhang, Min Gyu Kim and Hyoyoung Lee

We are grateful to the editor, editorial staff, and reviewers for their critical comments and valuable suggestions. The manuscript has been modified after addressing all the suggestions listed below:

(Explanations of the reviewers' comments are marked in **BLUE**, and the changes made in the revision are shown with **YELLOW** highlight accordingly).

Author (s)' Points-by-points responses to Reviewers:

Reviewer #1 (Remarks to the Author):

The authors have successfully addressed the comments, and have improved the quality of the manuscript for publication.

Response: We are grateful for the time and effort Reviewer #1 has spent in reviewing our manuscript. We appreciate the referee for the kind comment.

Reviewer #2 (Remarks to the Author):

The revision has been greatly improved and the authors have addressed fully my

concernings. Now it can be accepted for publication.

Response: We are grateful for the time and effort Reviewer #2 has spent in reviewing our manuscript. We appreciate the referee for the kind comment.

Reviewer #3 (Remarks to the Author):

All of my concerns have been properly addressed and I would like to recommend the acceptance of the revised manuscript in its current version.

Response: We are grateful for the time and effort Reviewer #3 has spent in reviewing our manuscript. We appreciate the referee for the kind comment.

Reviewer #4 (Remarks to the Author):

I appreciate the authors' efforts in responding to the comments.

Response: We are grateful for the time and effort Reviewer #4 has spent reviewing our manuscript and kind suggestions. We have responded to your comments point-by-point and carefully revised the manuscript as you suggested.

Q4-1) However, I remain unconvinced by their response regarding the main MS signal of the DMM being at 76. Our personal data, along with most of the reported papers on DMM, confirm that the main signal is at 45 (often not used due to other contributions), with 75 representing the second peak (used for quantification and DMM identification), while 76 is a trace (<2%). The authors can verify this information by consulting general websites reporting the MS spectrum of DMM, beyond other characteristics, such as PubChem(<https://pubchem.ncbi.nlm.nih.gov/compound/Dimethoxymethane#section=Other-MS>) and NISTbook (<https://webbook.nist.gov/cgi/cbook.cgi?ID=C109875&Units=SI&Mask=200#Mass-Spec>).

Answer 4-1) We are very appreciative of Reviewer 4's professional comments and help in

further strengthening the manuscript. We totally agree with your opinion regarding the relative intensity $m/z=45$, 75, and 76 of DMM in the Gas Chromatography-Mass spectrometry (GC-MS) result.

In our last revision of the manuscript, the highest intensity abundance signal is $m/z=45$, and the second intensity abundance signal is $m/z=75$, not $m/z=76$ of pure DMM. The intensity of $m/z=76$ is less than the $m/z=75$ (Figure R1), which is consistent with your comments. However, the intensity of $m/z=76$ of prepared DMM (Figure R1a) and commercial DMM (Figure R1b) is about 9% and 12%, respectively, which is higher than your suggestions (<2%). This phenomenon may be observed due to different GC-MS models, intensity, and detection methods. Therefore, the detection method was optimized in this version, and the re-checked intensity of $m/z=76$ of prepared DMM and commercial DMM showed a decreased to about 4% and 3.6%, respectively (Figure R2 a and b). The result in this version is almost consistent with your comments and the database in the NIST book. **Therefore, based on your suggestions, GC-MS results and authoritative journal, the ion fragment of $m/z=75$ as a benchmark can determine the DMM (TiO₂/Zeolite Bifunctional (Photo) Catalysts for a Selective Conversion of Methanol to Dimethoxymethane: On the Role of Brønsted Acidity. *J. Phys. Chem. C* 2018, 122, 51, 29359–29367).** We re-checked all products of non-labeled and labeled DMM by optimized method and modified the relevant description in the revised Manuscript as below.

Figure R1. **a** GC-MS data of prepared DMM. **b** GC-MS data of commercial DMM by using the previous method.

Figure R2. **a** GC-MS data of prepared DMM. **b** GC-MS data of commercial DMM by using an optimized method.

Figure 4. **e** GC-MS results of I) isotope labeled $^{13}\text{CH}_3\text{OH} + \text{CO}_2$ and II) isotope non-labeled $^{12}\text{CH}_3\text{OH} + \text{CO}_2$. **f** GC-MS results of I) isotope labeled $\text{CH}_3\text{OH} + ^{13}\text{CO}_2$ and II) isotope non-labeled $\text{CH}_3\text{OH} + ^{12}\text{CO}_2$. **g** GC-MS results of I) isotope labeled $^{13}\text{CH}_3\text{OH} + ^{13}\text{CO}_2$ and II) isotope non-labeled $^{12}\text{CH}_3\text{OH} + ^{12}\text{CO}_2$. **h** GC-MS results of I) isotope labeled $\text{CH}_3^{18}\text{OH} + \text{CO}_2$

and II) isotope non-labeled $\text{CH}_3^{16}\text{OH} + \text{CO}_2$.

In the non-labeled DMM, the highest intensity abundance signal (base ion peak) is $m/z=45$ ($[\text{CH}_3\text{OCH}_2]^+$). The ion fragment of $m/z=75$ ($[\text{CH}_2\text{OCH}_2\text{OCH}_3]^+$) is more stable than molecular ion fragment ($m/z=76$, $[\text{CH}_3\text{OCH}_2\text{OCH}_3]^+$) after ionized and fragmented by a mass spectrometer. The relative abundance intensity of $m/z=75$ (the second highest intensity) is much higher than $m/z=76$. Therefore, the second highest intensity ion fragment of $m/z=75$ can determine the DMM.⁵¹ As depicted in **Fig. 4e**, the second highest intensity ion peak of DMM derived from non-labeled $\text{CH}_3\text{OH} + \text{CO}_2$ was $m/z=75$ (**Fig. 4e-II**), while that of the DMM derived from labeled $^{13}\text{CH}_3\text{OH} + \text{CO}_2$ reached $m/z=77$ (**Fig. 4e-I**), evidently proving that the two carbon sources in DMM originate from CH_3OH and the one carbon originates from CO_2 . It is also observed that the m/z values of other fragments derived from $^{13}\text{CH}_3\text{OH} + \text{CO}_2$ (**Fig. 4e-I**) such as $^{13}\text{CH}_3^+$ ($m/z=16$) and $^{13}\text{CH}_3\text{O}^+$ ($m/z=32$), and $m/z=46$ $^{13}\text{CH}_3\text{O}^{12}\text{CH}_2^+$ are elevated by one unit higher than those of DMM derived from non-labeled CH_3OH and CO_2 ($m/z=15$ $^{12}\text{CH}_3^+$, $m/z=31$ $^{12}\text{CH}_3\text{O}^+$, and $m/z=45$ ($^{12}\text{CH}_3\text{O}^{12}\text{CH}_2^+$) (**Fig. 4e-II**), suggesting that the two terminal carbons of DMM ($^{13}\text{CH}_3\text{O}^{12}\text{CH}_2$) are derived from $^{13}\text{CH}_3\text{OH}$ and the central carbon comes from $^{12}\text{CO}_2$. In **Fig. 4f**, the ion fragment peak of $[\text{CH}_3]^+$ ($m/z=15$) and $[\text{CH}_3\text{O}]^+$ ($m/z=31$) are unchanged comparing $\text{CH}_3\text{OH} + ^{13}\text{CO}_2$ -derived DMM (**Fig. 4f-II**) and $\text{CH}_3\text{OH} + ^{12}\text{CO}_2$ -derived DMM (**Fig. 4f-I**). But, the $^{13}\text{CO}_2$ -derived DMM shows a base ion peak of $[\text{CH}_3\text{O}^{13}\text{CH}_2]^+$ ($m/z=46$) and the secondary intensity ion fragment peak $[\text{CH}_2\text{O}^{13}\text{CH}_2\text{OCH}_3]^+$ ($m/z=76$) (**Fig. 4f-I**), which is only one m/z higher than that of the $^{12}\text{CO}_2$ labeled case ($m/z=45$ $[\text{CH}_2\text{CH}_3\text{O}]^+$ and $m/z=75$ $[\text{CH}_2\text{OCH}_2\text{OCH}_3]^+$) (**Fig. 4f-II**), which means middle position carbon in the DMM molecule originates from a CO_2 source. In $^{13}\text{CH}_3\text{OH} + ^{13}\text{CO}_2$ labeled experiments, it is clearly showed the base ion peak is $m/z=47$ ($^{13}\text{CH}_3\text{O}^{13}\text{CH}_2^+$) and the second intensity fragment is $m/z=78$ ($^{13}\text{CH}_2\text{O}^{13}\text{CH}_2\text{O}^{13}\text{CH}_3^+$) (**Fig. 4g-I**), respectively, which is two and three units higher than those of non-labeled $m/z=45$ ($[\text{CH}_3\text{OCH}_2]^+$) and $m/z=75$ ($[\text{CH}_2\text{OCH}_2\text{OCH}_3]^+$) (**Fig. 4g-II**), respectively. In addition, comparing the result of isotope labeled $^{13}\text{CH}_3\text{OH} + ^{12}\text{CO}_2$ experiments (**Figure 4e-I**), the fragments of $m/z=47$ ($^{13}\text{CH}_3\text{O}^{13}\text{CH}_2^+$) and $m/z=78$ ($^{13}\text{CH}_2\text{O}^{13}\text{CH}_2\text{O}^{13}\text{CH}_3^+$) showed one unit higher than $m/z=46$ ($^{13}\text{CH}_3\text{O}^{12}\text{CH}_2^+$) and $m/z=77$ ($^{13}\text{CH}_2\text{O}^{12}\text{CH}_2\text{O}^{13}\text{CH}_3^+$), respectively. This result also confirmed that the middle carbon of DMM comes from CO_2 . Furthermore, comparing the result

of isotope labeled $^{12}\text{CH}_3\text{OH}+^{13}\text{CO}_2$ experiments (**Figure 4f-I**), the fragments of $m/z=47$ ($[^{13}\text{CH}_3\text{O}^{13}\text{CH}_2]^+$) and $m/z=78$ ($[^{13}\text{CH}_2\text{O}^{13}\text{CH}_2\text{O}^{13}\text{CH}_3]^+$) showed one and two unit higher than those of $m/z=46$ ($[^{12}\text{CH}_3\text{O}^{13}\text{CH}_2]^+$) and $m/z=76$ ($[^{12}\text{CH}_2\text{O}^{13}\text{CH}_2\text{O}^{12}\text{CH}_3]^+$), respectively, which exhibited that the two terminal carbons of DMM comes from CH_3OH . In the only $^{13}\text{CH}_3\text{OH}$ labeled isotope experiment (Supplementary Fig. 37), the $m/z=78$ peak corresponds to the ion of the $[^{13}\text{CH}_2\text{O}^{13}\text{CH}_2\text{O}^{13}\text{CH}_3]^+$ structure (Supplementary Fig. 37-I), exhibiting a 3-mass unit increase compared to the non-labeled $[^{12}\text{CH}_2\text{O}^{12}\text{CH}_2\text{O}^{12}\text{CH}_3]^+$ ion (Supplementary Fig. 37-II). Remarkably, the base ion peak and second highest intensity ion peak are observed at $m/z=47$ ($45 + 2$) $[^{13}\text{CH}_3\text{O}^{13}\text{CH}_2]^+$ and $m/z=78$ ($75 + 3$) $[^{13}\text{CH}_2\text{O}^{13}\text{CH}_2\text{O}^{13}\text{CH}_3]^+$, respectively (Supplementary Fig. 37-I). Comparing the GC-MS results of the $^{13}\text{CH}_3\text{OH} + \text{CO}_2$ (**Fig. 4e-I**) and pure $^{13}\text{CH}_3\text{OH}$ isotope (Supplementary Fig. 37-I) experiments further confirmed not only two CH_3OH oxidations but also CO_2RR involvement in our DMM formation pathway. The ^{18}O -labelled $\text{CH}_3^{18}\text{OH}$ with CO_2 isotope experiment ($\text{CH}_3^{18}\text{OH} + \text{CO}_2$) was conducted to trace the O atoms in the generated DMM (**Fig. 4h**). The base ion peak and second intensity peak of ^{18}O -labeled DMM are $m/z=47$ ($[\text{CH}_3^{18}\text{OCH}_2]^+$) and $m/z=79$ ($[\text{CH}_2^{18}\text{OCH}_2^{18}\text{OCH}_3]^+$) (**Fig. 4h-I**), respectively, which exhibited two and four units higher than those of non-labeled DMM ($m/z=45$ $[\text{CH}_3\text{OCH}_2]^+$ and $m/z=75$ $[\text{CH}_2\text{OCH}_2\text{OCH}_3]^+$) (**Fig. 4h-II**), verifying the all the O atoms in DMM are produced by the reaction of CH_3OH . (Revised Manuscript on pages 16-18)

51. Ftouni, K., et al. $\text{TiO}_2/\text{Zeolite}$ Bifunctional (Photo) catalysts for a selective conversion of methanol to dimethoxymethane: on the role of brønsted acidity. *J. Phys. Chem. C* **122**, 29359-29367 (2018). (Revised Manuscript on page 35)

Q4-2) I invite the authors to investigate whether the observed 'better' activity under CO_2 might be due to impurities such as O_2 or moisture. Given the critical nature of this point and its potential to alter the interpretations reported by the authors, I cannot recommend the publication of this work.

Answer 4-2) We thank the referee very much for bringing this important question to our attention. The CH_3OH can be oxidized by O_2 to produce DMM in many reported research and industrial applications. In our case, only 99.99% of CO_2 was introduced in the CH_3OH solvent

before the reaction. And, since our reaction system is closed, O₂ in the air will not enter during the reaction process. Therefore, the O₂ detected in our system is a by-product, not an impurity, in producing the DMM process. Compared with CO₂ gas, the amount of O₂ is very small, and the detected production is about 1:1 with that of DMM, which also corresponds with the stoichiometric ratio, meaning almost no or very trace amount of O₂ reacting with CH₃OH. Combined with the results of isotope labeling experiments, the generation of DMM is mainly due to the coupling of methanol oxidation (MOR) and CO₂ reduction (CO₂RR).

Then, humid CO₂ flowed instead of CO₂ as a controlled experiment to explore the moisture effect. The produced amount of DMM is about 2307.10 μmol g⁻¹. The reason why the amount decreases compared with normal conditions (CO₂ + CH₃OH) may be due to H₂O oxidation in the reaction system, which competes with MOR. The high overpotential of H₂O oxidation to O₂ (1.23 V vs. a reversible hydrogen electrode (RHE)) than MOR (0.58 V vs. RHE) may result in slower reaction kinetics (Yuan, L., Qi, M. Y., Tang, Z. R., Xu, Y. J. Coupling strategy for CO₂ valorization integrated with organic synthesis by heterogeneous photocatalysis. *Angew. Chem. Int. Ed.* 133, 21320-21342 (2021)).

Supplementary Figure 35. The comparison of DMM production amount of humid CO₂ + CH₃OH and CO₂ + CH₃OH. (Revised Supporting information on page 30)

Humid CO₂ flowed instead of CO₂ as a controlled experiment to explore the moisture effect. The produced amount of DMM is about 2307.10 μmol g⁻¹. The reason why the amount decreases compared with normal conditions (CO₂ + CH₃OH) may be due to H₂O oxidation in the reaction system, which competes with MOR. The high overpotential of H₂O oxidation to O₂ (1.23 V vs. a reversible hydrogen electrode (RHE)) than MOR (0.58 V vs. RHE) may result in slower reaction kinetics.¹⁷ (Revised Manuscript on pages 15-16)

17. Yuan, L., Qi, M. Y., Tang, Z. R., Xu, Y. J. Coupling strategy for CO₂ valorization integrated with organic synthesis by heterogeneous photocatalysis. *Angew. Chem. Int. Ed.* **133**, 21320-21342 (2021).

The energy barrier of the reaction pathway in DFT calculations theoretically explains why DMM production shows better performance by coupling CO₂RR and MOR. The free energy profile and reaction pathway of the DMM synthesis were compared on the Ag.W-BTO in different routes (MOR pathway and MOR coupling with CO₂RR (MOR + CO₂RR) pathway, Fig. 7c). The DMM formation on Ag.W-BTO through two pathways is triggered by the thermodynamical spontaneity from CH₃OH to *CH₃OH with an energy barrier of -0.98 eV. Then, *CH₃OH is further oxidated to *CH₃O with an energy barrier of -0.65 eV. Thereafter, it was found that the process from *CH₃O to *CH₃O + *CO₂ with an energy barrier of -0.31 eV is more thermodynamically favorable than the oxidation of *CH₃O to *CH₂O with an energy barrier of -0.09 eV, which suggests that *CH₃O tends to couple with *CO₂ intermediates rather than to form individual *CH₂O intermediates in CH₃OH and CO₂ coexisting system. In addition, the process of the next step (*CH₂O + *CH₃O) in MOR pathway free energy barrier (0.98 eV) requires higher than *CH₃O + *COOH (0.53 eV) in MOR + CO₂RR route. That means DMM synthesis from the MOR + CO₂RR route is more thermodynamically favorable than the direct MOR, exhibiting that the yield of DMM in the MOR + CO₂RR route is higher than only MOR.

Figure 7. DFT calculations. **a** Calculated CO₂ and CH₃OH adsorption energy on the different active sites (Ag, W, Ti, and O_v) of Ag.W-BTO, respectively. **b** Charge density difference and Bader charge analysis of CO₂ adsorbed on the Ag-BTO and Ag.W-BTO surface. **c** Free energy diagram of DMM production via different pathway. **d** Geometries of reaction intermediates involved in MOR coupling with CO₂RR pathways.

We truly thank Reviewer #4 for the insightful comments and kind suggestions.

REVIEWERS' COMMENTS

Reviewer #4 (Remarks to the Author):

The new mass spectrometry data and clarifications provided by the author are now consistent with the literature regarding the assignment of various isotopes of dimethoxymethane. This new attribution confirms the hypothesis about the involvement of CO₂ in DMM production, and I am satisfied with the revised version.

Point-by-point Response for Nature Communications manuscript

(Manuscript ID: NCOMMS-23-63082B)

Manuscript Type: Article

Title: Coupling Photocatalytic of CO₂ reduction and CH₃OH oxidation for selective dimethoxymethane production

Author(s): Yixuan Wang, Yang Liu, Lingling Wang, Silambarasan Perumal, Hongdan Wang, Hyun Ko, Chung-Li Dong, Panpan Zhang, Shuaijun Wang, Ta Thi Thuy Nga, Young Dok Kim, Yujing Ji, Shufang Zhao, Ji-Hee Kim, Dong-Yub Yee, Yosep Hwang, Jinqiang Zhang, Min Gyu Kim and Hyoyoung Lee

We are grateful to the editor, editorial staff, and reviewers for their critical comments and valuable suggestions.

Author (s)' Points-by-points responses to Reviewer:

Reviewer #4 (Remarks to the Author):

The new mass spectrometry data and clarifications provided by the author are now consistent with the literature regarding the assignment of various isotopes of dimethoxymethane. This new attribution confirms the hypothesis about the involvement of CO₂ in DMM production, and I am satisfied with the revised version.

Response: We are grateful for the time and effort Reviewer #4 has spent in reviewing our manuscript. We appreciate the referee for the kind comment.